# Magnifying the Three Phases of GAN Training — Fitting, Refining and Collapsing

## Abstract

Generative Adversarial Networks (GANs) are efficient generative models but may suffer from mode mixture and mode collapse. We present an original global characterization of GAN training by dividing it into three successive phases — fitting, refining, and collapsing. Such a characterization underscores a strong correlation between mode mixture and the refining phase, as well as mode collapse and the collapsing phase. To analyze the causes and features of each phase, we propose a novel theoretical framework that integrates both continuous and discrete aspects of GANs, addressing a gap in existing literature that predominantly focuses on only one aspect. We develop a specialized metric to detect the phase transition from refining to collapsing and integrate it in an "early stopping" algorithm to optimize GAN training. Experiments on synthetic datasets and real-world datasets including MNIST, Fashion MNIST and CIFAR-10 substantiate our theoretical insights and highlight the efficacy of our algorithm.

## 1 Introduction

Generative Adversarial Networks (GANs) serve as a popular technique for unsupervisedly learning generative models of structured and complicated data (Goodfellow et al., 2014). GANs typically involve a generator that produces samples resembling a target dataset, and a discriminator that differentiates between real and generated samples.

One of the main challenges with GAN training is fine-tuning the combined dynamics of the discriminator and generator. Many troublesome phenomena can occur if the dynamics of the generator are not adequately matched with that of the discriminator. Among them, mode mixture (Lei et al., 2019; An et al., 2020; Tanielian et al., 2020) and mode collapse (Goodfellow, 2017) are most commonly observed. Mode collapse occurs when the generator generates limited sample varieties, while mode mixture happens when it blends different modes, producing unrealistic data.

Numerous variants of GAN have been designed to tackle these challenges through a careful blend of hyperparameter optimization and heuristics (Ioffe & Szegedy, 2015; Nowozin et al., 2016; Arjovsky et al., 2017; Li et al., 2017; Nguyen et al., 2017; Ghosh et al., 2018; Miyato et al., 2018). Another line of research focuses on developing theoretical frameworks that can better analyze and optimize GAN training (Lei et al., 2019; An et al., 2020; Sun et al., 2020; Gu et al., 2021; No et al., 2021; Becker et al., 2022; Huang & Zhang, 2023). Regarding theoretical analyses, existing literatures mainly center on the local behavior of GANs near stationary points (Sun et al., 2020; Becker et al., 2022) or the static landscape of GANs (Lei et al., 2019; An et al., 2020; Gu et al., 2021; No et al., 2021). And many of them make straightforward assumptions about the generator (No et al., 2021), the discriminator (No et al., 2021; Becker et al., 2022), or the distribution of real data (Sun et al., 2020; Becker et al., 2022). Please refer to appendix A for additional literature review.

In this paper, we present an original theoretical framework that characterizes the global behavior of GAN dynamics. We divide the training progress into three successive phases called fitting, refining, and collapsing. For each phase, we provide a thorough analysis using well-founded mathematical explanations, backed by elaborate experiments.

## 1.1 Motivation

By training the Non-Saturating GAN (Goodfellow et al., 2014) (NSGAN) on a 3-dimensional Gaussian mixture dataset and MNIST (LeCun et al., 1998), we record down the generated samples in fig. 1. In the first row, the blue dots stand for real samples drawn from the Gaussian mixture, while the generated samples are represented by orange dots. Initially, the generated samples concentrate near the origin. As training progresses, they diffuse, eventually filling the space spanned by real modes. The mode mixture issue is now the most severe. Subsequently, the generated samples are drawn near the modes and the straight lines connecting them. This process refines the generated samples and partially mitigate mode mixture. Next comes the *unexpected collapse*: the generated samples collapse rather than getting more delicate. Every multiple epochs, the number of modes in the generated samples is halved, going from eight to four, then to two, and eventually only one mode remains. This phenomenon is seldom addressed in the existing literature. Nevertheless, it challenges the conventional belief that mode collapse indicates the failure of GAN training. Indeed, if we can terminate GAN training at the opportune moment, the generated samples may showcase considerable diversity. In the second row, both the real MNIST images and the generated images are embedded using UMAP (Uniform Manifold Approximation and Projection) (McInnes et al., 2018) into the same 3-dimensional space, where a similar phenomenon has been observed.

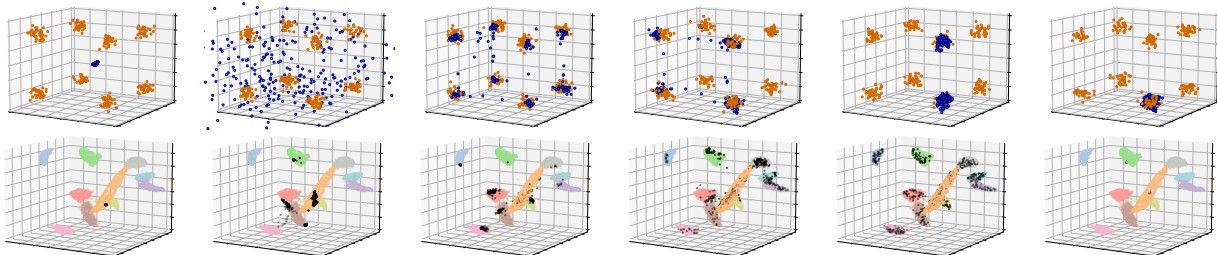

Figure 1: The real and generated samples by training NSGAN on a 3-dimensional Gaussian mixture dataset and MNIST. **First row: Gaussian mixture dataset.** Orange: Real samples. Blue: Generated samples. Epochs from left to right: 0, 10, 50, 200, 300, 360. Initially, the generated samples cluster near the origin, then spread out and occupy the space spanned by real modes. However, instead of becoming more refined, they eventually collapse to part of the modes. **Second row: MNIST embedded in a 3-dimensional space.** Colored: Real samples. Black: Generated samples. Epochs (Batches) from left to right: 0(0), 0(8), 0(32), 0(64), 32(0), 47(0). Similar phenomenon has been observed. See appendix G for more details.

Through generalization of analogous phenomena across various datasets, we introduce a novel framework delineating the three phases of GAN training: *fitting*, *refining*, and *collapsing*. For the Gaussian mixture dataset (i.e., the first row), fitting corresponds to the first two subfigures in fig. 1. Refining relates to the third subfigure. And collapsing aligns with the last three subfigures. The degree of mode mixture and mode collapse differs across the three phases, as described in table 1. Therefore, we would like to precisely identify the phase transition from refining to collapsing, and halt training at the right time. Stopping earlier results in more unrealistic samples, whereas stopping later leads to less diverse ones.

Table 1: Severity of mode mixture and mode collapse in three phases. Transitioning from fitting to refining improves sample quality by reducing mode mixture and collapse. Conversely, transitioning from refining to collapsing deteriorates sample quality, resulting in delicate yet similar samples.

|  | Mode Mixture | Mode Collapse |
|---|---|---|
| Fitting | Severe | Severe |
| Refining | Medium | Medium |
| Collapsing | Mild | Severe |

## 1.2 Our Contributions

Our contributions are threefold:

**A novel theoretical framework** based on NSGAN that integrates discrete and continuous facets of GANs through the use of particle models. Specifically speaking, in section 2.1, we propose a technique called *continuous data augmentation* to augment datasets from discrete samples so that they have continuous probability density functions. In section 2.2, we discuss about the discrete facet of NSGAN by interpreting it as a *particle model*.

**A comprehensive perspective of GANs** by presenting the novel tripartite phases of GAN training: fitting, refining, and collapsing. We highlight the characteristics of each phase through rigorous mathematical formulations and numerical experiments, as detailed in section 3, section 4, and section 5.

**A specialized metric and the induced "early stopping" algorithm** that optimizes GAN training by detecting the phase transition from refining to collapsing. We elaborate on this metric in section 5.2 and the early stopping algorithm in section 5.3. Notably, this metric is intrinsic to GAN training as it relies on information from the current state of the generator and discriminator, facilitating evaluation and optimization without direct reliance on generated or real images.

## 2 Preparatory Work

This section is dedicated to the preparatory work, including our theoretical framework and problem settings. We first discuss how to model real-world datasets in section 2.1. Therein, we introduce a novel approach called *continuous data augmentation*. We move on to discuss how to interpret Divergence GANs, particularly NSGAN, as a particle model in section 2.2. The fusion of the two methodologies forms our theoretical framework which is well-suited for analyzing GAN dynamics. All the proofs in this section can be found in appendix D.

### 2.1 How to Model Real-World Datasets?

The probability distributions of real-world datasets are usually modeled as a linear combination of Dirac measures that remain unchanged throughout GAN training (Sun et al., 2020; Becker et al., 2022). This kind of modeling may not be optimal since neural networks are typically trained from batches of data, and the batch size is relatively small compared with the size of the dataset. Therefore, the distribution of data varies substantially from batch to batch. To address such a misalignment, we propose *continuous data augmentation*, a way to augment datasets so that they have continuous probability density functions.

**Definition 2.1.** *Let $\boldsymbol{x}_1, \boldsymbol{x}_2, \ldots, \boldsymbol{x}_N \in \mathbb{R}^n$. Given a discrete probability measure $\mu = \frac{1}{N}\sum_{i=1}^{N}\delta_{\boldsymbol{x}_i}$, where $\delta_{\boldsymbol{x}_i}$ is the Dirac measure with unit mass concentrated on the point $\boldsymbol{x}_i$. The continuous data augmentation of $\mu$ is defined as a continuous probability measure $\hat{\mu}$ with density function*

$$\hat{f}(\boldsymbol{x}) = \frac{1}{N}\sum_{i=1}^{N}K_h(\boldsymbol{x}, \boldsymbol{x}_i)$$

*where $K_h(\boldsymbol{x}, \boldsymbol{y}) = K(\boldsymbol{x}/h, \boldsymbol{y}/h)/h$ is the scaled kernel with bandwidth $h > 0$.*

We always assume, if not otherwise stated, that $K$ is the Radial Basis Function (RBF) kernel

$$K(\boldsymbol{x}, \boldsymbol{y}) = \left(\sqrt{1/\pi}\right)^n \exp(-\|\boldsymbol{x} - \boldsymbol{y}\|_2^2).$$

This kind of formulation approximates the distribution of real-world datasets in a flexible way. When the bandwidth $h$ tends to 0, $\hat{\mu}$ converges to the original probability measure $\mu$. When $h$ is away from 0, samples from the continuously augmented dataset are more diversified while remaining close to certain real samples from the original dataset. Moreover, when training GANs, batches of data can be viewed as random samples from the augmented dataset, rather than remaining fixed throughout training, which better mirrors reality.

---

**Algorithm 1** Interpretation of Non-Saturating GAN (NSGAN) as a Particle Model

---
**Require:** The discriminator $d_\omega$ and the generator $g_\theta$, the noise prior $p_z$, batch size $m > 0$, stepsize $s > 0$

1: **for** number of training iterations **do**
2:     Train the discriminator $d_\omega$ as in (Goodfellow et al., 2014).
3:     Sample $\boldsymbol{z}_i$'s from the noise prior $p_z(\boldsymbol{z})$.
4:     Generate particles

$$\boldsymbol{Z}_i = g_\theta(\boldsymbol{z}_i), \quad i = 1, \ldots, m.$$

5:     Update the particles

$$\hat{\boldsymbol{Z}}_i = \boldsymbol{Z}_i + s \cdot \frac{\nabla d_\omega(\boldsymbol{Z}_i)}{2 d_\omega(\boldsymbol{Z}_i)}, \quad i = 1, \ldots, m.$$

6:     Apply the *stop gradient operator* to $\hat{Z}_i$ and update $g_\theta$ by descending its stochastic gradient:

$$\nabla_\theta \frac{1}{m} \sum_{i=1}^{m} \left\| g_\theta(\boldsymbol{z}_i) - \hat{\boldsymbol{Z}}_i \right\|_2^2.$$

7: **end for**

---

## 2.2 Rethinking Divergence GANs as Particle Models

Divergence GANs such as Vanilla GAN (Goodfellow et al., 2014), NSGAN (Goodfellow et al., 2014) and $f$-GAN (Nowozin et al., 2016) can be interpreted as *particle models* (Gao et al., 2019; Johnson & Zhang, 2019; Franceschi et al., 2023; Huang & Zhang, 2023; Yi et al., 2023). This paper focuses on the NSGAN for its practicality and conciseness. And we outline the methodology for other Divergence GANs in appendix I. The pseudocode of NSGAN as a particle model is presented in algorithm 1. We show in theorem 2.1 that algorithm 1 is essentially equivalent to the original NSGAN. Thus we also refer to generated samples as *particles* in this paper.

**Theorem 2.1.** *The update of $g_\theta$ via applying the stop gradient operator to $\hat{Z}_i$ and descending the gradient*

$$\nabla_\theta \frac{1}{m} \sum_{i=1}^{m} \left\| g_\theta(\boldsymbol{z}_i) - \hat{\boldsymbol{Z}}_i \right\|_2^2$$

*in algorithm 1 is equivalent to descending the gradient*

$$-\nabla_\theta \frac{1}{m} \sum_{i=1}^{m} \log \left( d_\omega(g_\theta(\boldsymbol{z}_i)) \right)$$

*in the original formulation of NSGAN.*

Unless otherwise stated, we assume the discriminator is optimal,[1] i.e., $d_\omega^*(\boldsymbol{x}) = p_{\text{data}}(\boldsymbol{x}) / \left( p_{\text{data}}(\boldsymbol{x}) + p_g(\boldsymbol{x}) \right)$, as established by Goodfellow et al. (2014). Under this assumption, the vector field $\nabla d_\omega(\boldsymbol{x}) / d_\omega(\boldsymbol{x})$ corresponds precisely to the velocity field of the Wasserstein gradient flow for a specific $f$-divergence, as shown by Yi et al. (2023). Consequently, it can be reformulated in terms of $r(\boldsymbol{x}) = p_{\text{data}}(\boldsymbol{x}) / p_g(\boldsymbol{x})$ as

$$\frac{\nabla d_w^*(\boldsymbol{x})}{d_w^*(\boldsymbol{x})} = \nabla r(\boldsymbol{x}) \cdot \frac{1}{r(\boldsymbol{x})(1 + r(\boldsymbol{x}))}.$$

We will use this velocity field as the foundation for our discussions throughout this paper. Specifically, we will apply this field in section 3 to derive and visualize the evolution dynamics of particles. In section 4, we will explore how the concept of steepness relates to this field, providing deeper insights into the severity of mode mixture. Finally, in section 5, we will formulate an early stopping metric based on this field, aiding in the optimization of the training process.

---

[1]For the sake of completeness, we also provide an analysis of a class of *suboptimal* discriminators in appendix C.

We make the following assumption on the noise prior $p_z(\boldsymbol{z})$ for reasons in appendix B.

**Assumption 2.1.** *Let $n$ be the dimension of real samples. We assume that the noise prior $p_z \sim \mathcal{N}(\mathbf{0}, \boldsymbol{I}_n)$ is an $n$-dimensional standard Gaussian distribution.*

# 3 The First Phase of GAN Training — Fitting

We begin with the first phase of GAN training: fitting. Roughly speaking, fitting refers to the process where the generated samples progressively spread to cover the space that envelopes the majority of the modes. To derive and visualize the evolution dynamics of particles, we employ a multiscale approach to model real-world distributions by building two models at different scales. Ensuring consistency across the analysis, the 3-dimensional Gaussian mixture used in section 1.1 serves as the basis. We introduce two minor adjustments to the original setup. Firstly, the 3-dimensional distribution is projected onto the $xy$-plane, and its associated marginal distribution is used for clearer visualization. Secondly, the covariance matrix of each Gaussian component is adjusted to $0.1\boldsymbol{I}$ to amplify the effect. The two models are elucidated respectively in section 3.1 and section 3.2.

## 3.1 Model 1: The Modes are Clustered

We first examine the scenario where the modes are clustered. Such a configuration is common in real-world datasets in the *local* sense. For instance, the two handwritten digits 1 and 7 are similar and their modes are close in MNIST (please refer to appendix E for a visualization of how close they are). In this subsection, we investigate three typical intermediate stages of GAN training: initialization, where generated samples cover all the modes, and where generated samples cover only one mode.

For initialization, we assume that $p_g$ equals the Gaussian distribution $\mathcal{N}([0,0], 0.2\boldsymbol{I}_2)$ and

$$p_{\text{data}} \sim \frac{1}{4}\mathcal{N}([1,1], 0.1\boldsymbol{I}_2) + \frac{1}{4}\mathcal{N}([1,-1], 0.1\boldsymbol{I}_2) + \frac{1}{4}\mathcal{N}([-1,1], 0.1\boldsymbol{I}_2) + \frac{1}{4}\mathcal{N}([-1,-1], 0.1\boldsymbol{I}_2).$$

The vector field that a particle is updated (i.e., $\nabla d_\omega^*(\boldsymbol{x})/d_\omega^*(\boldsymbol{x})$) is plotted in the first subfigure of fig. 2. We observe that the particles are drawn towards the nearest modes, and there is a positive correlation between the vector lengths, i.e., the intensity of attraction and the particles' distances from the modes. [2]

For the second case where the generated samples cover all the modes, we modify $p_g$ to follow the uniform distribution on $[-2,2] \times [-2,2]$. The vector field that a particle is updated (i.e., $\nabla d_\omega^*(\boldsymbol{x})/d_\omega^*(\boldsymbol{x})$) is plotted in the second subfigure of fig. 2. We observe that the particles close to the centers of the modes tend to remain stationary, whereas particles far from the modes will be updated to bring them closer to the nearest mode.[2]

Finally, we investigate the case where generated samples cover a single mode. Assume that $p_g \sim \mathcal{N}([1,1], \boldsymbol{I}_2)$, so that the generated samples cover the mode centered at $(1,1)$. Please refer to the third subfigure of fig. 2 for the plot of $\nabla d_\omega^*(\boldsymbol{x})/d_\omega^*(\boldsymbol{x})$. This time, the discriminator values are unevenly distributed near the four modes: the lowest for the mode covered by generated samples, and the highest for the mode furthest away from the generated samples. Moreover, the intensity of attraction, as indicated by vector lengths, peaks near the mode unoccupied by generated samples but diminishes near the crowded one. The intuition is that once a particle come into proximity to the unoccupied mode, it will be forcefully drawn towards it.[2]

## 3.2 Model 2: The Modes are Far Apart

In this subsection, we study the case where the modes are far apart. From a *global* perspective, real-world datasets often exhibit such a structure, particularly those with multiple categories. As an illustration, the modes of dogs and trucks in CIFAR-10 (Krizhevsky et al., 2009) are far apart because of the significant differences in their appearances (please refer to appendix E for a visualization of how far they are).

We demonstrate the difficulty for the generator to capture all the modes in such a case. Assume that

$$p_{\text{data}} \sim \frac{1}{4}\mathcal{N}([3,3], 0.1\boldsymbol{I}_2) + \frac{1}{4}\mathcal{N}([3,-3], 0.1\boldsymbol{I}_2) + \frac{1}{4}\mathcal{N}([-3,3], 0.1\boldsymbol{I}_2) + \frac{1}{4}\mathcal{N}([-3,-3], 0.1\boldsymbol{I}_2)$$

---

[2] Please refer to appendix D for the corresponding theoretical results.

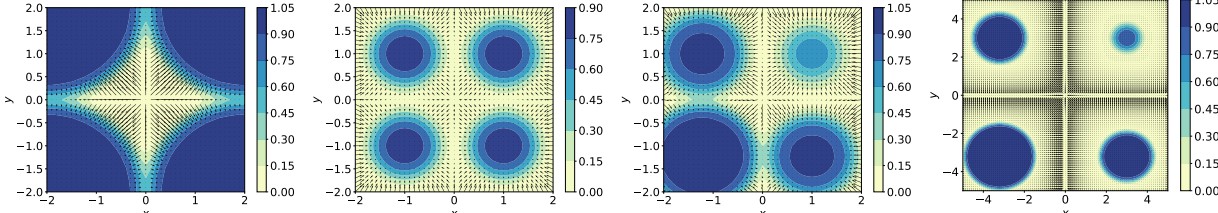

Figure 2: The vector field $\nabla d_\omega^*(\boldsymbol{x})/d_\omega^*(\boldsymbol{x})$ and the values of the discriminator in different models. **First**: Model 1 at initialization. **Second**: Model 1 with generated samples covering all modes. **Third**: Model 1 with generated samples covering only one mode. **Fourth**: Model 2.

and $p_g \sim \mathcal{N}([3,3], 3\boldsymbol{I}_2)$. The plot of $\nabla d_\omega^*(\boldsymbol{x})/d_\omega^*(\boldsymbol{x})$ can be found in the last subfigure of section 3.2. We notice a general weakening of attraction intensity near all modes, presenting challenges for particles to move towards an unoccupied mode subject to the vector field.[2] This situation could result in a "pre-refining" mode collapse or nonconvergence, preventing GAN training from advancing to the refining phase. Such collapse or nonconvergence might be attributed to detrimental network initialization or imbalanced training of the generator and the discriminator. Given its infrequency in practice, we will not delve deeper into this topic.

# 4 The Second Phase of GAN Training — Refining

This section focuses on the refining phase of GAN training, where generated samples become more refined, reducing the number of samples within the modes and alleviating the mode mixture. To measure the severity of mode mixture, we introduce a tool called "steepness." We demonstrate that in order to push $p_z$ towards the multimodal distribution $p_{\text{data}}$, the generator function $g$ must exhibit significant steepness. Using insights from the velocity field of particle evolution, we derive the formula for the evolution of steepness throughout the training process. All proofs related to this section can be found in appendix D.

## 4.1 Using Steepness to Measure the Severity of Mode Mixture

Recall that during the fitting phase, the generated samples progressively spread to cover the space that envelopes most of the modes. By the end of this phase, many generated samples will fall within these modes, resulting in severe mode mixture. According to the update rule for particles, a particle $\boldsymbol{x}$ located within the modes will be pushed in the direction of $\nabla d_w^*(\boldsymbol{x})/d_w^*(\boldsymbol{x})$, which generally points towards the nearest mode. There is a critical point between two adjacent modes where particles that start near this point are pulled apart in opposite directions. As training progresses, there exist two points $\boldsymbol{z}_1$ and $\boldsymbol{z}_2$ that are close to each other in the latent space, yet their corresponding images under the generator function $g_\theta$, namely $g_\theta(\boldsymbol{z}_1)$ and $g_\theta(\boldsymbol{z}_2)$, can be far apart. When $\boldsymbol{z}_1$ and $\boldsymbol{z}_2$ are infinitesimally close, this behavior indicates that the Jacobian of $g_\theta$ has a large entry. See fig. 3 for an illustration. This motivates the concept of "steepness" in definition 4.1, which generalizes the notion of a derivative in the one-dimensional case.

**Definition 4.1.** *Let $g : \mathbb{R}^n \to \mathbb{R}^n$ be continuously differentiable. We define its steepness $\mathcal{S}(g)$ as the upper bound of the max norm of its Jacobian, namely,*

$$\mathcal{S}(g) = \sup_{\boldsymbol{x} \in \mathbb{R}^n} \|J_g(\boldsymbol{x})\|_{\max}.$$

In the next two subsections, we will first derive the formula for the steepness of the optimal generator function. Then, we will use insights from the particle dynamics to derive how steepness evolves during the course of training. We will also give quantitative results on how steepness impacts the severity of mode mixture.

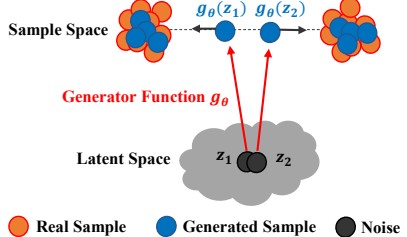 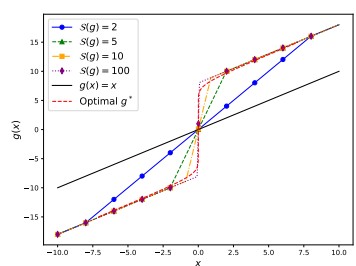 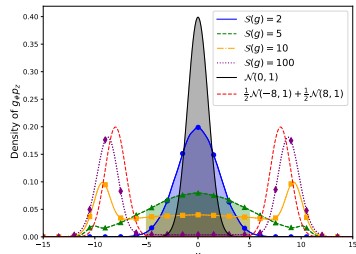

Figure 3: **Left**: Motivation of steepness. Generated samples (blue dots) are updated towards the modes (orange clusters). At certain points, small changes in the latent space cause large changes in the sample space, indicating large steepness of the generator function. **Middle**: Generator functions $g$ with varying steepness. **Right**: The density plot of $p_g = g_\# p_z$, with $p_{\text{data}} = 0.5\mathcal{N}(-8, 1) + 0.5\mathcal{N}(8, 1)$ The shaded areas represent the severity of mode mixture. Generator functions with higher steepness exhibit less severe mode mixture. Quantitative results are detailed in section 4.3.

## 4.2 Steepness of Measure-Preserving Maps

In this subsection, we study the steepness of the optimal generator function $g$ that satisfies $g_\# p_z = p_{\text{data}}$. Starting with the 1-dimensional case, we first give a full characterization of the measure-preserving maps. Let $\Phi(x)$ denotes the cumulative distribution function (CDF) of $\mathcal{N}(0, 1)$ and let $\Psi(x)$ be that of $p_{\text{data}}(x)$. Then $g = \Psi^{-1} \circ h \circ \Phi$, where $h$ is a measure-preserving map of $\mathcal{U}(0, 1)$, i.e., the uniform distribution on $(0, 1)$. Among the many measure-preserving maps with different $h$'s, special attention is given to where $h$ equals the identity map. In this instance, the corresponding $g$ is the optimal transport from $p_z$ to $p_{\text{data}}$ under the Wasserstein distance with strictly convex cost functions (Santambrogio, 2015), which includes the popular 2-Wasserstein distance as a special case. Please refer to appendix J for visualization of $g$'s with different $p_{\text{data}}$.

**Theorem 4.1.** *Assume that*

$$p_{data} \sim \frac{1}{N}\mathcal{N}(\mu_1, \sigma^2) + \frac{1}{N}\mathcal{N}(\mu_2, \sigma^2) + \cdots + \frac{1}{N}\mathcal{N}(\mu_N, \sigma^2),$$

*Here the $\mu_i$'s are in ascending order, and $\mu_{i+1} - \mu_i \geq 6\sigma$ for all $1 \leq i \leq N - 1$. Let $\Phi(x)$ denotes the cumulative distribution function (CDF) of $\mathcal{N}(0, 1)$ and let $\Psi(x)$ be that of $p_{data}(x)$. Then $g(x) := \Psi^{-1}(\Phi(x))$ satisfies*

$$\mathcal{S}(g) \geq \min_{1 \leq i \leq N-1} \sigma \cdot \exp\left(\frac{(\mu_{i+1} - \mu_i)^2}{8\sigma^2}\right) \cdot \exp(-q^2),$$

*where $q$ is the $(1 - 1/N)$-th quantile of the standard Gaussian distribution.*

We conclude that the magnitude of $\mathcal{S}(g)$ depends on the distance between two adjacent modes in 1-dimensional cases. This property can be generalized to higher dimensions, as articulated in theorem 4.2. This theorem provides an explicit lower bound of the steepness, which exhibits an exponential dependence on both the Euclidean distance $\|\bar{\boldsymbol{x}} - \boldsymbol{x}_i\|_2$ and the reciprocal of the bandwidth $1/h$. When the modes $\boldsymbol{x}_i$'s are considerably distant from each other or when the bandwidth $h$ diminishes, the steepness $\mathcal{S}(g)$ will be large.

**Theorem 4.2.** *Let $\nu$ be the truncated Gaussian distribution $\mathcal{N}_r(\mathbf{0}, \boldsymbol{I}_n)$ in the $n$-dimensional ball $\mathcal{B}_r(\mathbf{0})$ and assume that $\hat{\mu}$ is a probability measure with probability density function*

$$\hat{f}(\boldsymbol{x}) = \frac{1}{Nh}\sum_{i=1}^{N}\left(\sqrt{1/\pi}\right)^n \exp\left(-\frac{\|\boldsymbol{x} - \boldsymbol{x}_i\|_2^2}{h^2}\right).$$

*Suppose that $g \colon \mathcal{B}_r(\mathbf{0}) \to \mathbb{R}^n$ is continuously differentiable and piecewise injective. Then $\mathcal{S}(g) > M$, where*

$$M = \delta \cdot h^{1/n} \cdot \sqrt{\frac{\pi}{n}} \cdot \max_{1 \leq i \leq N} \exp\left(\frac{\|\bar{\boldsymbol{x}} - \boldsymbol{x}_i\|_2^2}{nh^2}\right).$$

*Here, $\bar{\boldsymbol{x}} = \sum_{i=1}^{N} \boldsymbol{x}_i / n$, and $\delta = \exp\left(-r^2/2\right)/\sqrt{2\pi}$.*

### 4.3 Evolution of the Generator's Steepness

We use insights from the particle dynamics to derive how steepness evolves during the course of training. Theorem 4.3 gives the recurrent formula in the setting where the optimal generator has a sufficiently large steepness $k_*$. The proof involves examining the trajectory of a designated particle throughout the training process, with detailed verification deferred to appendix D.

**Theorem 4.3.** *Assume that $p_{data} \sim \mathcal{N}(\mathbf{0}, k_*^2 \mathbf{I}_n)$ and that the discriminator is optimal, i.e., the discriminator consistently provides the precise moving direction for the particle.* [1] *Then $k_t$, the steepness of $g$ at discrete time step $t$ satisfies*

$$k_{t+1} = k_t + s \left( \frac{1}{k_t^2} - \frac{1}{k_*^2} \right) \cdot \frac{1}{1 + \frac{k_t \varphi(k_t \boldsymbol{x}_0 / k_*)}{k_* \varphi(\boldsymbol{x}_0)}},$$

*where $0 \leq t \leq T$, and $T$ is the maximum time. Here, $\varphi$ is the probability density function of $\mathcal{N}(\mathbf{0}, \mathbf{I}_n)$.*

We present quantitative results demonstrating how the steepness of generator functions affects the severity of mode mixture, as detailed in theorem 4.4. For an illustration, please refer to fig. 3, where we consider the case of $N = 2$ with $\mu_1 = -\mu_2 = -8$. In this figure, the shaded areas indicate the severity of mode mixture. Our observations reveal that generator functions with greater steepness lead to a reduction in the severity of mode mixture, which aligns with the findings in theorem 4.4.

**Theorem 4.4.** *Assume that $p_{data} \sim \sum_{i=1}^{N} \mathcal{N}(\mu_i, \sigma^2)/N$. Here, the $\mu_i$'s are in ascending order, with the condition that $\mu_i - \mu_{i-1} \geq 6\sigma$ for all $1 \leq i \leq N-1$. Furthermore, assume that the generator function $g$ is increasing and satisfies $\mathcal{S}(g) \leq k$. Additionally, assume that*

$$g^{-1} \left( \frac{\mu_i + \mu_{i+1}}{2} \right) = \Phi^{-1} \left( \Psi \left( \frac{\mu_i + \mu_{i+1}}{2} \right) \right),$$

*where $\Phi(x)$ denotes the cumulative distribution function (CDF) of the standard normal distribution $\mathcal{N}(0, 1)$, and $\Psi(x)$ is the CDF of the distribution $p_{data}(x)$. Then, the probability that the generated samples fall into the interval*

$$\bigcup_{i=1}^{N} [\mu_i + 3\sigma, \mu_{i+1} - 3\sigma],$$

*which indicates mode mixture, is at least*

$$\sum_{i=1}^{N} \left( \Phi \left( \Phi^{-1} \left( \Psi \left( \frac{\mu_i + \mu_{i+1}}{2} \right) \right) + \frac{\mu_{i+1} - \mu_i - 3\sigma}{2k} \right) - \Phi \left( \Phi^{-1} \left( \Psi \left( \frac{\mu_i + \mu_{i+1}}{2} \right) \right) - \frac{\mu_{i+1} - \mu_i - 3\sigma}{2k} \right) \right).$$

## 5 The Third Phase of GAN Training — Collapsing

This section focuses on the third phase of GAN training: collapsing. Rather than improving, the generated samples would eventually collapse to a limited number of modes. We start by offering an empirical understanding of this phenomenon in section 5.1. Following that, we introduce a metric that can detect the phase transition from refining to collapsing in section 5.2. Notably, such a metric completely originates from the velocity field of particle evolution. After theoretically deriving such a metric, we propose a novel early stopping algorithm that can judiciously halt GAN training when it nears or reaches the brink of collapse, preventing further deterioration in sample quality in section 5.3.

### 5.1 What Is the Reason for GANs' Collapsing?

We empirically demonstrate that collapsing stems from the "overfitting" of the discriminator. Overfitting generally describes the situation where a machine learning model makes precise predictions for the training data but fails to do so for unseen data. We employ this idea to denote the discriminator's propensity to give very high values to real samples and exceedingly low values to other samples. Please refer to appendix G

for a visualization of the optimal discriminator's such behavior. To see the connection, we focus on the way particles are updated in algorithm 1:

$$\hat{\boldsymbol{Z}} = \boldsymbol{Z} + s \cdot \frac{\nabla d(\boldsymbol{Z})}{2d(\boldsymbol{Z})}.$$

Here, $d$ represents the discriminator function (omitting the subscript $\omega$ for brevity), $\boldsymbol{Z}$ is a particle before update, and $\hat{\boldsymbol{Z}}$ is the same particle after update. If the discriminator "overfits", there will be a large disparity between its value within the modes and outside of the modes. Consequently, the norm of the gradient $\|\nabla d\|_2$ will reach its peak at the boundaries of the modes. This will displace $\hat{\boldsymbol{Z}}$ significantly from $\boldsymbol{Z}$ if $\boldsymbol{Z}$ is close to the boundary. The boundaries of the modes will therefore shrink, and this process will persist until all the particles have dispersed. Please refer to fig. 4 for an illustration and experimental observations.

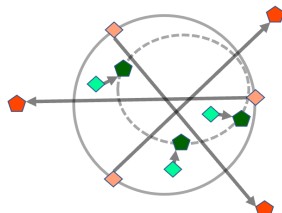 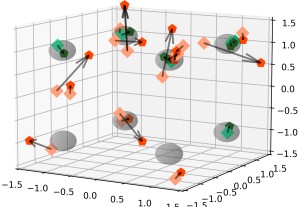

Figure 4: A depiction of why collapsing happens. **Left**: An illustration. **Right**: Experimental observations on the 3-dimensional Gaussian mixture dataset. In the illustration, the original particles (orange rhombi) near the boundary of the mode (solid gray circle / sphere) will be pushed to a significant distance (orange-red pentagons). The particles within the mode (green rhombi) will stay close to their original places (dark green pentagons). This contracts the boundary of the mode (gray dotted line).

## 5.2 Detecting the Phase Transition From Refining to Collapsing

In this section, we give a priori estimation of $\|\nabla d(\boldsymbol{x})/d(\boldsymbol{x})\|_2$ in the collapsing phase. Since collapsing manifests at the end of refining, it is important to analyze the discriminator's behavior when generated and real samples are close. Unlike the optimal discriminator in Vanilla GAN that assigns a uniform value of 0.5 to both real and generated samples when fully trained, the optimal discriminator in NSGAN assigns values near 0.5 to the central regions of modes and values near 0 to regions with scarce real samples. Amidst them, the values gradually diminish from 0.5 to 0. Please refer to appendix G for a visualization of the optimal discriminator's such behavior. This observation leads to a linear model of the discriminator in assumption 5.1. The rationale behind selecting the radius as $2\sqrt{2}h$ is given in appendix H.

**Assumption 5.1.** *Let the probability density function of the real distribution be*

$$p_{data}(\boldsymbol{x}) = \frac{1}{Nh} \sum_{i=1}^{N} \left( \sqrt{1/\pi} \right)^n \exp\left( -\frac{\|\boldsymbol{x} - \boldsymbol{x}_i\|_2^2}{h^2} \right),$$

*where we require $\min_{1 \le i,j \le N} \|\boldsymbol{x}_i - \boldsymbol{x}_j\|_2 \ge 2\sqrt{2}h$. We assume that at the end of the refining phase where generated samples closely resemble real samples, the discriminator $d(\boldsymbol{x})$ is of the form*

$$d(\boldsymbol{x}) = \begin{cases} \frac{1}{2} - \frac{\sqrt{2}}{8h}\|\boldsymbol{x} - \boldsymbol{x}_i\|_2, & \boldsymbol{x} \in B_{2\sqrt{2}h}(\boldsymbol{x}_i), \\ 0, & otherwise. \end{cases}$$

Now, we estimate $\|\nabla d(\boldsymbol{x})/d(\boldsymbol{x})\|_2$ based on assumption 5.1. Considering that collapsing typically takes place near the boundaries of the modes (refer to section 5.1), we calculate $\|\nabla d(\boldsymbol{x})/d(\boldsymbol{x})\|_2$ at $\tilde{\boldsymbol{x}}$ that locates $\sqrt{2}h$ away from $\boldsymbol{x}$. This choice originates from the fact that for Gaussian distributions, 95 percent of the samples fall within a sphere with a radius of two standard deviations from the mean. Through direct computation, we derive that $\|\nabla d(\tilde{\boldsymbol{x}})/d(\tilde{\boldsymbol{x}})\|_2 = \sqrt{2}/(2h)$. Note that this value solely relies on the bandwidth $h$ and remains unaffected by the dimension of the dataset.

---

**Algorithm 2** Early Stopping of GANs

---
**Require:** A GAN model including a generator $g_\theta$ and a discriminator $d_\omega$, the estimated bandwidth $h > 0$,
the scale factor $k_s > 0$, the number of modes $m \geq 1$, the number of warm-up training iterations $N_w$
  1: **for** number of training iterations **do**
  2:     Train the discriminator $d_\omega$ as in algorithm 1.
  3:     Compute the $(1 - 1/m)$ quantile of $\|\nabla d_\omega / d_\omega\|_2$ in a batch. Let the value be $q$.
  4:     **if** $q > k_s \cdot \sqrt{2}/(2h)$, current iteration $> N_w$ **then**
  5:         **break**
  6:     **end if**
  7:     Train the generator $g_\theta$ as in algorithm 1.
  8: **end for**

---

### 5.3   Early Stopping

We have theoretically derived $\|\nabla d(\boldsymbol{x})/d(\boldsymbol{x})\|_2$ in the collapsing phase. In this section, we propose a novel algorithm called "early stopping" that can halt GAN training at an early time when it nears or reaches the brink of collapse, preventing further deterioration in sample quality. The pseudocode is presented in algorithm 2.

There are three key ingredients: $k_s$, $m$ and $N_w$. The scale factor $k_s$ will be multiplied by $\sqrt{2}/(2h)$ to establish a threshold. Training is terminated if the value of $\|\nabla d_\omega / d_\omega\|_2$ exceeds this threshold. Then the best-performing model will be selected among the checkpoints saved before the stopping point. The scale factor is set proportional to the distances between two adjacent modes in the dataset. The underlying rationale is that when $\|\nabla d_\omega / d_\omega\|_2$ is small compared with the inter-mode distances, generated samples that are deviated from the modes can be re-attracted to the modes, causing no collapsing. Nevertheless, as $\|\nabla d_\omega / d_\omega\|_2$ grows to be comparable to the inter-mode distances, it is increasingly likely for the generated samples to gravitate towards alternate modes, posing a potential risk of mode collapse. The parameter $m$ denotes the number of modes and serves as a criterion for determining the proportion of samples within a batch that will be compared against the aforementioned threshold. Specifically, we opt to evaluate samples corresponding to the $(1 - 1/m)$th quantile. Underlying this choice is the presumption that once a particular mode initiates collapsing, it signifies the commencement of the GAN training transitioning into the collapsing phase. The number of warm-up training iterations $N_w$ is user-defined. It indicates how many training iterations GANs would undergo before transitioning into the second phase. This provision serves as a safeguard against premature cessation of training because during the fitting phase, $\|\nabla d_\omega / d_\omega\|_2$ may exceed the threshold as well.

## 6   Experiments

In this section, we present the experimental results. All the codes are available in the supplementary material.

### 6.1   Verifying Fitting and Refining

We demonstrate the existence of fitting and refining in real-world datasets. Our experiments focus on MNIST and Fashion MNIST because of the clarity of their modes. Detailed results as well as results for Fashion MNIST are deferred to appendix G. The experimental settings and rationale can be found in appendix F.

**Methodology.** We train NSGAN on MNIST and use a classification network $q(\boldsymbol{x})$ to analyze the generated images. Here, $\boldsymbol{x}$ is an image tensor, and $q(\boldsymbol{x})$ outputs a 10-dimensional vector $(p_0, p_1, \ldots, p_9)$, where $p_i \in [0, 1]$ represents the likelihood of $\boldsymbol{x}$ corresponding to the handwritten digit $i$. We count the pairs $(i, j)$ within a batch where both $p_i$ and $p_j$ exceed $10^{-2}$, and visualize the occurrences using heatmaps in fig. 5. We infer that such $\boldsymbol{x}$ exhibits a mixture of modes $i$ and $j$ by pairing $(i, j)$.

**Results.** Initially, only few entries in the heatmap are nonzero, meaning that the generated images all look similar. As training progresses, more entries become nonzero, reflecting the fitting phase where the generated

samples spread to cover the space that envelopes the modes. The off-diagnoal entries serve as indicators of mode mixture. We observe that they decrease in magnitude as training advances, validating the refining phase. However, the issue of mode mixture persists even at the closure of refining. These observations are in line with what we have derived in section 3 and section 4.

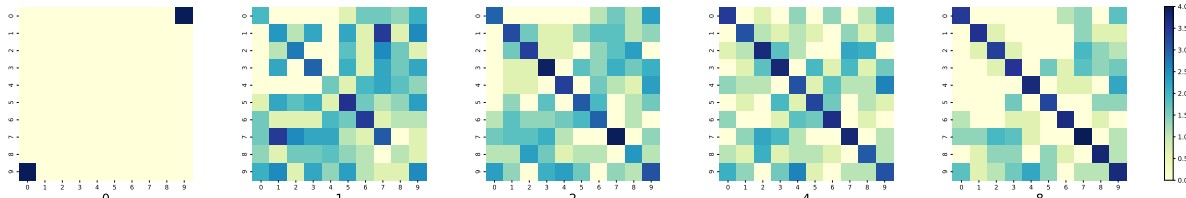

Figure 5: The logarithm of the occurrence of pairings $(i, j)$ plus 1 in a batch of size 256. Epochs from left to right: 0, 1, 2, 4, 8. At first, there are few nonzero entries. As training continues, more entries become nonzero, indicating the spread of generated samples across mode space, i.e., the fitting phase. Off-diagonal entries signal mode mixture, decreasing over training and validating the refining phase. However, mode mixture persists even after refining. The annotated heatmaps can be found in appendix G.

## 6.2 Early Stopping

We present the results of early stopping (algorithm 2) applied to both synthetic and real datasets, including 3-dimensional Gaussian mixture, MNIST, Fashion MNIST and CIFAR-10. The detailed experimental settings are provided in appendix F and the results of more training runs can be found in appendix G.

**Early stopping.** We train NSGAN on different datasets and record down $||\nabla d_\omega / d_\omega||_2$ each epoch until the maximum specified epochs. The threshold in early stopping is determined by $k_s \cdot \sqrt{2}/(2h)$, where $k_s$ equals the estimated distance between two modes and $h$ equals the estimated bandwidth. Instead of halting training when this norm exceeds the threshold, we opt to continue training, which allows us to assess the sample quality both before and after the stop. The experimental results for the 3-dimensional Gaussian mixture, MNIST, Fashion MNIST, and CIFAR-10 are shown in fig. 7.

**Comparison with FID Score and Duality Gaps.** In evaluating GAN performance, metrics are typically either domain-specific or domain-agnostic. We chose the FID score (Heusel et al., 2017) to represent the former, which focus on the quality of generated images, and duality gaps (Grnarova et al., 2019; Sidheekh et al., 2021) as a representative of the latter, which assess the optimization process itself. In fig. 8, we show that our intrinsic metric aligns with the FID score by detecting surges signaling mode collapse, effectively guiding training termination and reducing the need for checkpoints. Moreover, in appendix G we demonstrate that our metric not only aligns with duality gaps but is also more sensitive in detecting mode collapse.

**Validating the early stopping metric.** We validate the effectiveness of our early stopping metric by demonstrating that, upon applying the technique of injecting noise into the intermediate layers of the discriminator to combat mode collapse, the metric is pushed back. The results can be found in fig. 6.

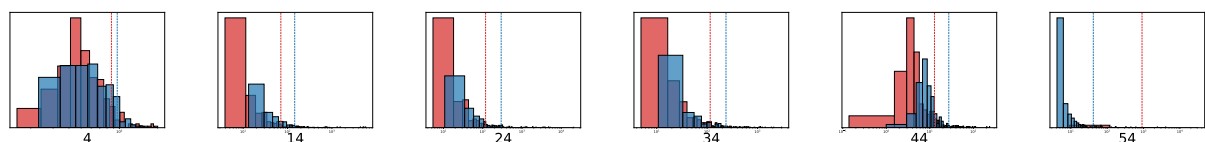

Figure 6: Histograms of the values of $||\nabla d_\omega / d_\omega||_2$ and their 90th percentile across epochs. Red for the model with noise and blue for the model without noise. The noise-free GAN collapses at the 54th epoch. Preceding the 54th, the noised model nearly always exhibits lower $||\nabla d_\omega / d_\omega||_2$ values compared to its noise-free counterpart. Post 54th epoch, this relationship reverses. Notably, in the noise-free model, $||\nabla d_\omega / d_\omega||_2$ tends towards zero, contributing to this observed divergence. See appendix G for additional results.

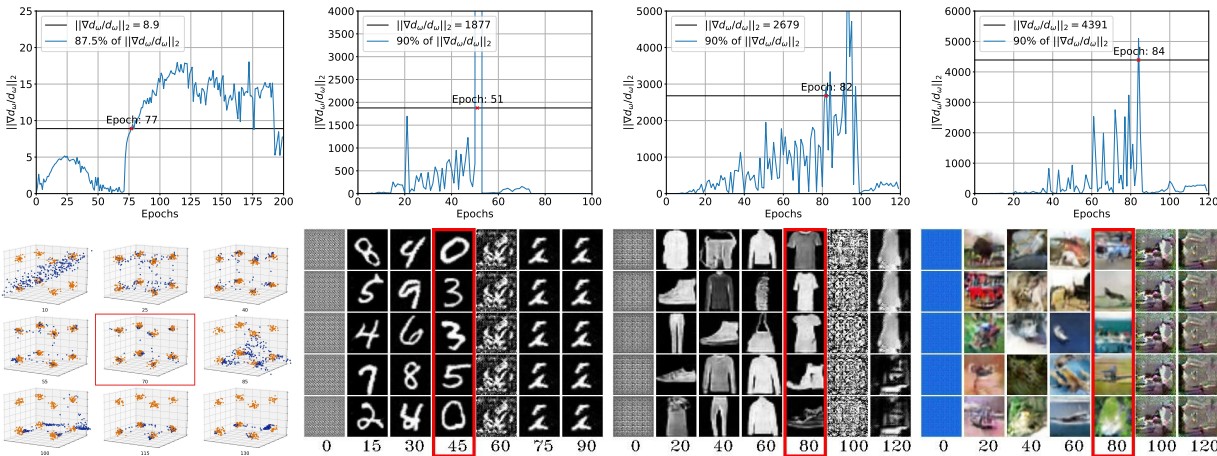

Figure 7: Experimental results of early stopping. The figures in each column, from left to right, represent the results from the following datasets: Gaussian mixture, MNIST, Fashion MNIST, and CIFAR-10, respectively. The images framed in red are the most realistic ones within the presented ones. **Gaussian mixture**: The generated samples consistently demonstrate high quality after the fitting phase. As training approaches the stopping point, a discernible deviation in sample quality is observed. Eventually, the samples collapse to half of the modes. **MNIST**: Prior to the stopping point, the generated samples exhibit high quality and variety. After the stopping point, they are first contaminated by noise. And then collapse to a specific mode. **Fashion MNIST**: Before the stopping point, the generated images demonstrate both clarity and diversity. After the stopping point, they first become noise, and then collapse to some of the modes. **CIFAR-10**: Before the stopping point, the generated samples were crisp and varied. After the stopping point, they collapse and transition between different modes. The results of more training runs can be found in appendix G

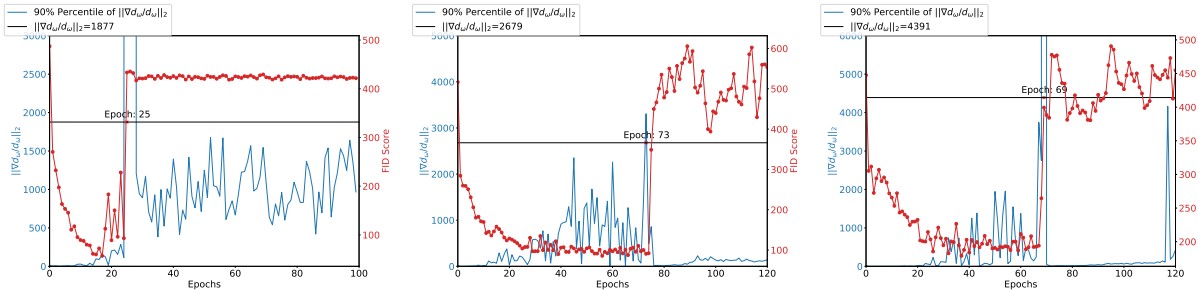

Figure 8: The tendency of $\|\nabla d_\omega/d_\omega\|_2$ and the FID scores on MNIST, Fashion MNIST and CIFAR-10, from left to right. Red circled for the FID scores and blue for our metric. A consistent pattern is observed: Whenever our metric surges past the threshold, the FID scores nearly concurrently escalate to high values, signifying a notable deterioration in sample quality. Please refer to appendix G for additional results.

# 7 Conclusion

This study presents a tripartite framework for GAN training, encompassing the phases of fitting, refining, and collapsing. Through rigorous mathematical formulations and corroborative numerical experiments, we highlight the characteristics of each phase. Our comprehensive examination unveils the intricate dynamics associated with the challenges of mode collapse and mode mixture. Moreover, to enhance the efficiency and robustness of GAN training, we introduced a novel early stopping algorithm that is intrinsic to GAN training and does not rely on extrinsic metrics like the FID score. In the future, we plan to enhance our early stopping algorithm for multi-modal datasets and various GAN variants. We also aim to extend its applicability to other generative models. For additional discussions, please refer to appendix K.

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

## A  Additional Literature Review

In this section, we provide more detailed literature review.

**Generative models.** Learning the generative model based on large amounts of data is a fundamental task in machine learning and statistics. Popular techniques include Variational Autoencoders (Kingma & Welling, 2014; Chen et al., 2017; Razavi et al., 2019; Child, 2021), Generative Adversarial Networks (Goodfellow et al., 2014; Radford et al., 2016; Arjovsky et al., 2017; Gulrajani et al., 2017; Nguyen et al., 2017; Ghosh et al., 2018; Lin et al., 2018; Brock et al., 2019; Karras et al., 2020), flow-based generative models (Dinh et al., 2017; Kingma & Dhariwal, 2018; Chen et al., 2019; Grathwohl et al., 2019), autoregressive models (Van den Oord et al., 2016; Van Den Oord et al., 2016), energy-based models (Xie et al., 2018; Gao et al., 2021) and diffusion models (Ho et al., 2020; Song et al., 2021; Karras et al., 2022). Among these models, GANs' ability for rapid sampling, unsupervised feature learning and broad applicability makes them the primary focus of this study.

**Practical considerations of GANs.** In the realm of GAN, mode collapse (Goodfellow, 2017) is one of the major challenges which has received a lot of attention. It refers to the situation where the generator produces samples on only a few modes instead of the entire data distribution. The issue of mode collapse has been addressed mainly from three perspectives: modifying the network architecture, designing new objective functions and using normalization techniques. Regarding the network architecture, existing approaches involve increasing the number of generator (Ghosh et al., 2018) or discriminator (Nguyen et al., 2017), using joint architectures (Larsen et al., 2016). From the objective function side, various metrics such as the Wasserstein distance (Arjovsky et al., 2017), $f$-divergence (Nowozin et al., 2016), least squares distance (Mao et al., 2017), maximum mean discrepancy (Li et al., 2017) are employed. Normalization techniques such as batch normalization (Ioffe & Szegedy, 2015), layer normalization (Ba et al., 2016) and spectral normalization (Miyato et al., 2018) have also achieved superb empirical performance. Mode mixture (Lei et al., 2019) is another troublesome phenomenon in which the generated samples fall outside the real distribution and are thus unrealistic. Existing approaches include picking generated samples using a rejection sampling method (Tanielian et al., 2020), or generating samples with discontinuous optimal transport rather than deep neural networks (Lei et al., 2019; An et al., 2020; Gu et al., 2021).

**Theoretical frameworks of GANs.** Another line of research approaches mode collapse and mode mixture by establishing theoretical frameworks for better analyzing and optimizing GAN training. These researches fall into two categories: landscape analysis and dynamic analysis. Landscape analysis is static because it simply examines the results of GAN training; it ignores the interaction between the discriminator and generator during training. For instance, Sun et al. (2020) analyzed the landscape of a family of GANs called separable-GAN. They proved that the landscape of separable-GAN has exponentially many bad basins, all of which are deemed as mode-collapse. In contrast, No et al. (2021) demonstrated that Wasserstein GAN with an infinitely broad generator has no spurious stationary points by modeling both the generator and the discriminator using random feature theory. Lei et al. (2019) used results from optimal transport theory to account for mode mixture. Dynamic analysis, on the other hand, considers how the discriminator and generator interact. Franceschi et al. (2022) considered GANs from the perspective of Neural Tangent Kernel (NTK). Becker et al. (2022) suggested the "Isolated Points Model" to explain the causes of GANs' instability. Another dynamical way of modeling GANs is to regard it as a particle model (Huang & Zhang,

2023; Franceschi et al., 2023). This kind of modeling is used in conjunction with Fokker–Planck equation theories by Huang and Zhang to demonstrate the convergence of GANs to the global stationary point (Huang & Zhang, 2023).

**Relation of GANs and particle models.** There has been an emerging trend in recent years to conceptualize GANs as particle models. We present the interpretation of NSGAN as a particle model in algorithm 1. Huang & Zhang (2023) examined a similar interpretation of vanilla GAN, but did not specifically discuss NSGAN. Gao et al. (2019) used a variational gradient flow approach to analyse GANs, without placing much emphasis on the connection to particle models. Franceschi et al. (2023) unified GANs within the context of particle models and interpret GANs as "interactive particle models".

**Phased process in diffusion models.** Recently, analogous phase transition phenomena, akin to those elucidated in our paper, have been uncovered in diffusion models. For example, Biroli et al. (2024) showed that the generative process in diffusion models undergoes a "speciation" transition, revealing data structure from noise, followed by a "collapse" transition, converging dynamics to memorized data points, akin to condensation in a glass phase. Sclocchi et al. (2024) found that the backward diffusion process acting after a time $t$ is governed by a phase transition at some threshold time, where the probability of reconstructing high-level features suddenly drops and the reconstruction of low-level features evolves smoothly across the whole diffusion process. Li & Chen (2024) studied properties of critical windows that are are narrow time intervals in sampling during which particular features of the final image emerge.

# B  Choice of Latent Dimension

In this section, we provide the rationale behind our choice of the latent dimension in assumption 2.1. At the population level, Yi et al. (2023) demonstrated that NSGAN minimizes the $f$-divergence $D_f(p_{\text{data}}\|p_g)$ with

$$f(u) = -(u+1)\log\frac{u}{u+1} + u(1 - 2\log 2) - 1.$$

Let $\mu$ and $\nu$ be mutually singular measures on $\mathbb{R}^n$, Yang et al. (2022) proved that

$$D_f(\mu\|\nu) = f(0) + f^*(0) > 0,$$

where $f^*$ stands for the Fenchel conjugate of $f$. If the latent dimension is less than $n$, then $g_{\theta\#}p_z$ is supported on a low-dimensional manifold, so that $g_{\theta\#}p_z$ and $\nu$ will be mutually singular. Thus there is always a positive gap in $f$-divergence between $g_{\theta\#}p_z$ and $\nu$. In other words, $g_{\theta\#}p_z$ cannot approximate $\nu$ well even if the GAN model has been trained perfectly. To prevent such inherent misalignment, we assume that the latent dimension always equals $n$. Combined with the continuous data augmentation of real-world datasets, we assume that the noise prior $p_z(\boldsymbol{z})$ is an $n$-dimensional standard Gaussian distribution, denoted as $\mathcal{N}(\boldsymbol{0}, \boldsymbol{I}_n)$, where $n$ is the dimension of real samples.

# C  Analyses of a Class of Suboptimal Discriminators

## C.1  The Class of Suboptimal Discriminators

In this section, we consider a class of suboptimal discriminators that can be represented as

$$\hat{d}_\omega(\boldsymbol{x}) = \frac{p_{\text{data}}(\boldsymbol{x})}{p_{\text{data}}(\boldsymbol{x}) + f(r(\boldsymbol{x})) \cdot p_g(\boldsymbol{x})},$$

where $r(\boldsymbol{x}) = p_{\text{data}}(\boldsymbol{x})/p_g(\boldsymbol{x})$ is the density ratio and $f$ is a scalar function. When $f \equiv 1$, the discriminator is optimal, as established by (Goodfellow et al., 2014). This representation arises from the training procedure of the discriminator, which effectively trains a binary classifier. This process is equivalent to using gradient descent to estimate $r(\boldsymbol{x})$. The function $f(r(\boldsymbol{x}))$ reflects the error between the true value and the estimated value because

$$\hat{d}_\omega(\boldsymbol{x}) = \frac{1}{1 + f(r(\boldsymbol{x}))\big(p_{\text{data}}(\boldsymbol{x})/p_g(\boldsymbol{x})\big)^{-1}} = \frac{1}{1 + \big(r(\boldsymbol{x})/f(r(\boldsymbol{x}))\big)^{-1}}.$$

In this context, the role of $f(r(\boldsymbol{x}))$ becomes clear — it measures the deviation of the suboptimal discriminator from the optimal one. When $f(r(\boldsymbol{x}))$ deviates from 1, it indicates that the discriminator is not perfectly distinguishing between the real and generated data, thus introducing some bias or error into the estimation process. Such a framework allows us to analyze and understand the behavior of suboptimal discriminators and their impact on the overall performance of the generative adversarial network.

## C.2 The Influence of the Suboptimal Discriminator to the Vector Field

In this subsection, we investigate the influence of the suboptimal discriminator on the vector field that governs the movement of particles. This analysis complements the discussion in section 3.

**Proposition C.1.** *Assume that $f \in C^2(0, +\infty)$. Then, at a point $\boldsymbol{x}$ where $p_{data}(\boldsymbol{x})p_g(\boldsymbol{x}) > 0$, the cosine of the angle $\theta$ between the suboptimal vector $\nabla \hat{d}_\omega(\boldsymbol{x})/(2\hat{d}_\omega(\boldsymbol{x}))$ and the optimal vector $\nabla d^*_\omega(\boldsymbol{x})/(2d^*_\omega(\boldsymbol{x}))$ is given by*

$$\cos \theta = \frac{\langle \nabla \hat{d}_\omega(\boldsymbol{x}), \nabla d^*_\omega(\boldsymbol{x}) \rangle}{\left\| \nabla \hat{d}_\omega(\boldsymbol{x}) \right\|_2 \left\| \nabla d^*_\omega(\boldsymbol{x}) \right\|_2} = \mathrm{sign}\left( \frac{f(r(\boldsymbol{x}))}{r(\boldsymbol{x})} - f'(r(\boldsymbol{x})) \right).$$

*Consequently, there exists $\delta > 0$ that depends on $f$ such that whenever $r(\boldsymbol{x}) < \delta$, the two vectors are in the same direction.*

*Proof.* To calculate the angle between two vectors, we can ignore their scalar coefficients. Therefore, we only need to determine the angle between $\nabla \hat{d}_\omega(\boldsymbol{x})$ and $\nabla d^*_\omega(\boldsymbol{x})$. Using the results derived in theorem D.2, this calculation reduces to finding the angle between

$$-p_{\mathrm{data}}(\boldsymbol{x})\nabla p_g(\boldsymbol{x}) + p_g(\boldsymbol{x})\nabla p_{\mathrm{data}}(\boldsymbol{x})$$

and

$$- p_{\mathrm{data}}(\boldsymbol{x})\nabla\big(\alpha(\boldsymbol{x}) \cdot p_g(\boldsymbol{x})\big) + \big(\alpha(\boldsymbol{x}) \cdot p_g(\boldsymbol{x})\big)\nabla p_{\mathrm{data}}(\boldsymbol{x})$$
$$= \alpha(\boldsymbol{x})\big( - p_{\mathrm{data}}(\boldsymbol{x})\nabla p_g(\boldsymbol{x}) + p_g(\boldsymbol{x})\nabla p_{\mathrm{data}}(\boldsymbol{x})\big) - p_{\mathrm{data}}(\boldsymbol{x})p_g(\boldsymbol{x})\nabla\alpha(\boldsymbol{x}),$$

where $\alpha(\boldsymbol{x}) = f(r(\boldsymbol{x}))$. We use the same technique and divide both vectors by the scalar $p_{\mathrm{data}}(\boldsymbol{x})p_g(\boldsymbol{x})\alpha(\boldsymbol{x})$. By applying the chain rule, we only need to compute the angle between

$$-\nabla \log p_g(\boldsymbol{x}) + \nabla \log p_{\mathrm{data}}(\boldsymbol{x}) = \nabla \log r(\boldsymbol{x})$$

and

$$\nabla \log r(\boldsymbol{x}) - \nabla \log \alpha(\boldsymbol{x}).$$

We proceed with the final calculations:

$$\cos \theta = \frac{\langle \nabla \log r(\boldsymbol{x}), \nabla \log(r(\boldsymbol{x})/f(r(\boldsymbol{x}))) \rangle}{\|\nabla \log r(\boldsymbol{x})\|_2 \|\nabla \log(r(\boldsymbol{x})/f(r(\boldsymbol{x})))\|_2}.$$

For the numerator, we have $\nabla \log r(\boldsymbol{x}) = \nabla r(\boldsymbol{x})/r(\boldsymbol{x})$, and

$$\nabla \log f(r(\boldsymbol{x})) = \frac{f'(r(\boldsymbol{x}))}{f(r(\boldsymbol{x}))} \cdot \nabla r(\boldsymbol{x}),$$

implying that $\nabla \log r(\boldsymbol{x})$ and $\nabla \log(r(\boldsymbol{x})/f(r(\boldsymbol{x})))$ are both parallel to $\nabla r(\boldsymbol{x})$. Therefore,

$$\cos \theta = \mathrm{sign}\left( \frac{1}{r(\boldsymbol{x})} - \frac{f'(r(\boldsymbol{x}))}{f(r(\boldsymbol{x}))} \right)$$
$$= \mathrm{sign}\left( \frac{f(r(\boldsymbol{x}))}{r(\boldsymbol{x})} - f'(r(\boldsymbol{x})) \right).$$

By the continuity of $f''$, there exists $\varepsilon > 0$ such that for $x \in [0, \varepsilon)$, we have $|f''(x)| < M$. As a result, for $\boldsymbol{x}$ such that $r(\boldsymbol{x}) < \delta := \min\big(\varepsilon, \sqrt{2f(0)/M}\big)$, we have

$$\cos \theta = \mathrm{sign}\big(f(r(\boldsymbol{x})) - r(\boldsymbol{x})f'(r(\boldsymbol{x}))\big) = \mathrm{sign}\big(f(0) + r(\boldsymbol{x})^2 f''(\xi)/2\big) = 1$$

for some $\xi \in (0, r(\boldsymbol{x}))$, where we use Taylor's expansion with the Lagrange remainder. □

We now briefly discuss the implications of proposition C.1. Firstly, this proposition considers $f \equiv 1$ as a special case, in which $\cos\theta = 1$ for any choice of $\boldsymbol{x}$. Secondly, although the proposition seems to hold only for $\boldsymbol{x}$ where $r(\boldsymbol{x})$ is small, this is sufficient for our purposes. In this subsection, we are focusing on the fitting phase, where $r(\boldsymbol{x})$ is typically small for $\boldsymbol{x} \sim p_g(\boldsymbol{x})$. Finally, it may seem counter-intuitive that the vector field of the suboptimal discriminator aligns perfectly with that of the optimal discriminator. However, it is important to note that while the directions of these two vector fields may be the same, their magnitudes can differ. We choose not to delve further into this topic because the magnitudes can be adjusted by varying the step sizes.

### C.3 The Influence of the Suboptimal Discriminator to the Evolution of Steepness

In this subsection, we investigate the influence of the suboptimal discriminator on the evolution of steepness. This analysis complements the discussion in section 4.

**Proposition C.2.** *Assume that $p_{data} \sim \mathcal{N}(\boldsymbol{0}, k_*^2 \boldsymbol{I}_n)$ and that the discriminator is suboptimal and takes the form*

$$\hat{d}_\omega(\boldsymbol{x}) = \frac{p_{data}(\boldsymbol{x})}{p_{data}(\boldsymbol{x}) + f(r(\boldsymbol{x})) \cdot p_g(\boldsymbol{x})},$$

*where $r(\boldsymbol{x}) = p_{data}(\boldsymbol{x})/p_g(\boldsymbol{x})$, and $f$ is a function measuring the deviation of $\hat{d}_\omega(\boldsymbol{x})$ from the optimal discriminator. Then $k_t$, the steepness of $g$ at discrete time step $t$ satisfies*

$$k_{t+1} = k_t + s\left(\frac{1}{k_t^2} - \frac{1}{k_*^2}\right) \cdot \frac{f(r(k_t\boldsymbol{x}_0)) - r(k_t\boldsymbol{x}_0)f'(r(k_t\boldsymbol{x}_0))}{r(k_t\boldsymbol{x}_0) + f(r(k_t\boldsymbol{x}_0))},$$

*where $0 \leq t \leq T$, and $T$ is the maximum time. Here, $\varphi$ is the probability density function of $\mathcal{N}(\boldsymbol{0}, \boldsymbol{I}_n)$ and*

$$r(k_t\boldsymbol{x}_0) = \frac{k_t\varphi(k_t\boldsymbol{x}_0/k_*)}{k_*\varphi(\boldsymbol{x}_0)}.$$

*Proof.* Let $\varphi(\boldsymbol{x})$ be the probability density function of the $n$-dimensional standard Gaussian distribution

$$\varphi(\boldsymbol{x}) = \frac{1}{\sqrt{(2\pi)^n}} \cdot \exp\left(-\frac{1}{2}\boldsymbol{x}^\top\boldsymbol{x}\right).$$

Then the probability density function of $\mathcal{N}(\boldsymbol{0}, k^2\boldsymbol{I}_n)$ is $\varphi(\boldsymbol{x}/k)/k$. Let $\boldsymbol{x}_t = k_t\boldsymbol{x}_0$ denotes the position of the particle at time $t$. Here, $k_t$ represents the steepness of the generator function. We investigate the evolution of the particle subject to the vector field given by $\nabla\hat{d}_\omega(\boldsymbol{x})/\hat{d}_\omega(\boldsymbol{x})$, which can be written in terms of $r(\boldsymbol{x})$ as

$$\boldsymbol{x}_{t+1} = \boldsymbol{x}_t + s \cdot \frac{\big(f(r(\boldsymbol{x}_t)) - r(\boldsymbol{x}_t)f'(r(\boldsymbol{x}_t))\big)\nabla r(\boldsymbol{x}_t)}{r(\boldsymbol{x}_t)\big(r(\boldsymbol{x}_t) + f(r(\boldsymbol{x}_t))\big)}, \quad t = 1, 2, \ldots, T.$$

By the formula of $\varphi(\boldsymbol{x})$, we deduce that $\nabla\varphi(\boldsymbol{x}) = -\varphi(\boldsymbol{x})\boldsymbol{x}$. Below we compute $\nabla r(\boldsymbol{x})$ by the chain rule:

$$\nabla r(\boldsymbol{x}) = \frac{k_t}{k_*} \cdot \frac{\nabla\varphi(\boldsymbol{x}/k_*) \cdot \varphi(\boldsymbol{x}/k_t) - \varphi(\boldsymbol{x}/k_*)\nabla\varphi(\boldsymbol{x}/k_t)}{\varphi(\boldsymbol{x}/k_t)^2}$$

$$= \frac{k_t}{k_*} \cdot \left(\frac{1}{k_t^2} - \frac{1}{k_*^2}\right) \cdot \frac{\varphi(\boldsymbol{x}/k_*)}{\varphi(\boldsymbol{x}/k_t)} \cdot \boldsymbol{x}.$$

Using $\boldsymbol{x}_t = k_t\boldsymbol{x}_0$, we derive the following recurrent formula for $\{k_t\}_{t=0}^T$:

$$k_{t+1} = k_t + s\left(\frac{1}{k_t^2} - \frac{1}{k_*^2}\right) \cdot \frac{f(r(k_t\boldsymbol{x}_0)) - r(k_t\boldsymbol{x}_0)f'(r(k_t\boldsymbol{x}_0))}{r(k_t\boldsymbol{x}_0) + f(r(k_t\boldsymbol{x}_0))},$$

where

$$r(k_t\boldsymbol{x}_0) = \frac{k_t\varphi(k_t\boldsymbol{x}_0/k_*)}{k_*\varphi(\boldsymbol{x}_0)}.$$

$\square$

Note that this proposition considers $f \equiv 1$ as a special case, leading to the same conclusion as in theorem 4.3.

# D    Proofs to Theorems

Here, we aggregate all the theorems presented in the paper and furnish proofs for some of them.

## D.1    Equivalence of NSGAN with Its Particle Model Interpretation

**Theorem D.1.** *The update of $g_\theta$ via applying the stop gradient operator to $\hat{Z}_i$ and descending the gradient*

$$\nabla_\theta \frac{1}{m} \sum_{i=1}^m \left\| g_\theta(\boldsymbol{z}_i) - \hat{\boldsymbol{Z}}_i \right\|_2^2$$

*in algorithm 1 is equivalent to descending the gradient*

$$-\nabla_\theta \frac{1}{m} \sum_{i=1}^m \log\left(d_\omega(g_\theta(\boldsymbol{z}_i))\right)$$

*in the original formulation of NSGAN.*

*Proof.* We prove by directly computing the gradient using the chain rule. In fact, we have

$$\begin{aligned}
\nabla_\theta \frac{1}{m} \sum_{i=1}^m \left\| g_\theta(\boldsymbol{z}_i) - \hat{\boldsymbol{Z}}_i \right\|_2^2 &= \frac{2}{m} \sum_{i=1}^m \nabla_\theta g_\theta(\boldsymbol{z}_i)^\top \cdot \left(g_\theta(\boldsymbol{z}_i) - \hat{\boldsymbol{Z}}_i\right) \\
&= -\frac{s}{m} \sum_{i=1}^m \nabla_\theta g_\theta(\boldsymbol{z}_i)^\top \cdot \frac{\nabla d_\omega(\boldsymbol{Z}_i)}{d_\omega(\boldsymbol{Z}_i)} \\
&= -s \nabla_\theta \frac{1}{m} \sum_{i=1}^m \log\left(d_\omega(g_\theta(\boldsymbol{z}_i))\right).
\end{aligned}$$

Note that in the first equation, we implicitly use the fact that $\nabla_\theta \hat{Z}_i = 0$ due to the assumption that the stop gradient operator is applied to $\hat{Z}_i$.    □

## D.2    Properties of Particle Update Dynamics — The General Result

**Theorem D.2.** *Assume that the discriminator is optimal, i.e., $d_\omega^*(\boldsymbol{x}) = p_{data}(\boldsymbol{x})/(p_{data}(\boldsymbol{x}) + p_g(\boldsymbol{x}))$. Denote $r(\boldsymbol{x}) = p_{data}(\boldsymbol{x})/p_g(\boldsymbol{x})$. At a point $\boldsymbol{x}$ where $r(\boldsymbol{x}) \approx 0$, $\boldsymbol{x}$ is updated following approximately $\nabla \log\left(r(\boldsymbol{x})\right)$. Conversely, when $r(\boldsymbol{x}) \gg 1$, $\boldsymbol{x}$ is updated following approximately $\nabla\left(-1/r(\boldsymbol{x})\right)$.*

*Proof.* In the following, we abbreviate $d_\omega^*(\boldsymbol{x})$ as $d(\boldsymbol{x})$. We then rewrite $\nabla d(\boldsymbol{x})/d(\boldsymbol{x})$ in terms of $r(\boldsymbol{x})$:

$$\begin{aligned}
\frac{\nabla d(\boldsymbol{x})}{d(\boldsymbol{x})} &= \frac{-p_{\text{data}}(\boldsymbol{x})\nabla p_g(\boldsymbol{x}) + p_g(\boldsymbol{x})\nabla p_{\text{data}}(\boldsymbol{x})}{(p_{\text{data}}(\boldsymbol{x}) + p_g(x))p_{\text{data}}(\boldsymbol{x})} \\
&= \nabla\left(\frac{p_{\text{data}}(\boldsymbol{x})}{p_g(\boldsymbol{x})}\right) \cdot \frac{p_g(\boldsymbol{x})^2}{p_{\text{data}}(\boldsymbol{x})(p_{\text{data}}(\boldsymbol{x}) + p_g(\boldsymbol{x}))} \\
&= \nabla r(\boldsymbol{x}) \cdot \frac{1}{r(\boldsymbol{x})(1 + r(\boldsymbol{x}))}.
\end{aligned}$$

When $r(\boldsymbol{x}) \approx 0$, we have

$$\frac{1}{r(\boldsymbol{x})(1 + r(\boldsymbol{x}))} \approx \frac{1}{r(\boldsymbol{x})}.$$

As a result,

$$\frac{\nabla d(\boldsymbol{x})}{d(\boldsymbol{x})} \approx \nabla \log r(\boldsymbol{x}).$$

When $r(\boldsymbol{x}) \gg 1$, we have

$$\frac{1}{r(\boldsymbol{x})(1 + r(\boldsymbol{x}))} \approx \frac{1}{r(\boldsymbol{x})^2}.$$

Consequently,

$$\frac{\nabla d(\boldsymbol{x})}{d(\boldsymbol{x})} \approx \nabla \left( -\frac{1}{r(\boldsymbol{x})} \right). \qquad \square$$

We hereby outline the implications of this theorem. The value of $\log\big(r(\boldsymbol{x})\big)$ changes dramatically as $\boldsymbol{x}$ decreases from 1 to 0, leading to correspondingly large magnitudes of $\|\nabla \log\big(r(\boldsymbol{x})\big)\|_2$ when $r(\boldsymbol{x}) \approx 0$. This indicates that in the regions where $p_g(\boldsymbol{x})$ significantly exceeds $p_{\text{data}}(\boldsymbol{x})$, particles are propelled towards distant points. Conversely, $\nabla\big( -1/r(\boldsymbol{x})\big)$ changes more gradually with increasing $\boldsymbol{x}$, resulting in smaller magnitudes of $\|\nabla\big( -1/r(\boldsymbol{x})\big)\|_2$ when $r(\boldsymbol{x}) \gg 1$. In such regions where $p_g(\boldsymbol{x})$ is lower than $p_{\text{data}}(\boldsymbol{x})$, particles tend to remain relatively stationary. These align with our observations in section 3.

### D.3    Properties of Particle Update Dynamics — The Data-Dependent Results

**Proposition D.1.** *Assume that*

$$p_{data} \sim \frac{1}{4}\mathcal{N}([1,1], 0.1\boldsymbol{I_2}) + \frac{1}{4}\mathcal{N}([1,-1], 0.1\boldsymbol{I_2}) + \frac{1}{4}\mathcal{N}([-1,1], 0.1\boldsymbol{I_2}) + \frac{1}{4}\mathcal{N}([-1,-1], 0.1\boldsymbol{I_2})$$

*and that $p_g \sim \mathcal{N}([0,0], 0.2\boldsymbol{I_2})$. Let $\boldsymbol{x} = [x_1, x_2]$. Then the vector field that governs particles' update is given by*

$$\nabla r(\boldsymbol{x}) \cdot \frac{1}{r(\boldsymbol{x})(1 + r(\boldsymbol{x}))},$$

*where*

$$r(\boldsymbol{x}) = \frac{1}{2} \sum_{(a,b)\in\{(\pm 1, \pm 1)\}} \exp\left(-2.5\big((x_1 - 2a)^2 + (x_2 - 2b)^2\big) + 5a^2 + 5b^2\right)$$

*and*

$$\nabla r(\boldsymbol{x}) = -\frac{5}{2} \sum_{(a,b)\in\{(\pm 1, \pm 1)\}} \exp\left(-2.5\big((x_1 - 2a)^2 + (x_2 - 2b)^2\big) + 5a^2 + 5b^2\right) \begin{bmatrix} x_1 - 2a \\ x_2 - 2b \end{bmatrix}.$$

*Proof.* For each Gaussian distribution, the density function is

$$\mathcal{N}(\boldsymbol{\mu}, \boldsymbol{\Sigma})(\boldsymbol{x}) = \frac{1}{2\pi\sqrt{\det(\boldsymbol{\Sigma})}} \exp\left( -\frac{1}{2}(\boldsymbol{x} - \boldsymbol{\mu})^\top \boldsymbol{\Sigma}^{-1}(\boldsymbol{x} - \boldsymbol{\mu}) \right).$$

Here, $\boldsymbol{\mu} \in \{[1,1], [1,-1], [-1,1], [-1,-1]\}$, and $\boldsymbol{\Sigma} = 0.1\boldsymbol{I_2}$. Therefore,

$$\begin{aligned}
\mathcal{N}([a,b], 0.1\boldsymbol{I_2})(\boldsymbol{x}) &= \frac{1}{2\pi \cdot 0.1} \cdot \exp\left( -\frac{1}{2 \cdot 0.1}\big((x_1 - a)^2 + (x_2 - b)^2\big) \right) \\
&= \frac{1}{0.2\pi} \cdot \exp\left(-5\big((x_1 - a)^2 + (x_2 - b)^2\big)\right).
\end{aligned}$$

Thus,

$$p_{\text{data}}(\boldsymbol{x}) = \frac{1}{0.8\pi} \sum_{(a,b)\in\{(\pm 1, \pm 1)\}} \exp\left(-5\big((x_1 - a)^2 + (x_2 - b)^2\big)\right).$$

For $p_g(\boldsymbol{x})$ which is normally distributed with mean $[0,0]$ and covariance $0.2\boldsymbol{I_2}$, we have

$$p_g(\boldsymbol{x}) = \frac{1}{0.4\pi} \cdot \exp\left(-2.5(x_1^2 + x_2^2)\right).$$

Combining the above results, we have

$$r(\boldsymbol{x}) = \frac{1}{2} \sum_{(a,b) \in \{(\pm 1, \pm 1)\}} \exp\left(-2.5\big((x_1 - 2a)^2 + (x_2 - 2b)^2\big) + 5a^2 + 5b^2\right).$$

Next, we compute $\nabla r(\boldsymbol{x})$:

$$\nabla r(\boldsymbol{x}) = \frac{1}{2} \sum_{(a,b) \in \{(\pm 1, \pm 1)\}} \nabla \exp\left(-2.5\big((x_1 - 2a)^2 + (x_2 - 2b)^2\big) + 5a^2 + 5b^2\right).$$

For each term $\exp\left(-2.5\big((x_1 - 2a)^2 + (x_2 - 2b)^2\big) + 5a^2 + 5b^2\right)$, its gradient is:

$$\nabla \exp\left(-2.5\big((x_1 - 2a)^2 + (x_2 - 2b)^2\big) + 5a^2 + 5b^2\right)$$
$$= -5 \exp\left(-2.5\big((x_1 - 2a)^2 + (x_2 - 2b)^2\big) + 5a^2 + 5b^2\right) \begin{bmatrix} x_1 - 2a \\ x_2 - 2b \end{bmatrix}.$$

Thus,

$$\nabla r(\boldsymbol{x}) = -\frac{5}{2} \sum_{(a,b) \in \{(\pm 1, \pm 1)\}} \exp\left(-2.5\big((x_1 - 2a)^2 + (x_2 - 2b)^2\big) + 5a^2 + 5b^2\right) \begin{bmatrix} x_1 - 2a \\ x_2 - 2b \end{bmatrix}.$$

Putting the expressions of $r(\boldsymbol{x})$ and $\nabla r(\boldsymbol{x})$ together, we will have

$$\nabla r(\boldsymbol{x}) \cdot \frac{1}{r(\boldsymbol{x})(1 + r(\boldsymbol{x}))}.$$

When we take a closer look at the numerator $\nabla r(\boldsymbol{x})$, we observe that it is a weighted sum of the vectors originating from $\boldsymbol{x}$ and pointing towards two times the centers of the four modes, which are $(2, 2)$, $(2, -2)$, $(-2, 2)$, and $(-2, -2)$. Due to the exponential decay property of the exponential function, the influence of these vectors diminishes rapidly with distance. Consequently, the vector field is predominantly influenced by the mode in the same quadrant as $\boldsymbol{x}$. Specifically, if we assume without loss of generality that $\boldsymbol{x}$ lies in the first quadrant, the vector field will be approximately $[2 - x_1, 2 - x_2]^\top$, up to a scaling factor. $\qquad\square$

**Proposition D.2.** *Assume that*

$$p_{data} \sim \frac{1}{4}\mathcal{N}([1, 1], 0.1\boldsymbol{I}_2) + \frac{1}{4}\mathcal{N}([1, -1], 0.1\boldsymbol{I}_2) + \frac{1}{4}\mathcal{N}([-1, 1], 0.1\boldsymbol{I}_2) + \frac{1}{4}\mathcal{N}([-1, -1], 0.1\boldsymbol{I}_2)$$

*and that $p_g \sim \mathcal{U}\big([-2, 2] \times [-2, 2]\big)$. Let $\boldsymbol{x} = [x_1, x_2]$. Then the vector field that governs particles' update is given by*

$$\nabla r(\boldsymbol{x}) \cdot \frac{1}{r(\boldsymbol{x})(1 + r(\boldsymbol{x}))},$$

*where*

$$r(\boldsymbol{x}) = \frac{20}{\pi} \sum_{(a,b) \in \{(\pm 1, \pm 1)\}} \exp\left(-5\big((x_1 - a)^2 + (x_2 - b)^2\big)\right) \cdot \mathbf{1}_{\boldsymbol{x} \in [-2,2] \times [-2,2]}$$

*and*

$$\nabla r(\boldsymbol{x}) = -\frac{200}{\pi} \sum_{(a,b) \in \{(\pm 1, \pm 1)\}} \exp\left(-5\big((x_1 - a)^2 + (x_2 - b)^2\big)\right) \begin{bmatrix} x_1 - a \\ x_2 - b \end{bmatrix} \cdot \mathbf{1}_{\boldsymbol{x} \in [-2,2] \times [-2,2]}.$$

*Proof.* For each Gaussian distribution, the density function is

$$\mathcal{N}(\boldsymbol{\mu}, \boldsymbol{\Sigma})(\boldsymbol{x}) = \frac{1}{2\pi\sqrt{\det(\boldsymbol{\Sigma})}} \exp\left(-\frac{1}{2}(\boldsymbol{x} - \boldsymbol{\mu})^\top \boldsymbol{\Sigma}^{-1}(\boldsymbol{x} - \boldsymbol{\mu})\right).$$

Here, $\boldsymbol{\mu} \in \{[1,1],[1,-1],[-1,1],[-1,-1]\}$, and $\boldsymbol{\Sigma} = 0.1\boldsymbol{I}_2$. Therefore,

$$\mathcal{N}([a,b], 0.1\boldsymbol{I}_2)(\boldsymbol{x}) = \frac{1}{2\pi \cdot 0.1} \cdot \exp\left(-\frac{1}{2 \cdot 0.1}\left((x_1-a)^2 + (x_2-b)^2\right)\right)$$
$$= \frac{1}{0.2\pi} \cdot \exp\left(-5\left((x_1-a)^2 + (x_2-b)^2\right)\right).$$

Thus,

$$p_{\text{data}}(\boldsymbol{x}) = \frac{1}{0.8\pi} \sum_{(a,b)\in\{(\pm1,\pm1)\}} \exp\left(-5\left((x_1-a)^2 + (x_2-b)^2\right)\right)$$

For $p_g(\boldsymbol{x})$ which is uniformly distributed, we have

$$p_g(\boldsymbol{x}) = \frac{1}{16} \cdot \mathbf{1}_{\boldsymbol{x}\in[-2,2]\times[-2,2]}.$$

Combining the above results,

$$r(\boldsymbol{x}) = \frac{20}{\pi} \sum_{(a,b)\in\{(\pm1,\pm1)\}} \exp\left(-5\left((x_1-a)^2 + (x_2-b)^2\right)\right) \cdot \mathbf{1}_{\boldsymbol{x}\in[-2,2]\times[-2,2]}.$$

Now, we compute $\nabla r(\boldsymbol{x})$:

$$\nabla r(\boldsymbol{x}) = \frac{20}{\pi} \sum_{(a,b)\in\{(\pm1,\pm1)\}} \nabla \exp\left(-5\left((x_1-a)^2 + (x_2-b)^2\right)\right) \mathbf{1}_{\boldsymbol{x}\in[-2,2]\times[-2,2]}.$$

For each term $\exp\left(-5\left((x_1-a)^2 + (x_2-b)^2\right)\right)$, its gradient is:

$$\nabla \exp\left(-5\left((x_1-a)^2 + (x_2-b)^2\right)\right) = -10\exp\left(-5\left((x_1-a)^2 + (x_2-b)^2\right)\right)\begin{bmatrix} x_1-a \\ x_2-b \end{bmatrix}.$$

Thus,

$$\nabla r(\boldsymbol{x}) = -\frac{200}{\pi} \sum_{(a,b)\in\{(\pm1,\pm1)\}} \exp\left(-5\left((x_1-a)^2 + (x_2-b)^2\right)\right)\begin{bmatrix} x_1-a \\ x_2-b \end{bmatrix} \cdot \mathbf{1}_{\boldsymbol{x}\in[-2,2]\times[-2,2]}.$$

Putting the expressions of $r(\boldsymbol{x})$ and $\nabla r(\boldsymbol{x})$ together, we will have

$$\nabla r(\boldsymbol{x}) \cdot \frac{1}{r(\boldsymbol{x})(1+r(\boldsymbol{x}))}.$$

When we take a closer look at the numerator $\nabla r(\boldsymbol{x})$, we observe that it is a weighted sum of the vectors originating from $\boldsymbol{x}$ and pointing towards the centers of the four modes, which are $(1,1)$, $(1,-1)$, $(-1,1)$, and $(-1,-1)$. Due to the exponential decay property of the exponential function, the influence of these vectors diminishes rapidly with distance. Consequently, the vector field is predominantly influenced by the mode in the same quadrant as $\boldsymbol{x}$. Specifically, if we assume without loss of generality that $\boldsymbol{x}$ lies in the first quadrant, the vector field will be approximately $[1-x_1, 1-x_2]^\top$, up to a scaling factor. $\qquad\square$

**Proposition D.3.** *Assume that*

$$p_{data} \sim \frac{1}{4}\mathcal{N}([1,1], 0.1\boldsymbol{I}_2) + \frac{1}{4}\mathcal{N}([1,-1], 0.1\boldsymbol{I}_2) + \frac{1}{4}\mathcal{N}([-1,1], 0.1\boldsymbol{I}_2) + \frac{1}{4}\mathcal{N}([-1,-1], 0.1\boldsymbol{I}_2)$$

*and that $p_g \sim \mathcal{N}([1,1], \boldsymbol{I}_2)$. Let $\boldsymbol{x} = [x_1, x_2]$. Then the vector field that governs particles' update is given by*

$$\nabla r(\boldsymbol{x}) \cdot \frac{1}{r(\boldsymbol{x})(1+r(\boldsymbol{x}))},$$

*where*

$$r(\boldsymbol{x}) = \frac{5}{2} \sum_{(a,b)\in\{(\pm 1,\pm 1)\}} \exp\left(-\frac{9}{2}\left(\left(x_1 - \frac{10a-1}{9}\right)^2 + \left(x_2 - \frac{10b-1}{9}\right)^2\right) + \frac{5}{9}(a-1)^2 + \frac{5}{9}(b-1)^2\right)$$

*and*

$$\nabla r(\boldsymbol{x}) = -\frac{45}{2} \cdot$$

$$\sum_{(a,b)\in\{(\pm 1,\pm 1)\}} \exp\left(-\frac{9}{2}\left(\left(x_1 - \frac{10a-1}{9}\right)^2 + \left(x_2 - \frac{10b-1}{9}\right)^2\right) + \frac{5}{9}(a-1)^2 + \frac{5}{9}(b-1)^2\right)\begin{bmatrix} x_1 - \frac{10a-1}{9} \\ x_2 - \frac{10b-1}{9} \end{bmatrix}.$$

*Proof.* For each Gaussian distribution, the density function is

$$\mathcal{N}(\boldsymbol{\mu},\boldsymbol{\Sigma})(\boldsymbol{x}) = \frac{1}{2\pi\sqrt{\det(\boldsymbol{\Sigma})}} \exp\left(-\frac{1}{2}(\boldsymbol{x}-\boldsymbol{\mu})^\top\boldsymbol{\Sigma}^{-1}(\boldsymbol{x}-\boldsymbol{\mu})\right).$$

Here, $\boldsymbol{\mu} \in \{[1,1],[1,-1],[-1,1],[-1,-1]\}$, and $\boldsymbol{\Sigma} = 0.1\boldsymbol{I}_2$. Therefore,

$$\mathcal{N}([a,b],0.1\boldsymbol{I}_2)(\boldsymbol{x}) = \frac{1}{2\pi\cdot 0.1} \cdot \exp\left(-\frac{1}{2\cdot 0.1}\left((x_1-a)^2 + (x_2-b)^2\right)\right)$$

$$= \frac{1}{0.2\pi} \cdot \exp\left(-5\left((x_1-a)^2 + (x_2-b)^2\right)\right).$$

Thus,

$$p_{\text{data}}(\boldsymbol{x}) = \frac{1}{0.8\pi} \sum_{(a,b)\in\{(\pm 1,\pm 1)\}} \exp\left(-5\left((x_1-a)^2 + (x_2-b)^2\right)\right).$$

For $p_g(\boldsymbol{x})$ which is normally distributed with mean $[1,1]$ and covariance $\boldsymbol{I}_2$, we have

$$p_g(\boldsymbol{x}) = \frac{1}{2\pi} \cdot \exp\left(-0.5\left((x_1-1)^2 + (x_2-1)^2\right)\right).$$

Combining the above results, we have

$$r(\boldsymbol{x}) = \frac{5}{2} \sum_{(a,b)\in\{(\pm 1,\pm 1)\}} \exp\left(-\frac{9}{2}\left(\left(x_1 - \frac{10a-1}{9}\right)^2 + \left(x_2 - \frac{10b-1}{9}\right)^2\right) + \frac{5}{9}(a-1)^2 + \frac{5}{9}(b-1)^2\right).$$

Next, we compute $\nabla r(\boldsymbol{x})$:

$$\nabla r(\boldsymbol{x}) = \frac{5}{2} \sum_{(a,b)\in\{(\pm 1,\pm 1)\}} \nabla \exp\left(-\frac{9}{2}\left(\left(x_1 - \frac{10a-1}{9}\right)^2 + \left(x_2 - \frac{10b-1}{9}\right)^2\right) + \frac{5}{9}(a-1)^2 + \frac{5}{9}(b-1)^2\right).$$

For each term on the right-hand side, its gradient is:

$$\nabla \exp\left(-\frac{9}{2}\left(\left(x_1 - \frac{10a-1}{9}\right)^2 + \left(x_2 - \frac{10b-1}{9}\right)^2\right) + \frac{5}{9}(a-1)^2 + \frac{5}{9}(b-1)^2\right)$$

$$= -9 \cdot \exp\left(-\frac{9}{2}\left(\left(x_1 - \frac{10a-1}{9}\right)^2 + \left(x_2 - \frac{10b-1}{9}\right)^2\right) + \frac{5}{9}(a-1)^2 + \frac{5}{9}(b-1)^2\right)\begin{bmatrix} x_1 - (10a-1)/9 \\ x_2 - (10b-1)/9 \end{bmatrix}.$$

Thus,

$$\nabla r(\boldsymbol{x}) = -\frac{45}{2} \cdot$$

$$\sum_{(a,b)\in\{(\pm 1,\pm 1)\}} \exp\left(-\frac{9}{2}\left(\left(x_1 - \frac{10a-1}{9}\right)^2 + \left(x_2 - \frac{10b-1}{9}\right)^2\right) + \frac{5}{9}(a-1)^2 + \frac{5}{9}(b-1)^2\right)\begin{bmatrix} x_1 - \frac{10a-1}{9} \\ x_2 - \frac{10b-1}{9} \end{bmatrix}.$$

Putting the expressions of $r(\boldsymbol{x})$ and $\nabla r(\boldsymbol{x})$ together, we will have

$$\nabla r(\boldsymbol{x}) \cdot \frac{1}{r(\boldsymbol{x})(1 + r(\boldsymbol{x}))}.$$

When we take a closer look at the numerator $\nabla r(\boldsymbol{x})$, we observe that it is a weighted sum of the vectors originating from $\boldsymbol{x}$ and pointing towards $(1,1)$, $(-11/9, 1)$, $(1, -11/9)$, and $(-11/9, -11/9)$, respectively. Due to the exponential decay property of the exponential function, the influence of these vectors diminishes rapidly with distance. Consequently, the vector field is predominantly influenced by the mode in the same quadrant as $\boldsymbol{x}$. Specifically, if we assume without loss of generality that $\boldsymbol{x}$ lies in the first quadrant, the vector field will be approximately $[1 - x_1, 1 - x_2]^\top$, up to a scaling factor. □

**Proposition D.4.** *Assume that*

$$p_{data} \sim \frac{1}{4}\mathcal{N}([3,3], 0.1\boldsymbol{I}_2) + \frac{1}{4}\mathcal{N}([3,-3], 0.1\boldsymbol{I}_2) + \frac{1}{4}\mathcal{N}([-3,3], 0.1\boldsymbol{I}_2) + \frac{1}{4}\mathcal{N}([-3,-3], 0.1\boldsymbol{I}_2)$$

*and that $p_g \sim \mathcal{N}([3,3], 3\boldsymbol{I}_2)$. Let $\boldsymbol{x} = [x_1, x_2]$. Then the vector field that governs particles' update is given by*

$$\nabla r(\boldsymbol{x}) \cdot \frac{1}{r(\boldsymbol{x})(1 + r(\boldsymbol{x}))},$$

*where*

$$r(\boldsymbol{x}) = \frac{15}{4} \sum_{(a,b) \in \{(\pm 3, \pm 3)\}} \exp\left(-\frac{29}{6}\left(\left(x_1 - \frac{30a-3}{29}\right)^2 + \left(x_2 - \frac{30b-3}{29}\right)^2\right) + \frac{5}{29}(a-3)^2 + \frac{5}{29}(b-3)^2\right)$$

*and*

$$\nabla r(\boldsymbol{x}) = -\frac{145}{4} \cdot$$
$$\sum_{(a,b) \in \{(\pm 3, \pm 3)\}} \exp\left(-\frac{29}{6}\left(\left(x_1 - \frac{30a-3}{29}\right)^2 + \left(x_2 - \frac{30b-3}{29}\right)^2\right) + \frac{5}{29}(a-3)^2 + \frac{5}{29}(b-3)^2\right) \begin{bmatrix} x_1 - \frac{30a-3}{29} \\ x_2 - \frac{30b-3}{29} \end{bmatrix}.$$

*Proof.* For each Gaussian distribution, the density function is

$$\mathcal{N}(\boldsymbol{\mu}, \boldsymbol{\Sigma})(\boldsymbol{x}) = \frac{1}{2\pi\sqrt{\det(\boldsymbol{\Sigma})}} \exp\left(-\frac{1}{2}(\boldsymbol{x} - \boldsymbol{\mu})^\top \boldsymbol{\Sigma}^{-1}(\boldsymbol{x} - \boldsymbol{\mu})\right).$$

Here, $\boldsymbol{\mu} \in \{[3,3], [3,-3], [-3,3], [-3,-3]\}$, and $\boldsymbol{\Sigma} = 0.1\boldsymbol{I}_2$. Therefore,

$$\mathcal{N}([a,b], 0.1\boldsymbol{I}_2)(\boldsymbol{x}) = \frac{1}{2\pi \cdot 0.1} \cdot \exp\left(-\frac{1}{2 \cdot 0.1}\left((x_1 - a)^2 + (x_2 - b)^2\right)\right)$$
$$= \frac{1}{0.2\pi} \cdot \exp\left(-5\left((x_1 - a)^2 + (x_2 - b)^2\right)\right).$$

Thus,

$$p_{\text{data}}(\boldsymbol{x}) = \frac{1}{0.8\pi} \sum_{(a,b) \in \{(\pm 3, \pm 3)\}} \exp\left(-5\left((x_1 - a)^2 + (x_2 - b)^2\right)\right).$$

For $p_g(\boldsymbol{x})$ which is normally distributed with mean $[3,3]$ and covariance $3\boldsymbol{I}_2$, we have

$$p_g(\boldsymbol{x}) = \frac{1}{6\pi} \cdot \exp\left(-\frac{1}{6}\left((x_1 - 3)^2 + (x_2 - 3)^2\right)\right).$$

Combining the above results, we have

$$r(\boldsymbol{x}) = \frac{15}{4} \sum_{(a,b) \in \{(\pm 3, \pm 3)\}} \exp\left(-\frac{29}{6}\left(\left(x_1 - \frac{30a-3}{29}\right)^2 + \left(x_2 - \frac{30b-3}{29}\right)^2\right) + \frac{5}{29}(a-3)^2 + \frac{5}{29}(b-3)^2\right).$$

Next, we compute $\nabla r(\boldsymbol{x})$:

$$\frac{15}{4} \sum_{(a,b)\in\{(\pm 3,\pm 3)\}} \nabla \exp\left(-\frac{29}{6}\left(\left(x_1 - \frac{30a-3}{29}\right)^2 + \left(x_2 - \frac{30b-3}{29}\right)^2\right) + \frac{5}{29}(a-3)^2 + \frac{5}{29}(b-3)^2\right).$$

For each term on the right-hand side, its gradient is:

$$\exp\left(-\frac{29}{6}\left(\left(x_1 - \frac{30a-3}{29}\right)^2 + \left(x_2 - \frac{30b-3}{29}\right)^2\right) + \frac{5}{29}(a-3)^2 + \frac{5}{29}(b-3)^2\right) =$$

$$-\frac{29}{3}\exp\left(-\frac{29}{6}\left(\left(x_1 - \frac{30a-3}{29}\right)^2 + \left(x_2 - \frac{30b-3}{29}\right)^2\right) + \frac{5}{29}(a-3)^2 + \frac{5}{29}(b-3)^2\right)\begin{bmatrix} x_1 - (30a-3)/29 \\ x_2 - (30b-3)/29 \end{bmatrix}.$$

Thus,

$$\nabla r(\boldsymbol{x}) = -\frac{145}{4} \cdot$$

$$\sum_{(a,b)\in\{(\pm 3,\pm 3)\}} \exp\left(-\frac{29}{6}\left(\left(x_1 - \frac{30a-3}{29}\right)^2 + \left(x_2 - \frac{30b-3}{29}\right)^2\right) + \frac{5}{29}(a-3)^2 + \frac{5}{29}(b-3)^2\right)\begin{bmatrix} x_1 - \dfrac{30a-3}{29} \\ x_2 - \dfrac{30b-3}{29} \end{bmatrix}.$$

Putting the expressions of $r(\boldsymbol{x})$ and $\nabla r(\boldsymbol{x})$ together, we will have

$$\nabla r(\boldsymbol{x}) \cdot \frac{1}{r(\boldsymbol{x})(1 + r(\boldsymbol{x}))}.$$

When we take a closer look at the numerator $\nabla r(\boldsymbol{x})$, we observe that it is a weighted sum of the vectors originating from $\boldsymbol{x}$ and pointing towards $(27/29, 27/29)$, $(-33/29, 27/29)$, $(27/29, -33/29)$, and $(-33/29, -33/29)$, respectively. Due to the exponential decay property of the exponential function, the influence of these vectors diminishes rapidly with distance. Consequently, the vector field is predominantly influenced by the mode in the same quadrant as $\boldsymbol{x}$. Specifically, if we assume without loss of generality that $\boldsymbol{x}$ lies in the first quadrant, the vector field will be approximately $[27/29 - x_1, 27/29 - x_2]^\top$, up to a scaling factor. Regarding the term $1 + r(\boldsymbol{x})$ in the denominator, we observe that its magnitude is large when $\boldsymbol{x}$ is far from the coordinates $x_1 = 0$, $x_2 = 0$, and the centers of the modes. This increased magnitude compared to the scenario in proposition D.3 explains the overall weakening of the attraction intensity near all the modes. □

### D.4 Characterization of Measuring-Preserving Maps

**Lemma D.1.** *(Durrett, 2019) Let $\boldsymbol{X}$ be a random variable taking values on $\mathbb{R}$ and let $F_{\boldsymbol{X}}(x)$ be its CDF. Then*

$$F_{\boldsymbol{X}}^{-1}\big(\mathcal{U}(0,1)\big) \sim \boldsymbol{X}$$

*and*

$$F_X(\boldsymbol{X}) \sim \mathcal{U}(0,1),$$

*where $\mathcal{U}(0,1)$ denotes the uniform distribution on $(0,1)$.*

**Theorem D.3.** *Let $\Phi(x)$ denotes the cumulative distribution function (CDF) of $\mathcal{N}(0,1)$ and let $\Psi(x)$ be that of $p_{data}(x)$. If $g$ satisfies $g_{\#}p_z = p_{data}$, then $g = \Psi^{-1} \circ h \circ \Phi$, where $h$ is a measure-preserving map of $\mathcal{U}(0,1)$, i.e., the uniform distribution on $(0,1)$.*

*Proof.* We only need to show that $\Psi \circ g \circ \Phi^{-1}$ is a measure-preserving map of $\mathcal{U}(0,1)$. In fact, by lemma D.1, we have

$$(\Psi \circ g \circ \Phi^{-1})_{\#}\mathcal{U}(0,1) = (\Psi \circ g)_{\#}p_z = \Psi_{\#}p_{\text{data}} = \mathcal{U}(0,1).$$ □

### D.5 Steepness of Measure-Preserving Map in $1$-Dimension

**Theorem D.4.** *Assume that*

$$p_{data} \sim \frac{1}{N}\mathcal{N}(\mu_1, \sigma^2) + \frac{1}{N}\mathcal{N}(\mu_2, \sigma^2) + \cdots + \frac{1}{N}\mathcal{N}(\mu_N, \sigma^2),$$

*Here the $\mu_i$'s are in ascending order, and $\mu_{i+1} - \mu_i \geq 6\sigma$ for all $1 \leq i \leq N - 1$. Let $\Phi(x)$ denotes the cumulative distribution function (CDF) of $\mathcal{N}(0, 1)$ and let $\Psi(x)$ be that of $p_{data}(x)$. Then $g(x) := \Psi^{-1}(\Phi(x))$ satisfies*

$$\mathcal{S}(g) \geq \min_{1 \leq i \leq N-1} \sigma \cdot \exp\left(\frac{(\mu_{i+1} - \mu_i)^2}{8\sigma^2}\right) \cdot \exp(-q^2),$$

*where $q$ is the $(1 - 1/N)$-th quantile of the standard Gaussian distribution.*

*Proof.* Instead of computing the derivative of $g$, we compute that of $g^{-1}$. By the formula for the derivative of inverse functions, we have

$$(g^{-1})'(y) = \frac{\Psi'(y)}{\Phi'\big(\Phi^{-1}(\Psi(y))\big)}$$

$$= \frac{1}{N\sigma} \sum_{i=1}^{N} \exp\left(-\frac{(y - \mu_i)^2}{2\sigma^2}\right) \cdot \exp\left(\frac{(\Phi^{-1}(\Psi(y))^2}{2}\right)$$

$$\leq \max_{1 \leq i \leq N-1} \frac{1}{N\sigma} \cdot N \cdot \exp\left(-\frac{(\mu_{i+1} - \mu_i)^2}{8\sigma^2}\right) \cdot \exp\left(\frac{(\Phi^{-1}(\Psi((\mu_i + \mu_{i+1})/2)))^2}{2}\right)$$

$$\leq \max_{1 \leq i \leq N-1} \frac{1}{\sigma} \cdot \exp\left(-\frac{(\mu_{i+1} - \mu_i)^2}{8\sigma^2}\right) \cdot \exp(q^2).$$

where $q$ is the $(1 - 1/N)$-th quantile of the standard Gaussian distribution. Again, by the formula for the derivative of inverse functions, we have

$$\mathcal{S}(g) \geq \min_{1 \leq i \leq N-1} \sigma \cdot \exp\left(\frac{(\mu_i - \mu_{i+1})^2}{8\sigma^2}\right) \cdot \exp(-q^2). \qquad \square$$

### D.6 Steepness of Measure-Preserving Maps in Higher Dimensions

The standard result in (Durrett, 2019) specifically addresses the case of lemma D.2 where $K = 1$. And it can be straightforwardly extended to encompass any $K$.

**Lemma D.2.** *(Durrett, 2019) Let $\boldsymbol{X} \sim \rho(\boldsymbol{x})\mathrm{d}\boldsymbol{x}$ be a $n$-dimensional random vector. Let $\mathcal{D} \subset \mathbb{R}^n$ satisfy $\mathbb{P}(\boldsymbol{X} \in \mathcal{D}) = 1$. Assume that the map*

$$\varphi\colon \mathcal{D} = \biguplus_{k=1}^{K} \mathcal{D}_i \to \mathbb{R}^n$$

*satisfies the following requirements: for each $1 \leq k \leq K$, $\varphi := \varphi|_{\mathcal{D}_k}$ is injective and its inverse function is continuously differentiable. Then the probability density function of $\boldsymbol{Y} = \varphi(\boldsymbol{X})$ is*

$$\rho_{\boldsymbol{Y}}(\boldsymbol{y}) = \sum_{k=1}^{K} \rho_{\boldsymbol{X}}\left(\varphi^{-1}(\boldsymbol{y})\right) \cdot \left|\det\left(J_{\varphi_k^{-1}(\boldsymbol{y})}\right)\right| \cdot \mathbf{1}_{\varphi(\mathcal{D}_k)}(\boldsymbol{y}).$$

*Equivalently, for any $\boldsymbol{x} \in \mathcal{D}$,*

$$\rho_{\boldsymbol{Y}}(\varphi(\boldsymbol{x})) = \sum_{k=1}^{K} \rho_{\boldsymbol{X}}(\boldsymbol{x}) \cdot |\det(J_\varphi(\boldsymbol{x}))|^{-1} \cdot \mathbf{1}_{\varphi(\mathcal{D}_k)}(\varphi(\boldsymbol{x})).$$

**Theorem D.5.** *Let $\nu$ be the truncated Gaussian distribution $\mathcal{N}_r(\mathbf{0}, \mathbf{I}_n)$ in the $n$-dimensional ball $\mathcal{B}_r(\mathbf{0})$ and assume that $\hat{\mu}$ is a probability measure with probability density function*

$$\hat{f}(\boldsymbol{x}) = \frac{1}{Nh} \sum_{i=1}^{N} \left(\sqrt{1/\pi}\right)^n \exp\left(-\frac{\|\boldsymbol{x} - \boldsymbol{x}_i\|_2^2}{h^2}\right).$$

*Suppose that $g \colon \mathcal{B}_r(\mathbf{0}) \to \mathbb{R}^n$ is continuously differentiable and piecewise injective. Then $\mathcal{S}(g) > M$, where*

$$M = \delta \cdot h^{1/n} \cdot \sqrt{\frac{\pi}{n}} \cdot \max_{1 \le i \le N} \exp\left(\frac{\|\bar{\boldsymbol{x}} - \boldsymbol{x}_i\|_2^2}{nh^2}\right).$$

*Here, $\bar{\boldsymbol{x}} = \sum_{i=1}^{N} \boldsymbol{x}_i / n$, and $\delta = \exp\left(-r^2/2\right)/\sqrt{2\pi}$.*

*Proof.* Let $\mathcal{D}_k$ $(1 \le k \le K)$ be a partition of $\mathcal{B}_r(\mathbf{0})$ such that for each $1 \le k \le K$, $g|_{\mathcal{D}_k}$ is injective. We regard $g$ as the composition of two functions $g := g_2 \circ g_1$. Here, $g_1 \colon \mathcal{B}_r(\mathbf{0}) \to (0, 1)^n$ satisfies

$$g_1(\boldsymbol{x}) = g_1(x_1, x_2, \ldots, x_n) = (\Phi_r(x_1), \Phi_r(x_2), \ldots, \Phi_r(x_n)),$$

where $\Phi_r(\cdot)$ is the cumulative density function of the 1-dimensional standard Gaussian distribution truncated in $(-r, r)$. It is straightforward to show that the derivative of $\Phi_r$ has a positive lower bound, say,

$$\delta := \frac{1}{\sqrt{2\pi}} \exp\left(-\frac{r^2}{2}\right).$$

Thus $|\det J_{g_1}(\boldsymbol{x})| \ge \delta^n$ for any $\boldsymbol{x} \in \mathcal{B}_r(\mathbf{0})$.

By lemma D.1, $g_{1\#}\nu = \pi$, where $\pi$ is the uniform distribution on $(0, 1)^n$. In the rest of the proof, we direct our focus to $g_2 \colon (0, 1)^n \to \mathbb{R}^n$, which satisfies $g_{2\#}\pi = \hat{\mu}$. Because $g_2 = g \circ g_1^{-1}$ and $g$ is injective on $\mathcal{D}_i$ $(1 \le i \le N)$, we conclude that $g_2$ is injective on $g_1(\mathcal{D}_k)$ $(1 \le k \le K)$. By applying lemma D.2 to $g_2$ and $g_1(\mathcal{D}_k)$ $(1 \le k \le K)$, we deduce that for $\boldsymbol{y} \in (0, 1)^n$,

$$\hat{f}(g_2(\boldsymbol{y})) = \sum_{k=1}^{K} \frac{1}{|\det(J_{g_2}(\boldsymbol{y}))|} \cdot \mathbb{1}_{g_2(g_1(\mathcal{D}_k))}(g_2(\boldsymbol{y})) \ge \sum_{k=1}^{K} \frac{1}{|\det(J_{g_2}(\boldsymbol{y}))|} \cdot \mathbb{1}_{g_1(\mathcal{D}_k)}(\boldsymbol{y}) = \frac{1}{|\det(J_{g_2}(\boldsymbol{y}))|}.$$

Let $\mathcal{B}_R(\mathbf{0})$ be the $n$-dimensional open ball centered at the origin with radius $R = 2\max_{1 \le i \le N} \|\boldsymbol{x}_i\|_2$. We consider the point $\boldsymbol{y}_0$ satisfying

$$g_2(\boldsymbol{y}_0) = \arg\max_{\boldsymbol{x} \in \mathcal{B}_R(\mathbf{0})} \min_{1 \le i \le N} \|\boldsymbol{x} - \boldsymbol{x}_i\|_2.$$

If there are many of them, we randomly pick one. Let $\bar{\boldsymbol{x}} = \sum_{i=1}^{N} \boldsymbol{x}_i / n$. For this $\boldsymbol{y}_0$, we have

$$\hat{f}(g_2(\boldsymbol{y}_0)) \le \hat{f}(\bar{\boldsymbol{x}}) = \frac{1}{Nh} \sum_{i=1}^{N} \left(\sqrt{1/\pi}\right)^n \exp\left(-\frac{\|\bar{\boldsymbol{x}} - \boldsymbol{x}_i\|_2^2}{h^2}\right).$$

Hence

$$|\det(J_{g_2}(\boldsymbol{y}_0))| \ge \hat{f}(g_2(\boldsymbol{y}_0))^{-1} \ge h \cdot \left(\sqrt{\pi}\right)^n \min_{1 \le i \le N} \exp\left(\frac{\|\bar{\boldsymbol{x}} - \boldsymbol{x}_i\|_2^2}{h^2}\right).$$

Recall that we have $|\det(J_{g_1}(g_2(\boldsymbol{y}_0)))| \ge \delta^n$, where $\delta = \frac{1}{\sqrt{2\pi}} \exp\left(-\frac{r^2}{2}\right).$

Combine the above results and we have

$$|\det(J_g(\boldsymbol{y}_0))| = |\det(J_{g_2}(\boldsymbol{y}_0)) \det(J_{g_1}(g_2(\boldsymbol{y}_0)))| \ge h \cdot \left(\delta\sqrt{\pi}\right)^n \min_{1 \le i \le N} \exp\left(\frac{\|\bar{\boldsymbol{x}} - \boldsymbol{x}_i\|_2^2}{h^2}\right).$$

If $\mathcal{S}(g) < M$, then by Hadamard's theorem which states that the determinant of a matrix is smaller than or equal to the product of the 2-norms of its column vectors, one can show that

$$|\det(J_g(\boldsymbol{y}_0))| \le \left(M\sqrt{n}\right)^n.$$

Substitute for the expression of $M$, we will find that the above two expressions contradict. $\qquad\square$

### D.7 Evolution of Steepness

**Theorem D.6.** *Assume that $p_{data} \sim \mathcal{N}(\mathbf{0}, k_*^2 \boldsymbol{I}_n)$ and that the discriminator is optimal, i.e., the discriminator consistently provides the precise moving direction for the particle. Then $k_t$, the steepness of $g$ at discrete time step $t$ satisfies*

$$k_{t+1} = k_t + s\left(\frac{1}{k_t^2} - \frac{1}{k_*^2}\right) \cdot \frac{1}{1 + \frac{k_t \varphi(k_t \boldsymbol{x}_0/k_*)}{k_* \varphi(\boldsymbol{x}_0)}},$$

*where $0 \le t \le T$, and $T$ is the maximum time. Here, $\varphi$ is the probability density function of $\mathcal{N}(\mathbf{0}, \boldsymbol{I}_n)$.*

*Proof.* Let $\varphi(\boldsymbol{x})$ be the probability density function of the $n$-dimensional standard Gaussian distribution

$$\varphi(\boldsymbol{x}) = \frac{1}{\sqrt{(2\pi)^n}} \cdot \exp\left(-\frac{1}{2}\boldsymbol{x}^\top \boldsymbol{x}\right).$$

Then the probability density function of $\mathcal{N}(\mathbf{0}, k^2 \boldsymbol{I}_n)$ is $\varphi(\boldsymbol{x}/k)/k$. Let $\boldsymbol{x}_t = k_t \boldsymbol{x}_0$ denotes the position of the particle at time $t$. Here, $k_t$ represents the steepness of the generator function. We investigate the evolution of the particle subject to the vector field given by $\nabla d(\boldsymbol{x})/d(\boldsymbol{x})$. Assuming the discriminator is optimal, this process is governed by the following explicit formula (Yi et al., 2023):

$$\boldsymbol{x}_{t+1} = \boldsymbol{x}_t + s \cdot \frac{\nabla r(\boldsymbol{x}_t)}{r(\boldsymbol{x}_t)(r(\boldsymbol{x}_t) + 1)}, \quad t = 1, 2, \ldots, T.$$

Here, $s$ denotes the stepsize (which absorbs the constant 2 in the vector field), $T$ is the maximum time, and

$$r(\boldsymbol{x}) = \frac{\varphi(\boldsymbol{x}/k_*)/k_*}{\varphi(\boldsymbol{x}/k_t)/k_t}$$

is the ratio of the probability density function of $p_{\text{data}}$ and $p_g$. By the formula of $\varphi(\boldsymbol{x})$, we deduce that $\nabla \varphi(\boldsymbol{x}) = -\varphi(\boldsymbol{x})\boldsymbol{x}$. Below we compute $\nabla r(\boldsymbol{x})$ by the chain rule:

$$\nabla r(\boldsymbol{x}) = \frac{k_t}{k_*} \cdot \frac{\nabla \varphi(\boldsymbol{x}/k_*) \cdot \varphi(\boldsymbol{x}/k_t) - \varphi(\boldsymbol{x}/k_*)\nabla \varphi(\boldsymbol{x}/k_t)}{\varphi(\boldsymbol{x}/k_t)^2}$$

$$= \frac{k_t}{k_*} \cdot \left(\frac{1}{k_t^2} - \frac{1}{k_*^2}\right) \frac{\varphi(\boldsymbol{x}/k_*)}{\varphi(\boldsymbol{x}/k_t)} \cdot \boldsymbol{x}.$$

Using $\boldsymbol{x}_t = k_t \boldsymbol{x}_0$, we derive the following recurrent formula for $\{k_t\}_{t=0}^T$:

$$k_{t+1} = k_t + s\left(\frac{1}{k_t^2} - \frac{1}{k_*^2}\right) \cdot \frac{1}{1 + \frac{k_t \varphi(k_t \boldsymbol{x}_0/k_*)}{k_* \varphi(\boldsymbol{x}_0)}}. \qquad\square$$

### D.8 Quantitative Results on How Steepness Impacts the Severity of Mode Mixture

**Theorem D.7.** *Assume that $p_{data} \sim \sum_{i=1}^N \mathcal{N}(\mu_i, \sigma^2)/N$. Here, the $\mu_i$'s are in ascending order, with the condition that $\mu_i - \mu_{i-1} \ge 6\sigma$ for all $1 \le i \le N-1$. Furthermore, assume that the generator function $g$ is increasing and satisfies $\mathcal{S}(g) \le k$. Additionally, assume that*

$$g^{-1}\left(\frac{\mu_i + \mu_{i+1}}{2}\right) = \Phi^{-1}\left(\Psi\left(\frac{\mu_i + \mu_{i+1}}{2}\right)\right),$$

where $\Phi(x)$ denotes the cumulative distribution function (CDF) of the standard normal distribution $\mathcal{N}(0, 1)$, and $\Psi(x)$ is the CDF of the distribution $p_{data}(x)$. Then, the probability that the particles fall into the interval

$$\bigcup_{i=1}^{N}[\mu_i + 3\sigma, \mu_{i+1} - 3\sigma],$$

which indicates mode mixture, is at least

$$\sum_{i=1}^{N}\left(\Phi\left(\Phi^{-1}\left(\Psi\left(\frac{\mu_i + \mu_{i+1}}{2}\right)\right) + \frac{\mu_{i+1} - \mu_i - 3\sigma}{2k}\right) - \Phi\left(\Phi^{-1}\left(\Psi\left(\frac{\mu_i + \mu_{i+1}}{2}\right)\right) - \frac{\mu_{i+1} - \mu_i - 3\sigma}{2k}\right)\right).$$

*Proof.* Given $x \sim \mathcal{N}(0, 1)$, we need to calculate the probability that

$$x \in \bigcup_{i=1}^{N}[g^{-1}(\mu_i + 3\sigma), g^{-1}(\mu_{i+1} - 3\sigma)].$$

Since $g^{-1}\big((\mu_i + \mu_{i+1})/2\big)$ is determined by its optimal counterpart, it suffices to analyze how $g^{-1}(\mu_i + 3\sigma)$ and $g^{-1}(\mu_{i+1} - 3\sigma)$ deviate from this value. In other words, we only need to compute the maximum value of $g^{-1}(\mu_i + 3\sigma)$ and the minimum value of $g^{-1}(\mu_{i+1} - 3\sigma)$, as the probability that a standard Gaussian variable falls within an interval decreases with respect to its left endpoint and increases with respect to its right endpoint. Using the property that $\mathcal{S}(g) \leq k$, we have:

$$g^{-1}(\mu_i + 3\sigma) \leq g^{-1}\left(\frac{\mu_i + \mu_{i+1}}{2}\right) - \frac{\mu_{i+1} - \mu_i - 3\sigma}{2k},$$

and

$$g^{-1}(\mu_{i+1} - 3\sigma) \geq g^{-1}\left(\frac{\mu_i + \mu_{i+1}}{2}\right) + \frac{\mu_{i+1} - \mu_i - 3\sigma}{2k}.$$

By summing over all intervals, we derive that the probability that particles fall into

$$\bigcup_{i=1}^{N}[\mu_i + 3\sigma, \mu_{i+1} - 3\sigma]$$

is at least

$$\sum_{i=1}^{N}\left(\Phi\left(g^{-1}\left(\frac{\mu_i + \mu_{i+1}}{2}\right) + \frac{\mu_{i+1} - \mu_i - 3\sigma}{2k}\right) - \Phi\left(g^{-1}\left(\frac{\mu_i + \mu_{i+1}}{2}\right) - \frac{\mu_{i+1} - \mu_i - 3\sigma}{2k}\right)\right)$$

$$= \sum_{i=1}^{N}\left(\Phi\left(\Phi^{-1}\left(\Psi\left(\frac{\mu_i + \mu_{i+1}}{2}\right)\right) + \frac{\mu_{i+1} - \mu_i - 3\sigma}{2k}\right) - \Phi\left(\Phi^{-1}\left(\Psi\left(\frac{\mu_i + \mu_{i+1}}{2}\right)\right) - \frac{\mu_{i+1} - \mu_i - 3\sigma}{2k}\right)\right).$$

Note that for the case that $N = 2$ and $-\mu_1 = \mu_2 = \mu$, this probability simplifies to

$$\Phi\left(\frac{2\mu - 3\sigma}{2k}\right) - \Phi\left(-\frac{2\mu - 3\sigma}{2k}\right). \qquad \square$$

# E  Disparity Among Modes Across Different Datasets

## E.1  MNIST

**Proporcessing.** We first transform the images in MNIST by sequentially resizing the images to $64 \times 64$ pixels, converting them to PyTorch tensors, and normalizing the tensor values to the range of $[-1, 1]$.

**Computation.** We calculate the average image tensor for each label based on a set of 10 image tensors sharing the same label. Next, we compute the pairwise distances between these average tensors using the Frobenius norm. The resulting distances are visualized as a heatmap in fig. 9.

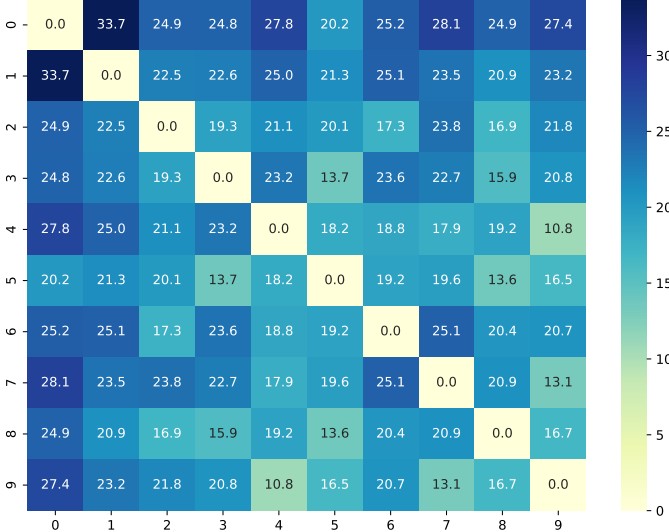

Figure 9: Frobenius distances between different modes in MNIST. The tensor of the modes are approximated by taking the average of image tensors that share the same label.

## E.2 Fashion MNIST

**Proporcessing.** We first transform the images in Fashion MNIST by first resizing the images to $64 \times 64$ pixels, converting them to PyTorch tensors, and normalizing the tensor values to the range of $[-1, 1]$.

**Computation.** We calculate the average image tensor for each label based on a set of 10 image tensors sharing the same label. Next, we compute the pairwise distances between these average tensors using the Frobenius norm. The resulting distances are visualized as a heatmap in fig. 10.

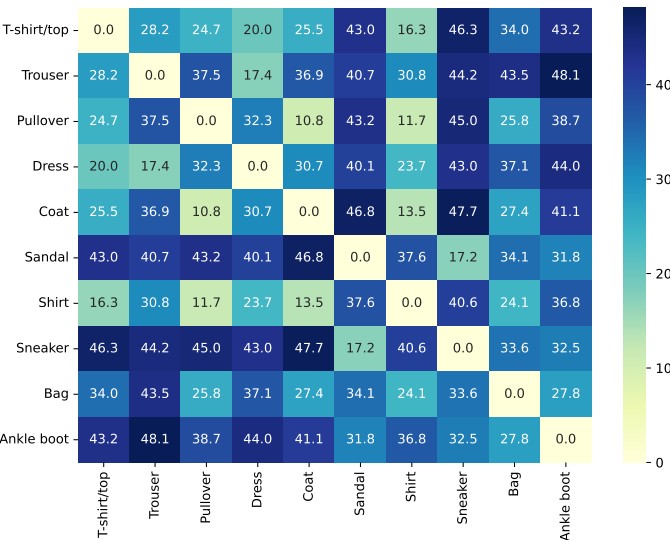

Figure 10: Frobenius distances between different modes in Fashion MNIST. The tensor of the modes are approximated by taking the average of image tensors that share the same label.

### E.3 CIFAR-10

**Proporcessing.** We first transform the images in CIFAR-10 by sequentially resizing the images to $64 \times 64$ pixels, converting them to PyTorch tensors, and normalizing the tensor values to the range of $[-1, 1]$.

**Computation.** We calculate the average image tensor for each label based on a set of 10 image tensors sharing the same label. Next, we compute the pairwise distances between these average tensors using the Frobenius norm. The resulting distances are visualized as a heatmap in fig. 11.

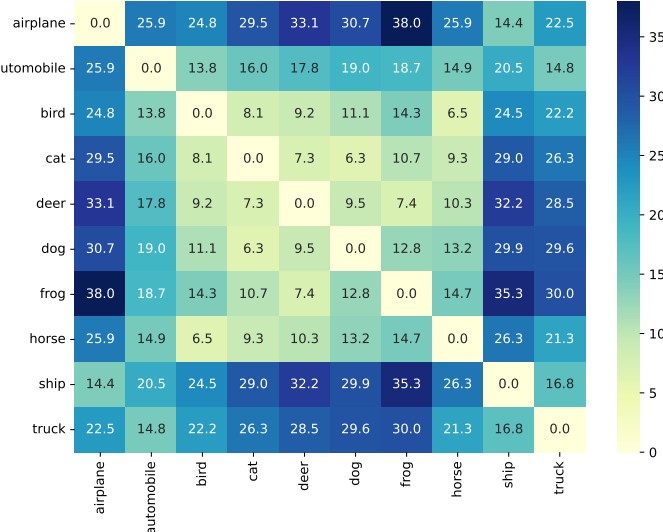

Figure 11: Frobenius distances between different modes in CIAFR-10. The tensor of the modes are approximated by taking the average of image tensors that share the same label.

## F   Detailed Experimental Settings

Our codes are all provided in the supplementary material.

### F.1   Verifying Fitting and Refining

**Methodology.** We demonstrate that the phases of fitting and refining exist in real-world datasets. To do this, we use a classification network $q(\boldsymbol{x})$ that takes an image tensor $\boldsymbol{x}$ as an input and outputs a 10-dimensional vector,

$$(p_0, p_1, \ldots, p_9),$$

where each $p_i \in [0, 1]$ denotes the likelihood of $\boldsymbol{x}$ corresponding to the $i$th category (e.g., the 1st category in MNIST corresponds to the handwritten digit 1 and the 2nd category in Fashion MNIST represents pullovers). Our focus gravitates towards those $p_i$'s that exhibit significant magnitudes. For discernibility, a threshold $\tau$ is set to $10^{-2}$. In other words, if $p_i > 10^{-2}$, then there is a notable probability that $\boldsymbol{x}$ belongs to the $i$th category. Empirical observations suggest that seldom do more than three $p_i$'s surpass the designated threshold. Hence, for any $\boldsymbol{x}$, we may pair $(i, j)$ when both $p_i$ and $p_j$ exceed $\tau$. By pairing, the intuition is that such $\boldsymbol{x}$ potentially resides *between* modes $i$ and $j$. In scenarios where only a single $p_i$ surpasses $\tau$, $i$ is paired with itself, implying that the $\boldsymbol{x}$ predominantly belongs to the $i$th category. We count the occurrences of the pairings $(i, j)$ $(0 \leq i, j \leq 9)$ in a batch of size 256 and visualize them with heatmaps in fig. 5, fig. 15 and fig. 16. In these figures, the value of the entry $(i, j)$ represents the logarithmically transformed occurrence frequency of pair $(i, j)$ within a batch, adjusted by one, thereby mitigating the impact of dominant diagonal values on the colorbar.

**Classification networks.** We use the MNIST classification network in MNIST classification network and the Fashion MNIST classification network in Fashion MNIST classification network.

**Number of training runs.** We conducted our experiments at least 50 times and consistently observed similar patterns across all trials. Therefore, we randomly selected two of these experiments to present in this paper.

## F.2 Early Stopping

**Early stopping on $3$-dimensional Gaussian mixture.** In this part, our codes borrow heavily from NSGAN. Both the generator and the discriminator are implemented as full-connected neural networks with SGD optimizers. Now we elaborate on how to calculate the threshold defined in algorithm 2. The threshold is given by $k_s \cdot \sqrt{2}/(2h)$. We set $k_s = 2$, the distance between two modes in the 3-dimensional Gaussian mixture dataset. For $h$, it equals $\sqrt{2}\sigma$, where $\sigma$ is the standard variation of every Gaussian component, which is $\sqrt{0.0125}$ in our setting. Therefore the threshold is

$$2 \times \sqrt{2}/(2 \times \sqrt{2} \times \sqrt{0.0125}) \approx 8.9.$$

As for the warm-up training iteration parameter $N_w$, we set it to 50.

**Early stopping on MNIST.** In this part, our generator and discriminator architectures borrow heavily from NSGAN on MNIST. Both the generator and the discriminator are implemented as convolutional neural networks with Adam optimizers. Now we elaborate on how to calculate the threshold defined in algorithm 2. The threshold is given by $k_s \cdot \sqrt{2}/(2h)$. We set $k_s = 33.7$, the distance between two farthest modes in MNIST (please refer to appendix E). For $h$, it equals $\sqrt{2}\sigma$, where $\sigma^2$ is derived as follows. We first compute the population variance of the images from each label, arriving at 10 values. Then we compute their average value, and divide this value by $64 \times 64 \times 1$, i.e., the total number of dimensions. Therefore the threshold is

$$33.7 \times \sqrt{2}/(2 \times \sqrt{2} \times \sqrt{0.33/64^2}) \approx 1877.$$

As for the warm-up training iteration parameter $N_w$, we set it to 20.

**Early stopping on fashion MNIST.** In this part, our generator and discriminator architectures borrow heavily from NSGAN on Fashion MNIST. Both the generator and the discriminator are implemented as convolutional neural networks with Adam optimizers. Now we elaborate on how to calculate the threshold defined in algorithm 2. The threshold is given by $k_s \cdot \sqrt{2}/(2h)$. We set $k_s = 48.1$, the distance between two farthest modes in Fashion MNIST (please refer to appendix E). For $h$, it equals $\sqrt{2}\sigma$, where $\sigma^2$ is derived as follows. We first compute the population variance of the images from each label, arriving at 10 values. Then we compute their average value, and divide this value by $64 \times 64 \times 1$, i.e., the total number of dimensions. Therefore the threshold is

$$48.1 \times \sqrt{2}/(2 \times \sqrt{2} \times \sqrt{0.33/64^2}) \approx 2679.$$

As for the warm-up training iteration parameter $N_w$, we set it to 50.

**Early stopping on CIFAR-10.** In this part, our generator and discriminator architectures borrow heavily from NSGAN on CIFAR-10. Both the generator and the discriminator are implemented as convolutional neural networks with Adam optimizers. Now we elaborate on how to calculate the threshold defined in algorithm 2. The threshold is given by $k_s \cdot \sqrt{2}/(2h)$. We set $k_s = 38.0$, the distance between two farthest modes in CIFAR-10 (please refer to appendix E). For $h$, it equals $\sqrt{2}\sigma$, where $\sigma^2$ is derived as follows. We first compute the population variance of the images from each label, arriving at 10 values. Then we compute their average value, and divide this value by $64 \times 64 \times 3$, i.e., the total number of dimensions. Therefore the threshold is

$$38.0 \times \sqrt{2}/(2 \times \sqrt{2} \times \sqrt{0.23/(64^2 \times 3)}) \approx 4391.$$

As for the warm-up training iteration parameter $N_w$, we set it to 50.

**Number of training runs.** On all of the datasets mentioned above, we conducted our experiments at least 100 times. We observed similar patterns across all trials, although the point at which the GANs collapsed varied. Therefore, we choose to present those that collapsed before a certain threshold to ensure consistency

in our reported results. It is important to note that the generated samples eventually collapsed in our experiments, either sooner or later, without contradicting the findings in our paper.

## G  Additional Experimental Results

### G.1  UMAP Embedding of MNIST

**Introduction to UMAP.** We first give a brief introduction to UMAP (Uniform Manifold Approximation and Projection) (McInnes et al., 2018). It is a widely-used dimensionality reduction technique that excels in preserving the global structure and local relationships of high-dimensional data when mapped to lower dimensions. UMAP operates by first constructing a high-dimensional graph representation of the data, capturing both local and global structure. This is achieved by identifying nearest neighbors for each data point and creating a weighted graph where the edge weights reflect the probability of connections between points. UMAP then optimizes a low-dimensional embedding by minimizing the cross-entropy between the high-dimensional and low-dimensional representations.

**Methodology.** We use UMAP to embed the MNIST dataset into a 3-dimensional space and transform the generated images into the same embedded space using the `transform` method. In the resulting plots, the dots representing different digits are colored differently, while the generated samples are shown as black dots.

**Results: The detailed training process.** We first present the detailed results of the training process to complement that in fig. 1. At the beginning, the generated samples tend to cluster within a small region of the entire space. As training advances, these initial clusters gradually disperse, with the generated samples spreading out more widely. Over time, the samples increasingly occupy the entire space spanned by the real data modes, effectively capturing the diversity present in the real dataset. This justifies the fitting phase. However, around the 47th epoch, this positive trend reverses. The generated samples begin to collapse towards only a fraction of the modes, which indicates the collapsing phase. It is important to note that the refining phase cannot be directly observed in these figures. While the generated samples appear to lie within the embedded modes, their quality may still vary due to potential information loss when reducing from high-dimensional space to low-dimensional space. This dimensionality reduction can obscure some details of the samples. For more details on how to test the refining phase, please refer to section 6.1.

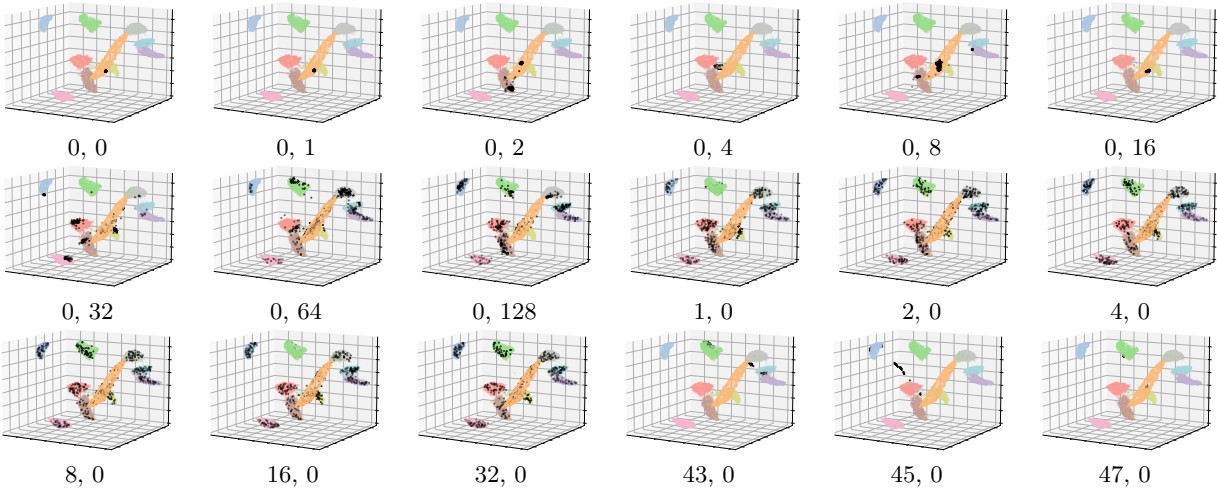

Figure 12: The UMAP visualization of the detailed training process of NSGAN on MNIST. The label "$e, b$" at the bottom of each plot denotes the $b$th batch within the $e$th epoch. Initially, the generated samples cluster within a small region. As training progresses, these clusters begin to disperse, with the samples spreading out gradually. Over time, the generated samples increasingly populate the entire space that is spanned by the real modes. However, at approximately the 47th epoch, this trend reverses. The generated samples begin to collapse to only a fraction of the modes.

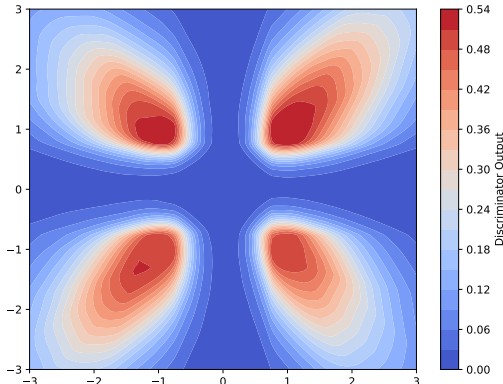

Figure 13: The UMAP visualization of generated samples at initialization using different initialization methods. **From left to right: Xavier normal, Xavier uniform, Kaiming normal, Kaiming uniform, orthogonal, and Dirac initialization.** The latent dimension (i.e., noise dimension) is the same as the image dimension. Despite these different approaches, the generated samples consistently cluster within a small region of the entire space rather than spanning it completely.

**Results: Different initialization method.** We then examine the distribution of generated samples under various network initialization methods to emphasize the *necessity* of the fitting phase. Initially, these generated samples tend to cluster within a small region of the entire space, rather than spanning it completely. This phenomenon can be explained by two primary factors. First, when the latent dimension (i.e., the noise dimension) is smaller than the image dimension, the generator maps the noise distribution onto a low-dimensional manifold, which inherently restricts its ability to span the entire space. Second, even when the latent dimension is equal to or greater than the image dimension, this issue remains. We empirically validate this by testing several popular initialization methods, including Xavier normal, Xavier uniform, Kaiming normal, Kaiming uniform, orthogonal, and Dirac initializations. Despite these different approaches, the generated samples consistently fail to fully span the space, highlighting the critical need for the fitting phase to achieve a more comprehensive distribution.

### G.2 Overfitting of the Discriminator

In this subsection, we elaborate on the optimal discriminator's behavior outlined in section 5.1. We consider the following synthetic dataset

$$p_{\text{data}} \sim \frac{1}{4}\mathcal{N}([1,1], 0.0125\boldsymbol{I}_2) + \frac{1}{4}\mathcal{N}([1,-1], 0.0125\boldsymbol{I}_2) + \frac{1}{4}\mathcal{N}([-1,1], 0.0125\boldsymbol{I}_2) + \frac{1}{4}\mathcal{N}([-1,-1], 0.0125\boldsymbol{I}_2),$$

and train the discriminator until optimal. We plot the values of the optimal discriminator in fig. 14. We observe that the discriminator values are close to 0.5 in the central regions of the modes and vanish in the regions far from the modes. Between them, the discriminator values smoothly change from 0.5 to 0.

Figure 14: The values of the optimal discriminator. The discriminator values are close to 0.5 in the central regions of the modes (i.e., $[\pm 1, \pm 1]$) and vanish in the regions far from the modes. Between them, the discriminator values smoothly change from 0.5 to 0.

## G.3 Verifying Fitting and Refining

**Annotated heatmaps for MNIST.** We verify the existence of fitting and refining on MNIST. Annotated heatmaps are employed to track the evolution of pairings $(i, j)$ occurrence within batches of size 256. The values depicted in these heatmaps represent the logarithm of occurrence counts plus 1, with darker colors indicating higher values. Each heatmap includes epoch numbers ranging from 0 to 38 displayed at the bottom. Initially, the heatmap has few nonzero entries, indicating limited sample diversity during the fitting phase. As training advances, more entries became nonzero, reflecting a broader distribution of generated samples across the mode space. Notably, the values of off-diagonal entries signifies the severity of mode mixture, which gradually decrease over the course of training, validating the refining phase. However, the issue of mode mixture persists even at the end of refining. By the 36th epoch, the heatmap only has two nonzero entries, suggesting the collapsing phase, where the generated samples become less diverse and concentrate around few modes. These observations provide empirical evidence for our proposed three phases of GAN training.

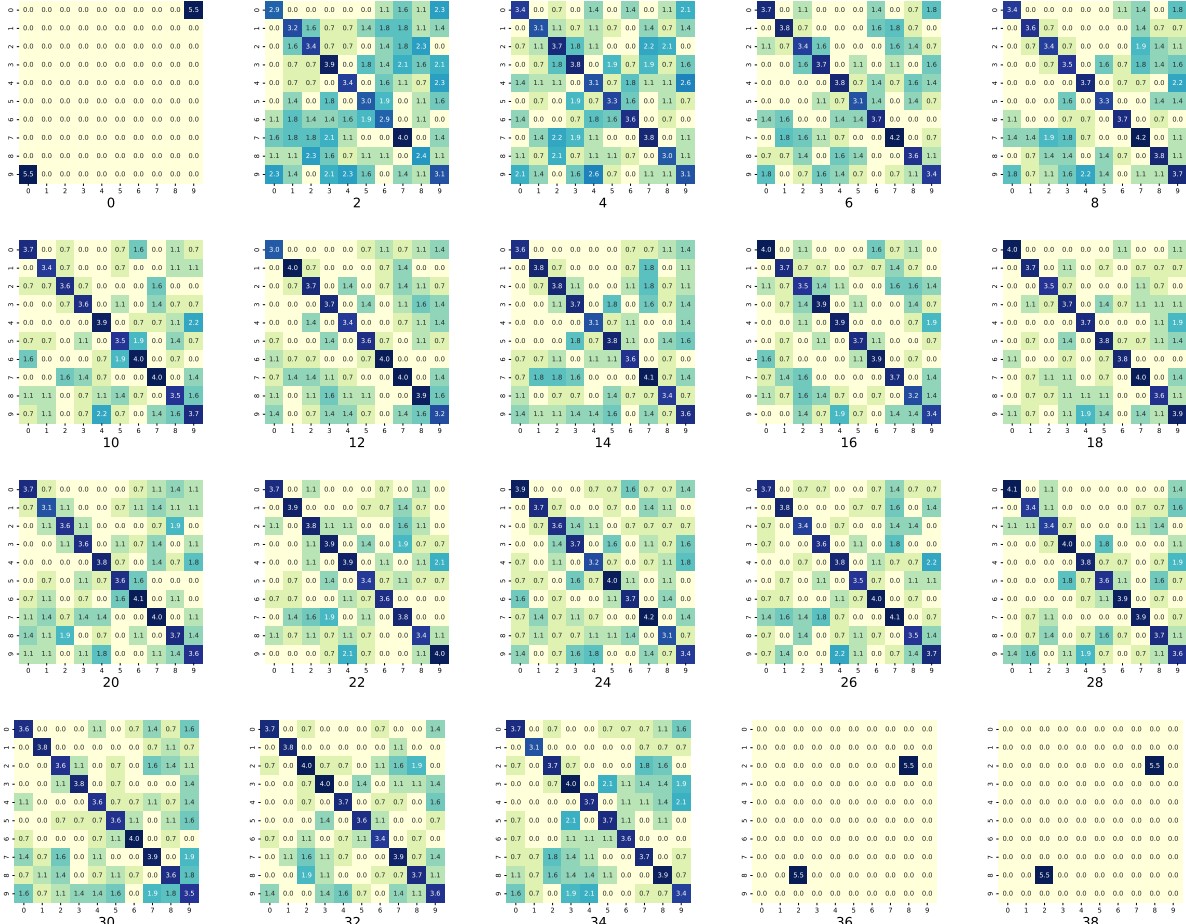

Figure 15: Annotated heatmaps for verifying fitting and refining in MNIST. The values are the logarithm of the occurrence of pairings $(i, j)$ plus 1 in a batch of size 256. Darker colors indicate higher values. The epochs, ranging from 0 to 38, are displayed at the bottom of each heatmap. Initially, there are few nonzero entries, suggesting limited sample diversity. As training progresses, more entries become nonzero, indicating wider sample distribution across mode space, which corresponds to the fitting phase. Off-diagonal entries reflect mode mixture, which diminishes over training, confirming the refining phase. Remarkably, mode mixture persists even at the closure of the refining phase. Note that by the 36th epoch, only two entries remain nonzero, indicating the collapsing phase.

**Verifying fitting and refining in Fashion MNIST.** We verify the existence of fitting and refining on Fashion MNIST using annotated heatmaps. The heatmap values are the logarithm of pairings $(i, j)$ occurrence plus 1 in batches of size 256, with darker colors indicating higher values. Each epoch is divided into 5 collections of batches, denoted as $e$, $b$ where $e$ is the epoch and $b$ is the batch collection within the epoch. Initially, there are only two nonzero entries, which suggests limited sample diversity. As training progresses, more entries become nonzero, indicating a broader sample distribution across the mode space during the fitting phase. Notably, unlike MNIST, the phases of fitting and refining in Fashion MNIST occur quickly, evidenced by the rapid stabilization of off-diagonal values. It is important to note that the large values in some off-diagonal entries do not necessarily imply severe mode mixture. For example, "T-shirt", "Pullover", and "Shirt" are frequently confused in Fashion MNIST classification tasks.

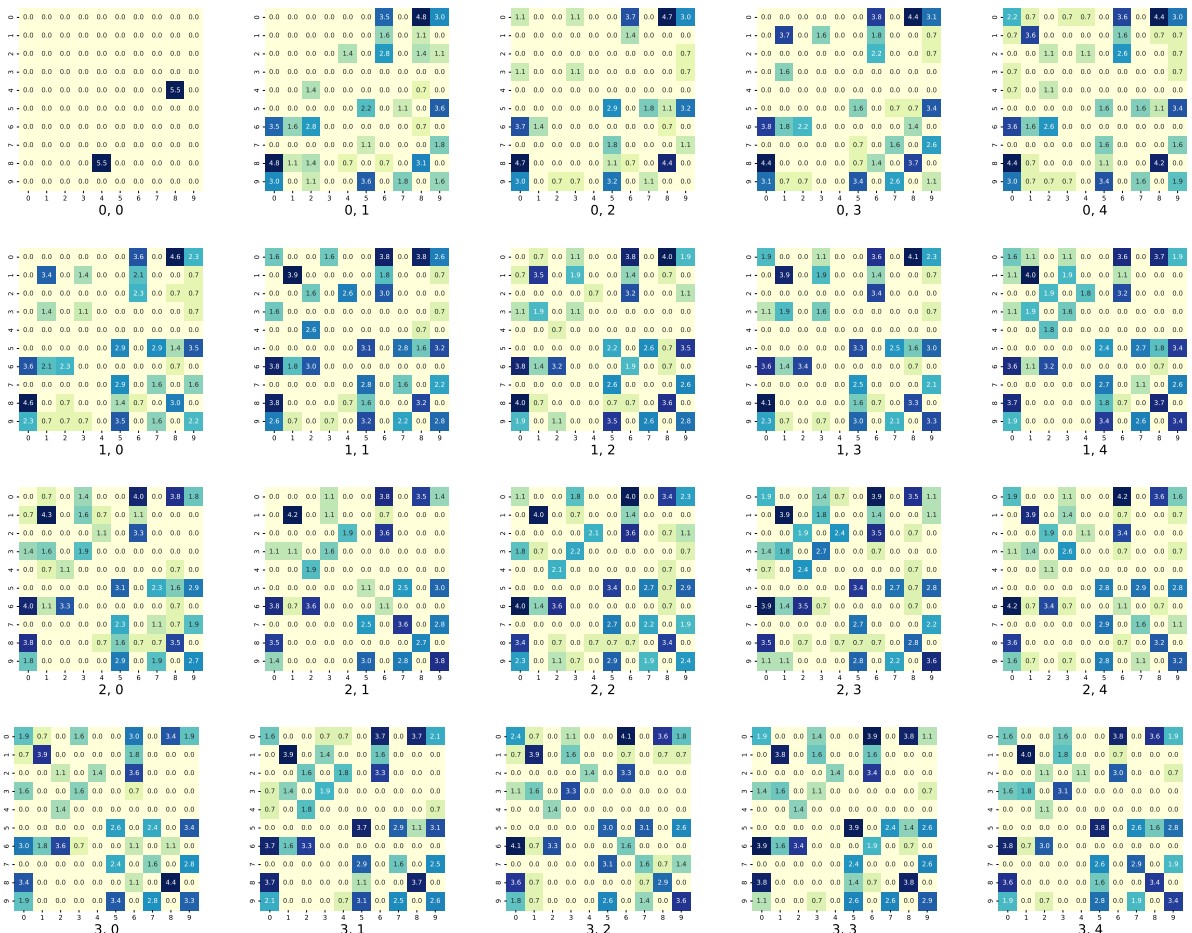

Figure 16: Annotated heatmaps for verifying fitting and refining in Fashion MNIST. The labels 0 to 9 mean "T-shirt/top", "Trouser", "Pullover", "Dress", "Coat", "Sandal", "Shirt", "Sneaker", "Bag", and "Ankle boot", respectively. The values are the logarithm of the occurrence of pairings $(i, j)$ plus 1 in a batch of size 256. Darker colors indicate higher values. Each epoch is equally divided into 5 collection of batches. The label "$e, b$" at the bottom of each heatmap denotes the $b$th collection within the $e$th epoch. Therefore, the heatmaps displayed are for the first 4 epochs only. Initially, the few nonzero entries indicate limited sample diversity. As training progresses, more entries became nonzero, reflecting a broader sample distribution across the mode space, which corresponds to the fitting phase. Unlike MNIST, the phases of fitting and refining in Fashion MNIST take place rapidly because the off-diagonal values stabilize quickly. It is important to note that the large values of some off-diagonal entries do not necessarily imply severe mode mixture; for instance, "T-shirt", "Pullover", and "Shirt" are often confused in Fashion MNIST classification tasks.

### G.4 Early Stopping on Gaussian Mixture: More Training Runs

We train NSGAN on the 3-dimensional Gaussian mixture dataset and record down $||\nabla d_\omega/d_\omega||_2$ each epoch until the maximum specified epochs. The threshold is set to 8.9 for reasons in appendix F.2. Instead of halting training when this norm exceeds the threshold, we opt to continue training, which allows us to assess the sample quality both before and after the stop.

In fig. 17, we present the tendency of $||\nabla d_\omega/d_\omega||_2$ against epochs in four trials. We consistently observe a pattern where these values initially increase, then decrease to nearly zero, subsequently rise above the threshold of 8.9, and continue to increase thereafter. This phenomenon can be empirically understood as follows: during the fitting phase, particles are updated to relatively large distances to explore modes; in the refining phase, most particles stabilize near the modes, while the remainder converge to the modes. Finally, in the collapsing phase discussed in section 5.1, particles are pushed away from the modes. The generated samples in the four trials are displayed in fig. 18, fig. 19, fig. 20, and fig. 21, respectively.

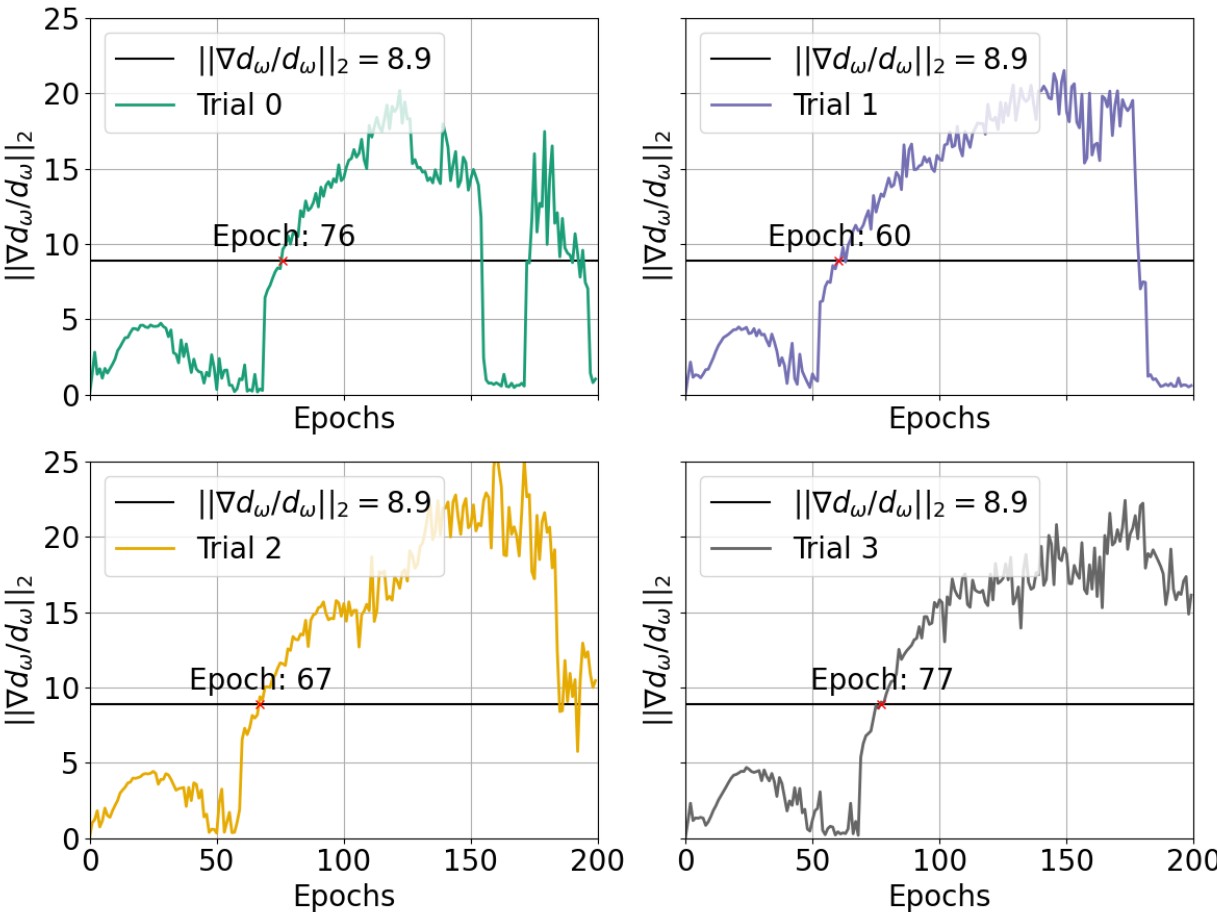

Figure 17: The values of $||\nabla d_\omega/d_\omega||_2$ against epochs in four trials. We mark the stopping epochs by red crosses. The patterns of $||\nabla d_\omega/d_\omega||_2$ values across epochs are consistent in all the four training runs. Initially, these values increase, then decrease to nearly zero, and subsequently rise above the threshold of 8.9, continuing to increase thereafter.

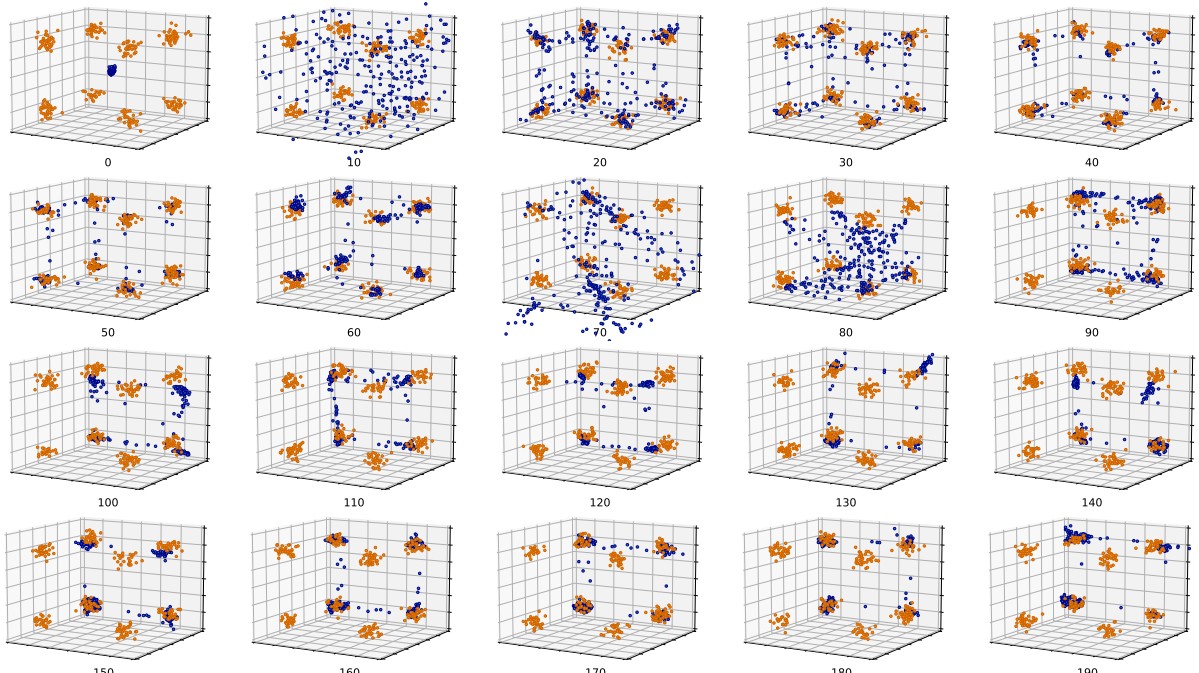

Figure 18: Generated samples in trial 0. The stopping point is the 76th epoch. Before the stopping point, generated samples demonstrate notable quality, particularly evident around the 50th epoch. As training approaches the stopping point, they start to deviate from the modes. After the stopping, they undergo a gradual deterioration, ultimately collapsing to approximately half of the 8 modes.

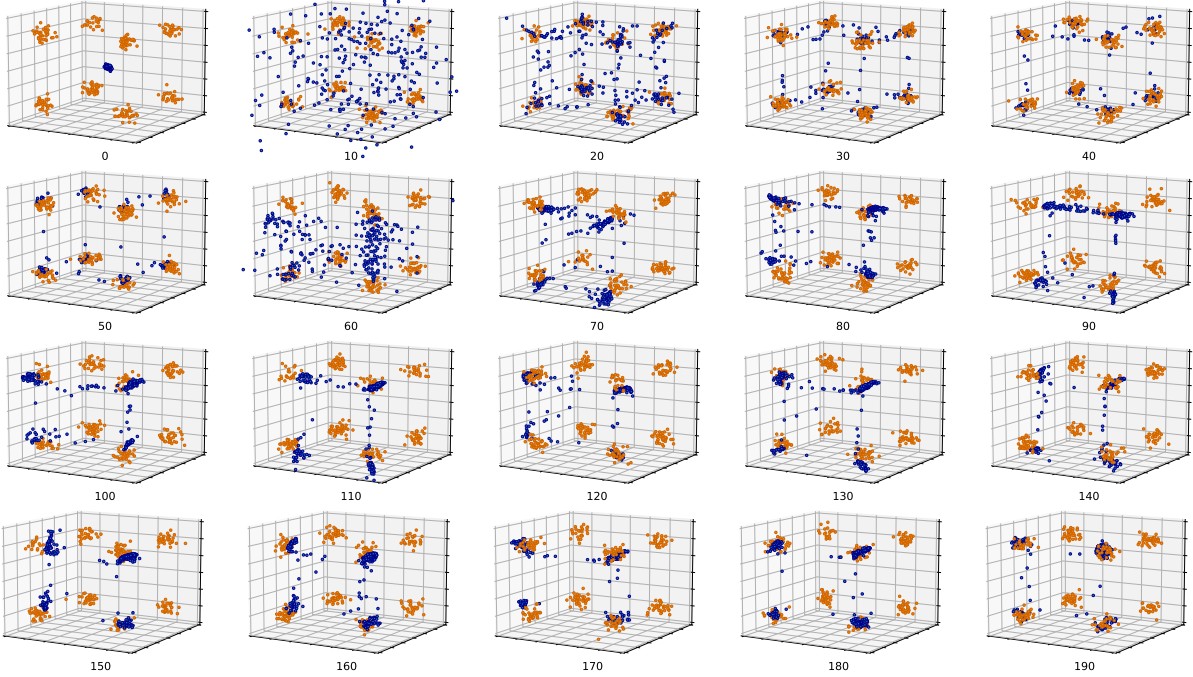

Figure 19: Generated samples in trial 1. The stopping point is the 60th epoch. Before the stopping point, generated samples demonstrate notable quality, particularly evident around the 50th epoch. As training approaches the stopping point, they start to deviate from the modes. After the stopping, they undergo a gradual deterioration, ultimately collapsing to approximately half of the 8 modes.

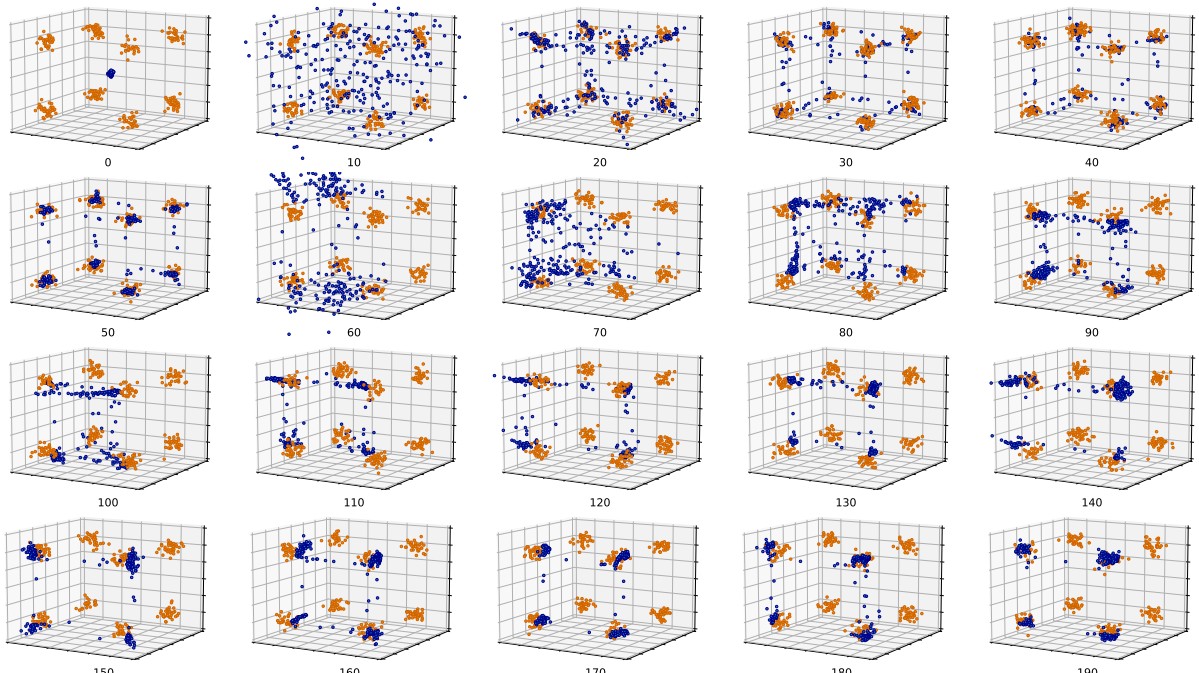

Figure 20: Generated samples in trial 2. The stopping point is the 67th epoch. Before the stopping point, generated samples demonstrate notable quality, particularly evident around the 50th epoch. As training approaches the stopping point, they start to deviate from the modes. After the stopping, they undergo a gradual deterioration, ultimately collapsing to approximately half of the 8 modes.

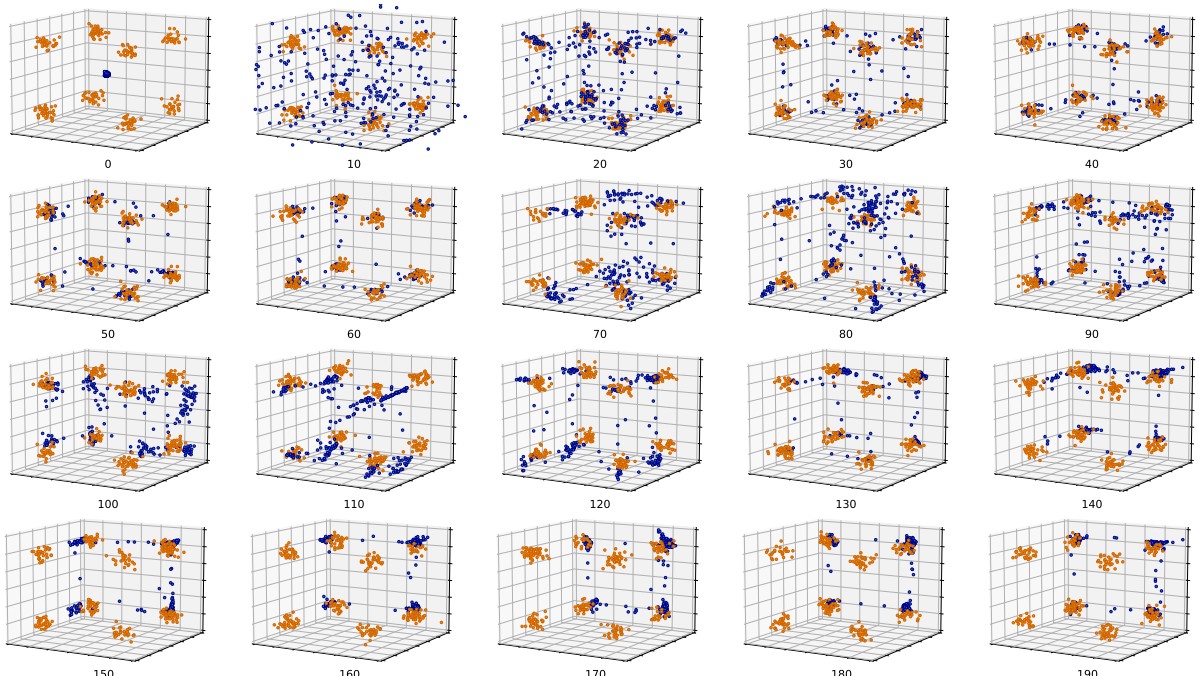

Figure 21: Generated samples in trial 3. The stopping point is the 77th epoch. Before the stopping point, generated samples demonstrate notable quality, particularly evident around the 60th epoch. As training approaches the stopping point, they start to deviate from the modes. After the stopping, they undergo a gradual deterioration, ultimately collapsing to approximately half of the 8 modes.

### G.5 Early Stopping on MNIST: More Training Runs

We train NSGAN on MNIST and record down $||\nabla d_\omega/d_\omega||_2$ each epoch until the maximum specified epochs. The threshold is set to 1877 for reasons in appendix F.2. Instead of halting training when this norm exceeds the threshold, we opt to continue training, which allows us to assess the sample quality both before and after the stop.

In fig. 22, we present the tendency of $||\nabla d_\omega/d_\omega||_2$ against epochs in four trials. We consistently observe a pattern where these values show a smooth initial increase, followed by turbulent fluctuations, culminating in a sudden surge surpassing the threshold of 1877, and then dropping to approximately zero thereafter. The generated samples in the four trials are displayed in fig. 23, fig. 24, fig. 25, and fig. 26, respectively. Note that these figures should be interpreted from left to right, where each column of images represents the generated samples from the epoch indicated at the bottom of the column.

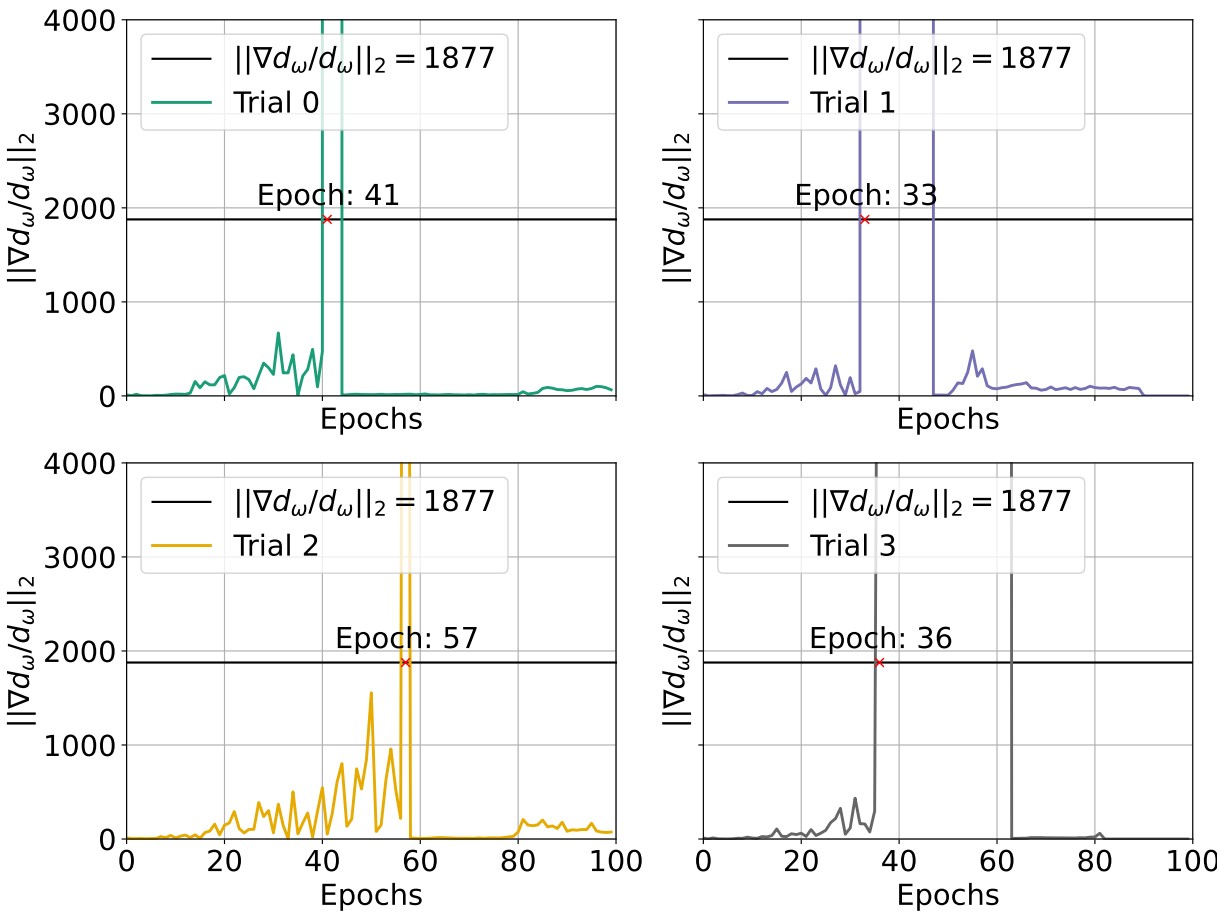

Figure 22: The values of $||\nabla d_\omega/d_\omega||_2$ against epochs in four trials. We mark the stopping epochs by red crosses. The patterns of $||\nabla d_\omega/d_\omega||_2$ values across epochs are consistent across in four training runs. The values exhibit a smooth initial increase, followed by turbulent fluctuations, culminating in a sudden surge above the 1877 threshold, before dropping to approximately zero.

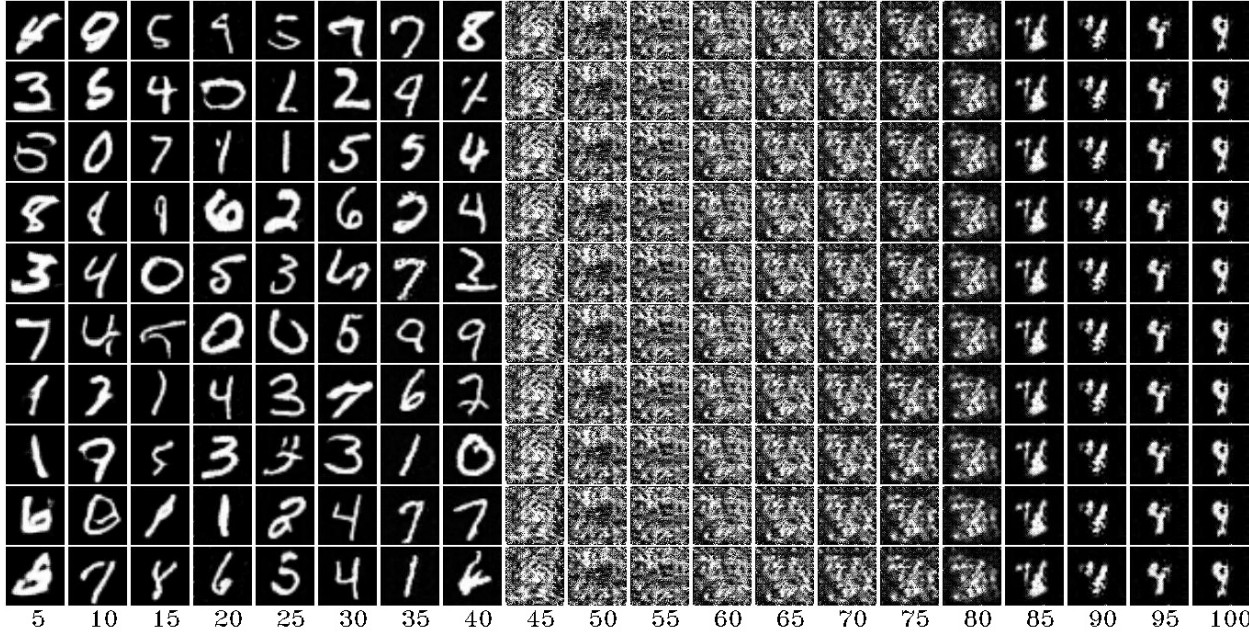

Figure 23: Generated images in trial 0. The stopping point is the 41st epoch. Before reaching the stopping point, the generated samples exhibit good quality, notably around the 30th epoch. Shortly after the stop, they are contaminated with noise. Then they gradually regain clarity and ultimately collapse to one of the modes.

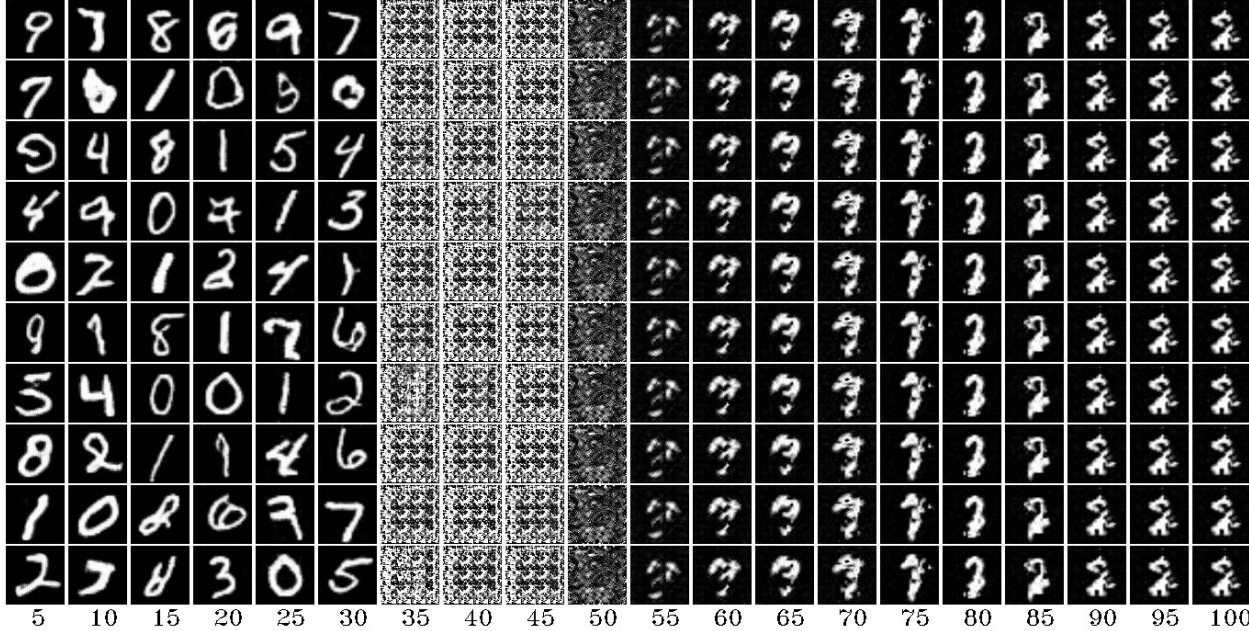

Figure 24: Generated images in trial 1. The stopping point is the 33rd epoch. Before reaching the stopping point, the generated samples exhibit good quality, notably around the 30th epoch. Shortly after the stop, they are contaminated with noise. Then they gradually regain clarity and ultimately collapse to one of the modes.

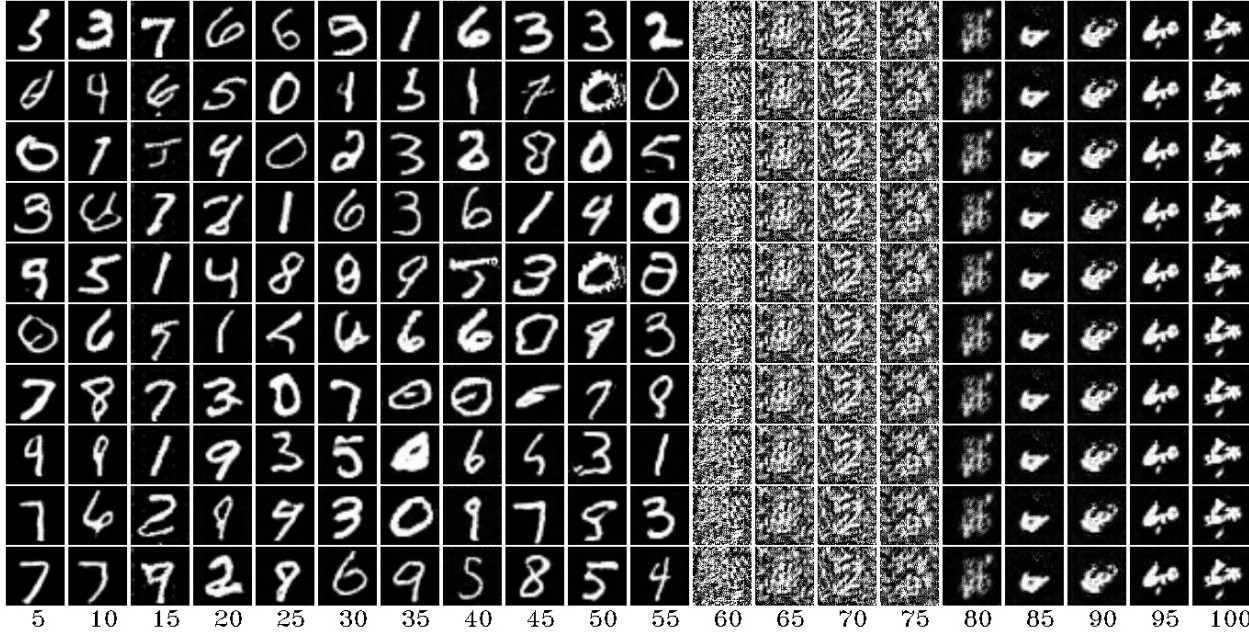

Figure 25: Generated images in trial 2. The stopping point is the 57th epoch. Before reaching the stopping point, the generated samples exhibit good quality, notably around the 55th epoch. Shortly after the stop, they are contaminated with noise. Then they gradually regain clarity and ultimately collapse to one of the modes.

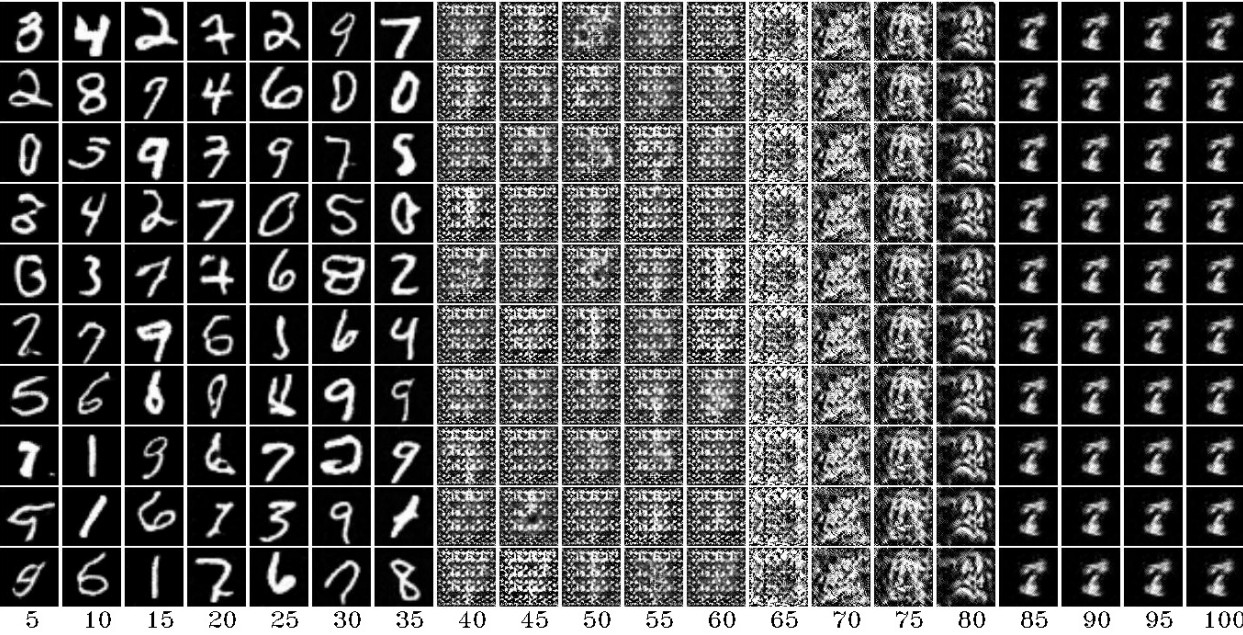

Figure 26: Generated images in trial 3. The stopping point is the 36th epoch. Before reaching the stopping point, the generated samples exhibit good quality, notably around the 35th epoch. Shortly after the stop, they are contaminated with noise. Then they gradually regain clarity and ultimately collapse to one of the modes.

### G.6 Early Stopping on Fashion MNIST: More Training Runs

We train NSGAN on Fashion MNIST and record down $||\nabla d_\omega/d_\omega||_2$ each epoch until the maximum specified epochs. The threshold is set to 2679 for reasons in appendix F.2. Instead of halting training when this norm exceeds the threshold, we opt to continue training, which allows us to assess the sample quality both before and after the stop.

In fig. 27, we present the tendency of $||\nabla d_\omega/d_\omega||_2$ against epochs in four trials. We consistently observe a pattern where these values exhibit a smooth initial increase, followed by turbulent fluctuations, culminating in a sudden surge to surpass the threshold of 2679. Note that in the phases of fitting and refining, $||\nabla d_\omega/d_\omega||_2$ may exceed the threshold as well. This situation justifies the necessity of introducing a warm-up training iteration parameter $N_w$ in algorithm 2. The generated samples in the four trials are displayed in fig. 28, fig. 29, fig. 30, and fig. 31, respectively. Note that these figures should be interpreted from left to right, where each column of images represents the generated samples from the epoch indicated at the bottom of the column.

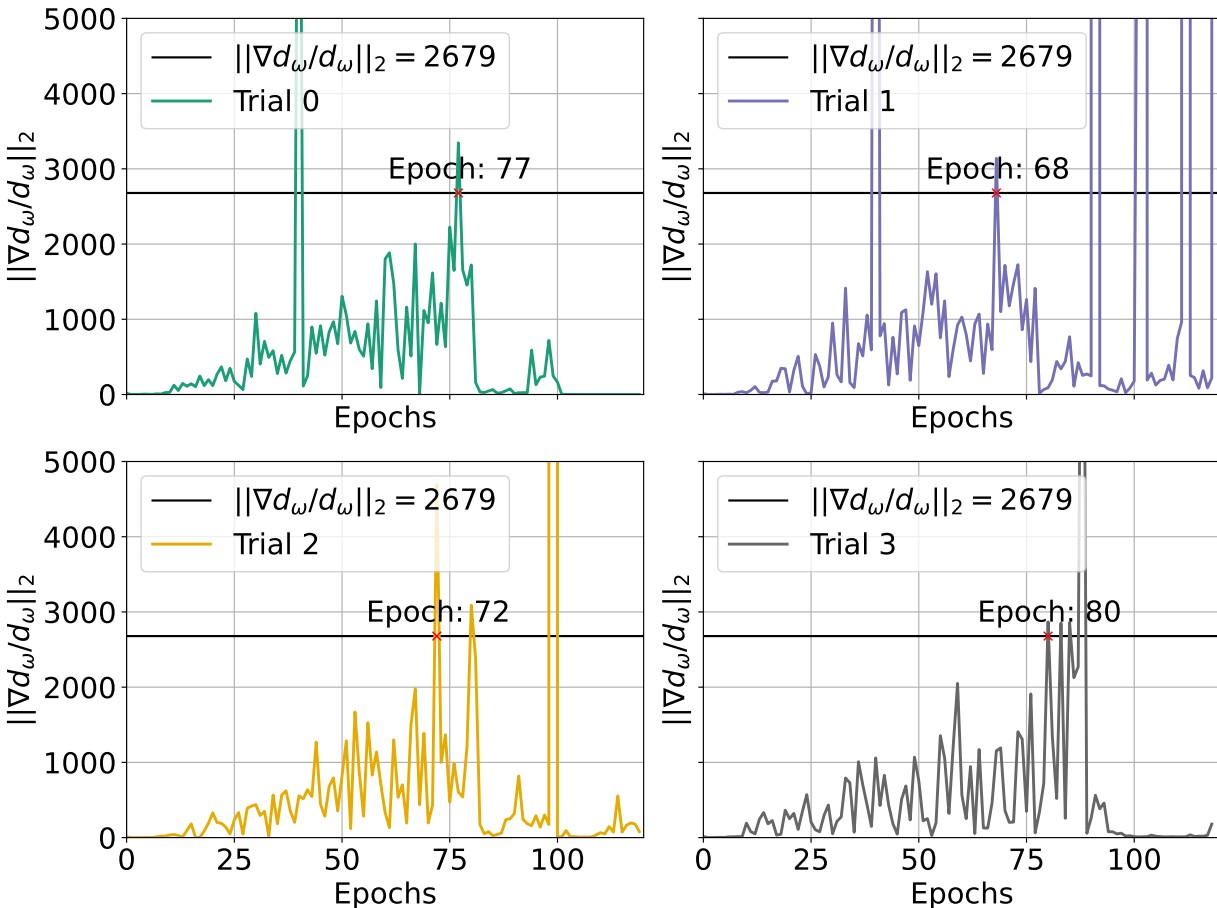

Figure 27: The values of $||\nabla d_\omega/d_\omega||_2$ against epochs in four trials. We mark the stopping epochs by red crosses. Note that only the values of the epochs after $N_w = 50$ (where $N_w$ is the warm-up training iteration parameter defined in algorithm 2) that exceed 2679 may be considered as potential stopping points. The patterns of $||\nabla d_\omega/d_\omega||_2$ values across epochs are consistent in all four training runs. Initially, there is a gradual rise, followed by turbulent fluctuations, and eventually a sharp surge above the 2679 threshold.

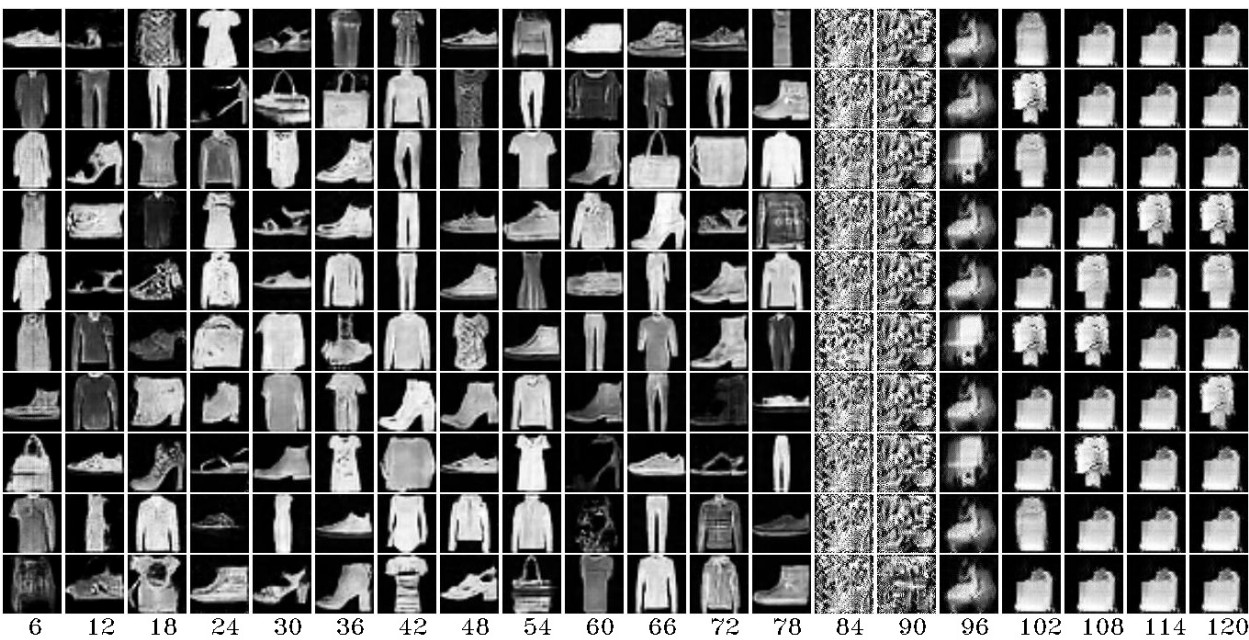

Figure 28: Generated images in trial 0. The stopping point is the 77th epoch. Before the stopping point, the generated samples demonstrate good quality, particularly evident around the 66th epoch. Shortly after surpassing this point, they become contaminated by noise. After that, there is a gradual recovery of clarity, ultimately resulting in convergence towards two specific modes.

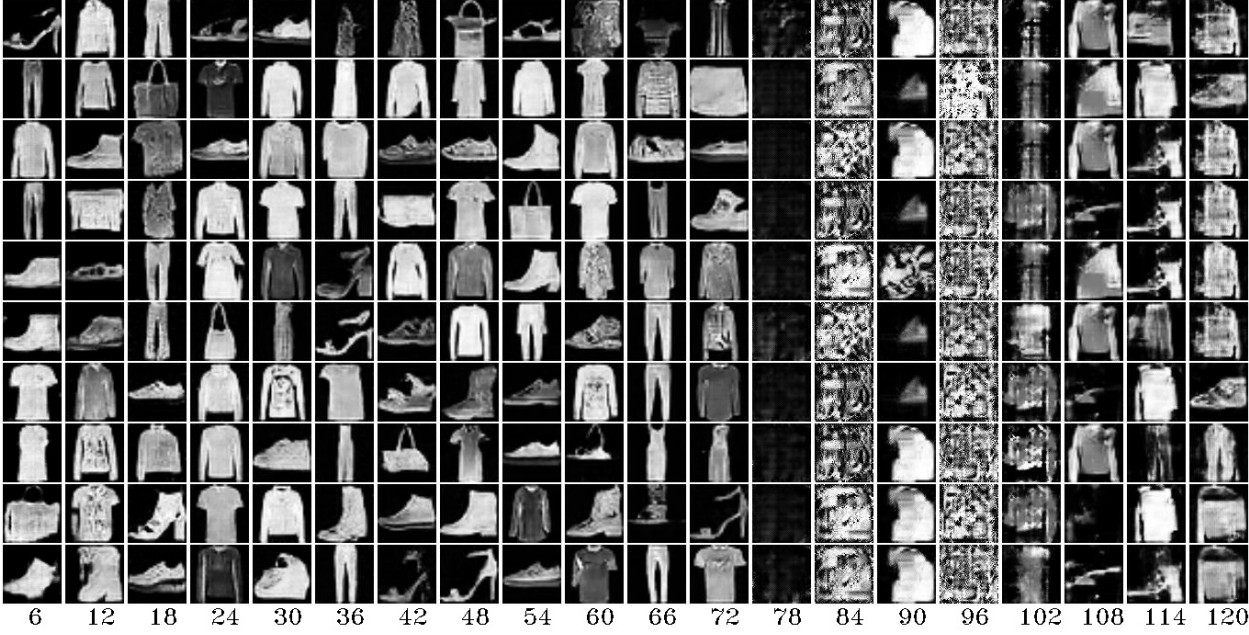

Figure 29: Generated images in trial 1. The stopping point is the 68th epoch. Before the stopping point, the generated samples demonstrate good quality, particularly evident around the 66th epoch. Shortly after surpassing this point, they become contaminated by noise. After that, there is a gradual recovery of clarity, ultimately resulting in convergence towards three specific modes.

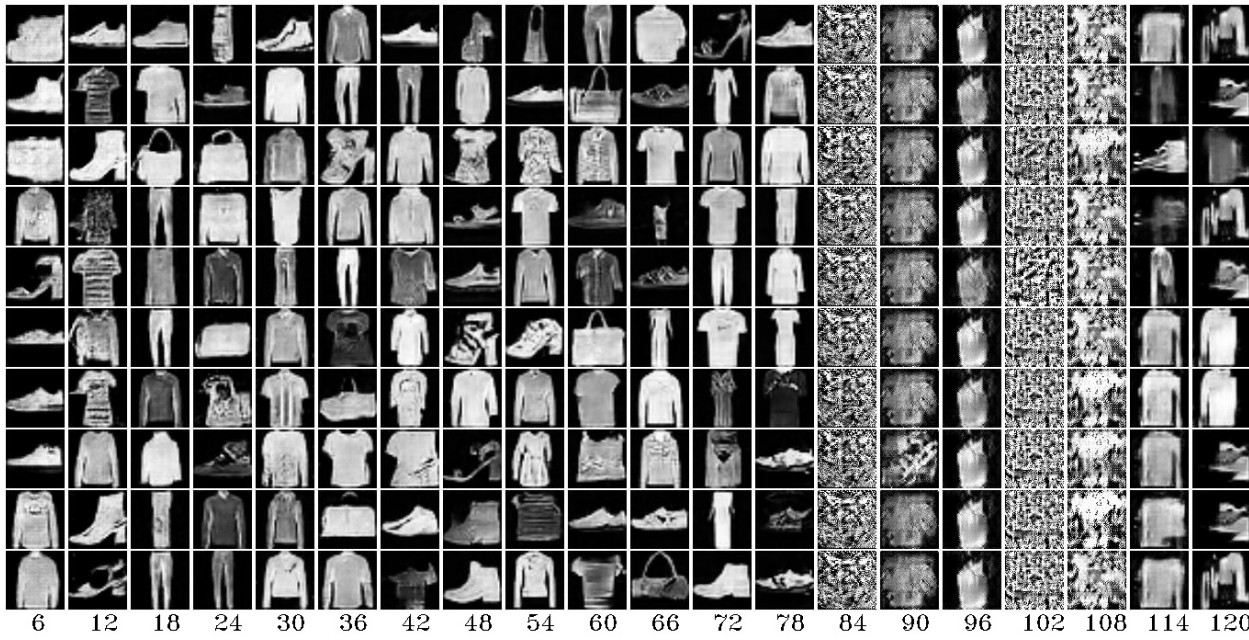

Figure 30: Generated images in trial 2. The stopping point is the 72nd epoch. Before the stopping point, the generated samples demonstrate good quality, particularly evident around the 66th epoch. Shortly after surpassing this point, they become contaminated by noise. After that, there is a gradual recovery of clarity, ultimately resulting in convergence towards four specific modes.

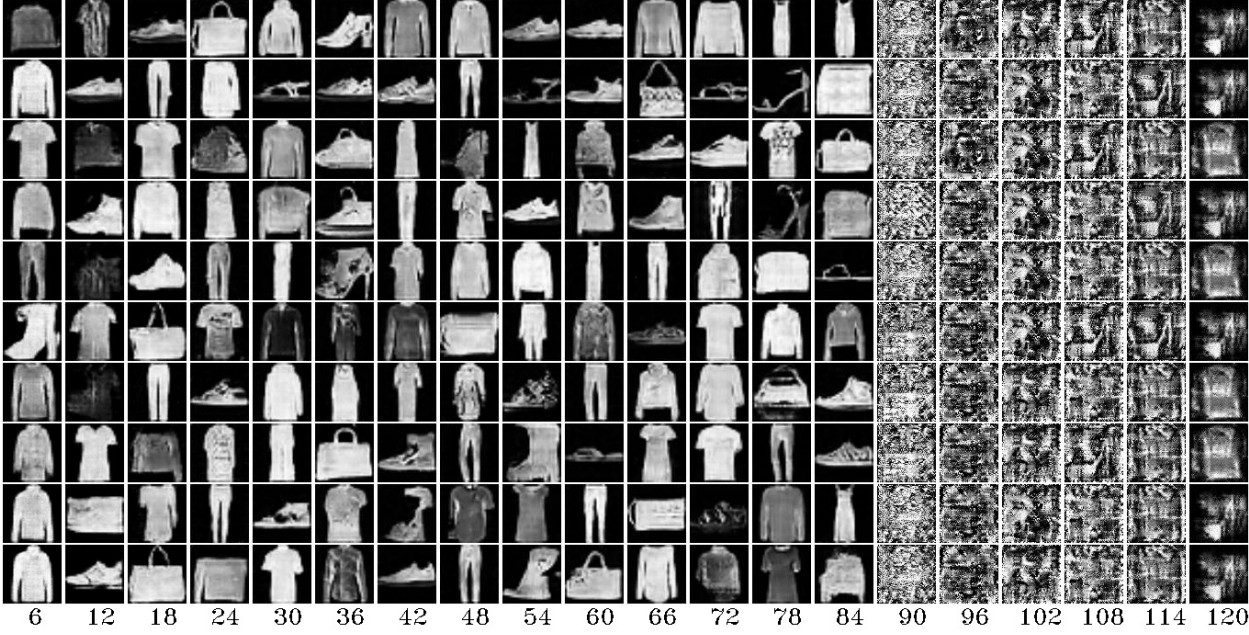

Figure 31: Generated images in trial 3. The stopping point is the 80th epoch. Before the stopping point, the generated samples demonstrate good quality, particularly evident around the 78th epoch. Shortly after surpassing this point, they become contaminated by noise. After that, there is a gradual recovery of clarity, ultimately resulting in convergence towards two specific modes.

### G.7 Early Stopping on CIFAR-10: More Training Runs

We train NSGAN on CIFAR-10 and record down $||\nabla d_\omega/d_\omega||_2$ each epoch until the maximum specified epochs. The threshold is set to 4391 for reasons in appendix F.2. Instead of halting training when this norm exceeds the threshold, we opt to continue training, which allows us to assess the sample quality both before and after the stop.

In fig. 32, we present the tendency of $||\nabla d_\omega/d_\omega||_2$ against epochs in four trials. We consistently observe a pattern where these values exhibit a smooth initial increase, followed by turbulent fluctuations, culminating in a sudden surge to surpass the threshold of 4391. The generated samples in the four trials are displayed in fig. 33, fig. 34, fig. 35, and fig. 36, respectively. Note that these figures should be interpreted from left to right, where each column of images represents the generated samples from the epoch indicated at the bottom of the column.

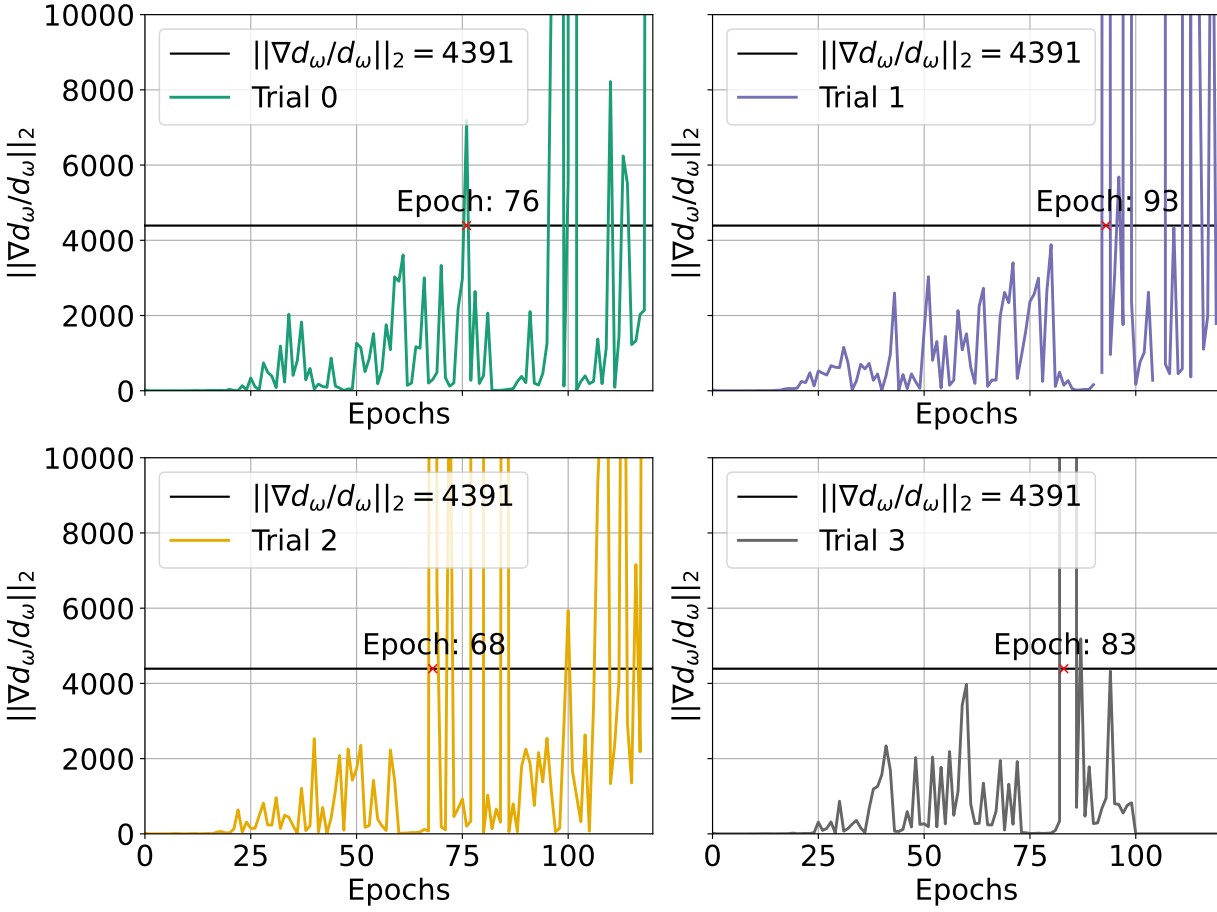

Figure 32: The values of $||\nabla d_\omega/d_\omega||_2$ against epochs in four trials. We mark the stopping epochs by red crosses. The patterns of $||\nabla d_\omega/d_\omega||_2$ values across epochs are consistent in all four training runs. The values show a gradual rise at the beginning, followed by turbulent fluctuations, and then suddenly surge above the threshold of 4391.

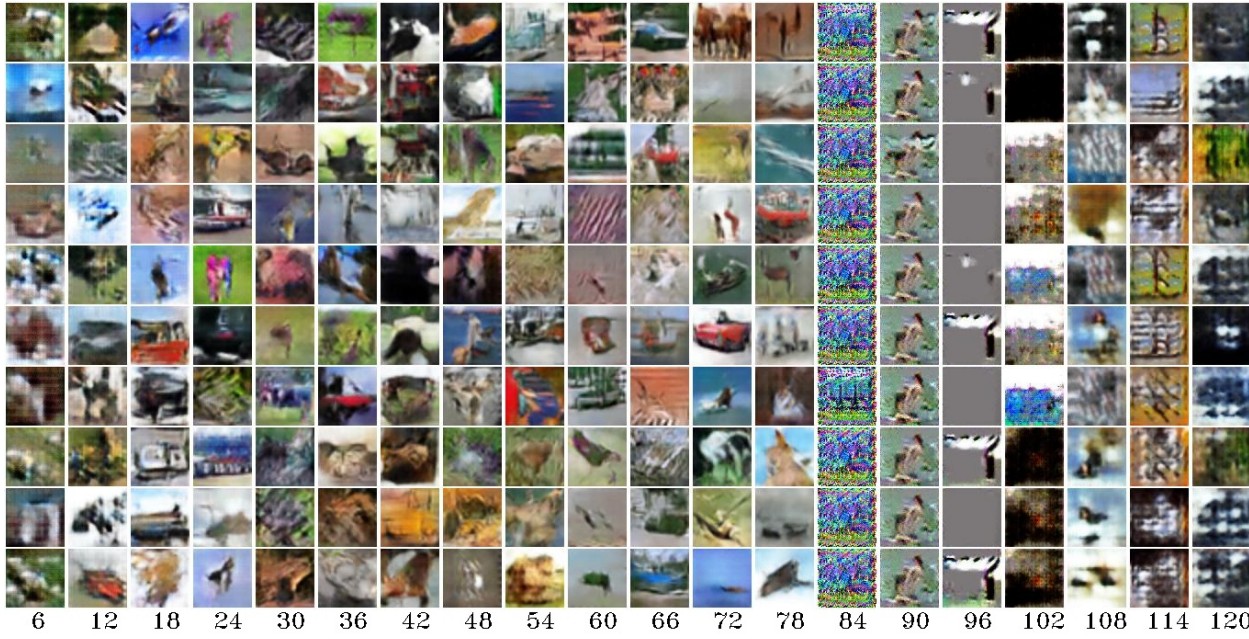

6 12 18 24 30 36 42 48 54 60 66 72 78 84 90 96 102 108 114 120

Figure 33: Generated images in trial 0. The stopping point is the 76th epoch. Before the stopping point, the generated samples demonstrate good quality, particularly evident around the 72th epoch. Shortly after surpassing this point, they become contaminated by noise. Afterwards, they regain clarity, but transition between different modes.

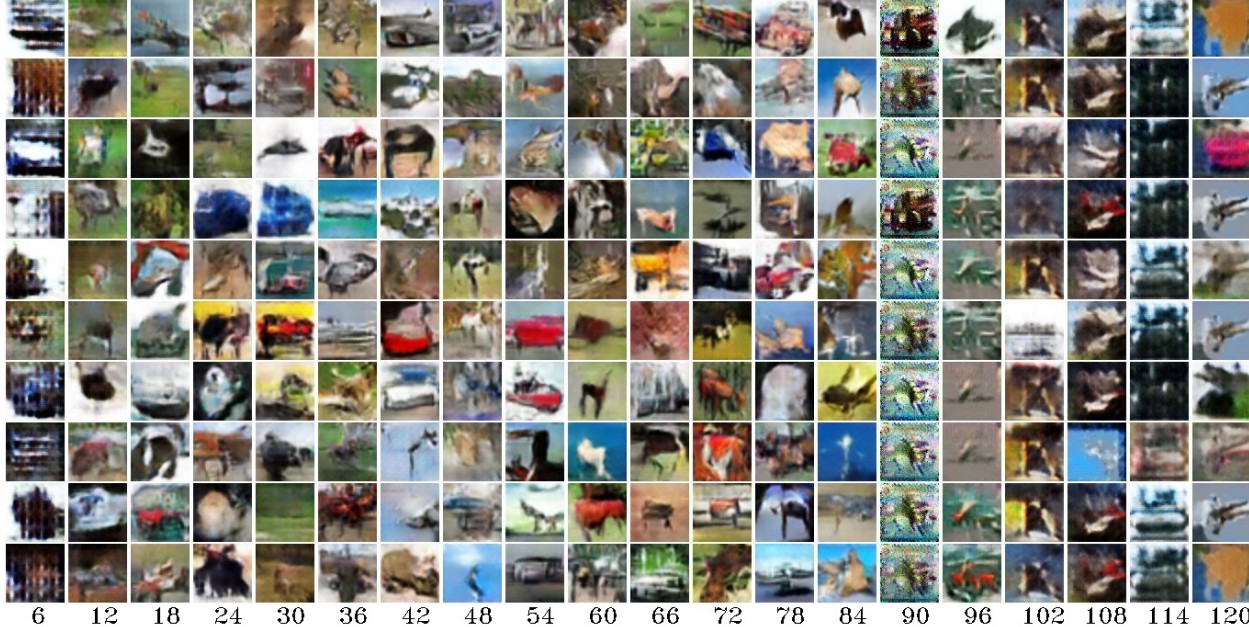

6 12 18 24 30 36 42 48 54 60 66 72 78 84 90 96 102 108 114 120

Figure 34: Generated images in trial 1. The stopping point is the 93rd epoch. Before the stopping point, the generated samples demonstrate good quality, particularly evident around the 78th epoch. Shortly after surpassing this point, they become contaminated by noise. Afterwards, they regain clarity, but transition between different modes.

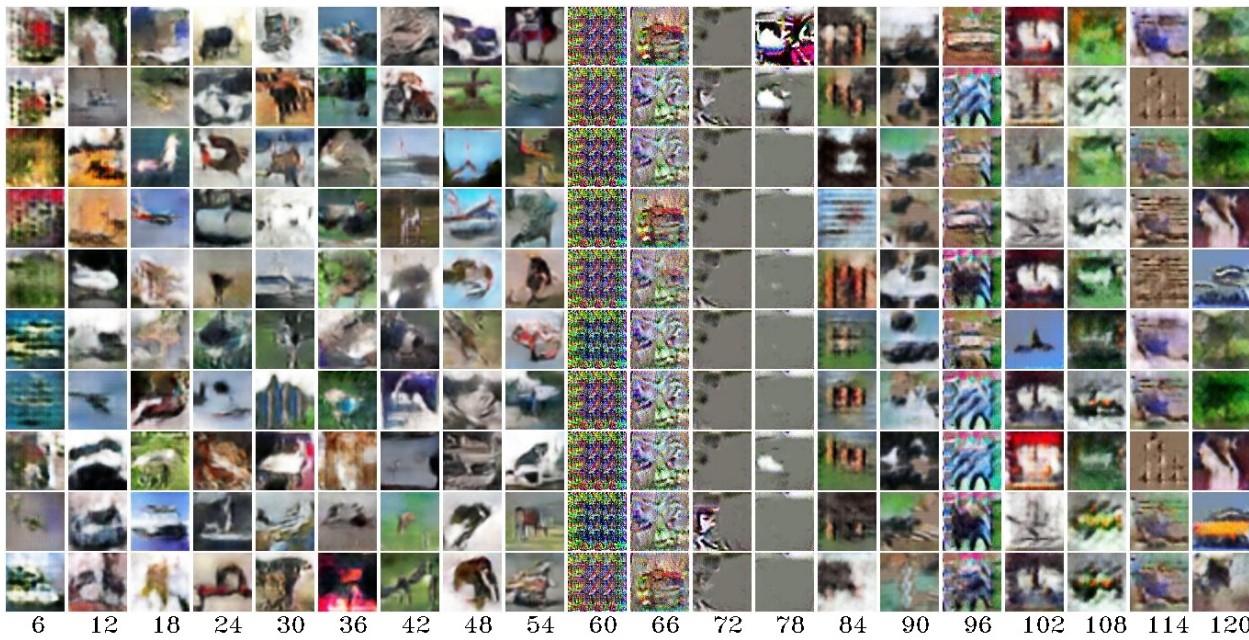

Figure 35: Generated images in trial 2. The stopping point is the 68th epoch. Before the stopping point, the generated samples demonstrate good quality, particularly evident around the 54th epoch. Approaching the stopping point, they become contaminated by noise. Afterwards, they regain clarity, but transition between different modes.

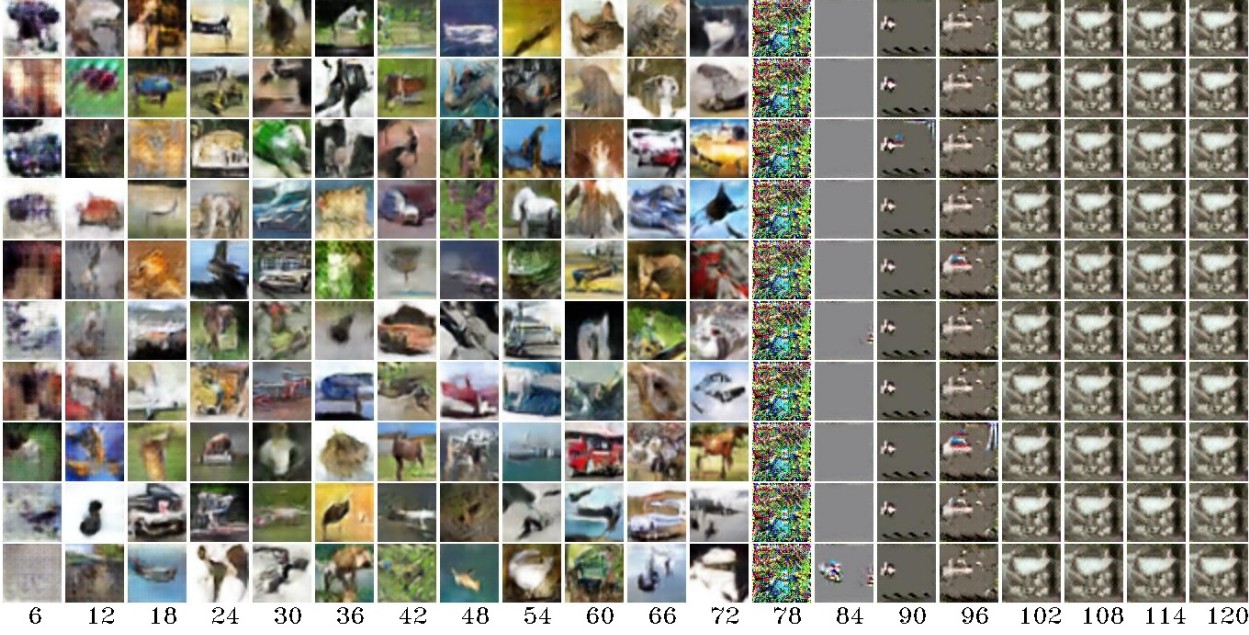

Figure 36: Generated images in trial 3. The stopping point is the 83rd epoch. Before the stopping point, the generated samples demonstrate good quality, particularly evident around the 66th epoch. Shortly after surpassing this point, they become contaminated by noise. Afterwards, they regain clarity and converge towards one specific mode.

### G.8 Comparison Between Our Early Stopping Metric With the FID Score and the Duality Gaps

When evaluating GAN performance, metrics generally fall into two categories: domain-specific and domain-agnostic. For our comparisons, we chose the FID score (Heusel et al., 2017) to represent domain-specific metrics, which focus on the quality of generated images, and duality gaps (Grnarova et al., 2019; Sidheekh et al., 2021) as a representative of domain-agnostic metrics, which assess the optimization process itself. Notably, our early stopping metric $\|\nabla d_\omega/d_\omega\|_2$ (referred to as "the metric" hereafter) is domain-agnostic and does not require real or generated images.

**Comparison with the FID score.** We plot the metric and the FID score in the same figure with shared $x$-axis. The results of four trials for MNIST, Fashion MNIST and CIFAR-10 are shown respectively in fig. 37, fig. 38, and fig. 39.

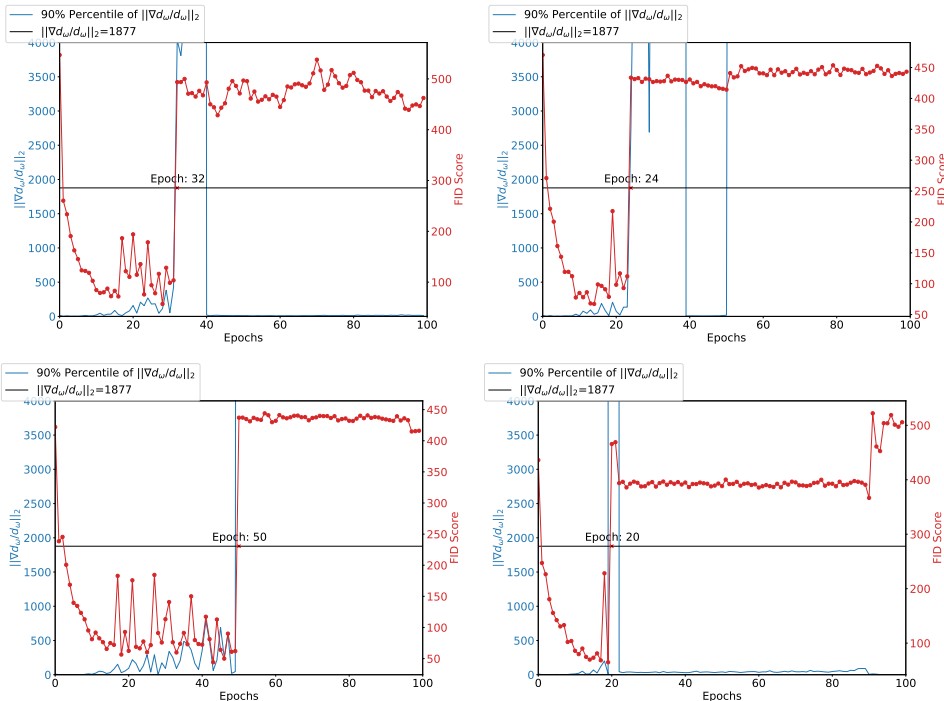

Figure 37: The tendency of $\|\nabla d_\omega/d_\omega\|_2$ and the FID scores on **MNIST in four trials**. Initially, the metric shows a smooth rise, accompanied by a steady decrease in the FID score, indicating the phase of fitting and refining. Subsequently, the metric exhibits turbulent evolution, and the FID score also rises, signaling the conclusion of refining and the prelude of collapsing. Then both the metric and FID score soar. Consequently, the FID score remains consistently high, indicating the collapsing phase.

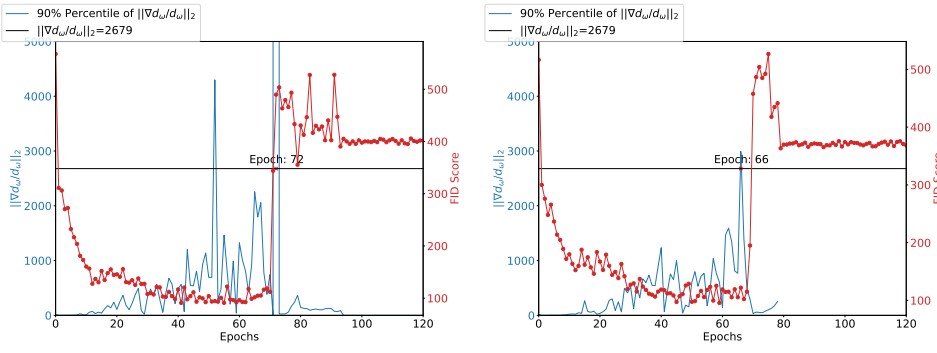

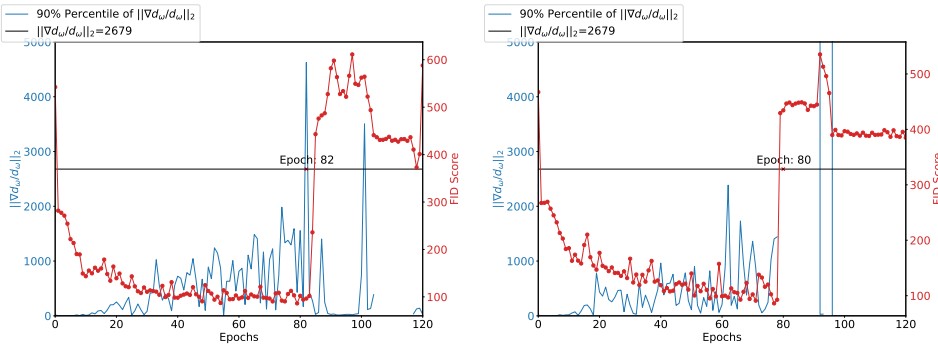

Figure 38: The tendency of $\|\nabla d_\omega / d_\omega\|_2$ and the FID scores on **Fashion MNIST in four trials**. Initially, the metric shows a smooth rise, accompanied by a steady decrease in the FID score, indicating the phase of fitting and refining. Subsequently, the metric exhibits turbulent evolution, but the FID score remains steady. This aligns with our empirical observations in appendix G.6, where generated images demonstrate good quality despite the rapid transition of particles from modes to modes. Then both the metric and FID score soar. Consequently, the FID score remains consistently high, indicating the collapsing phase.

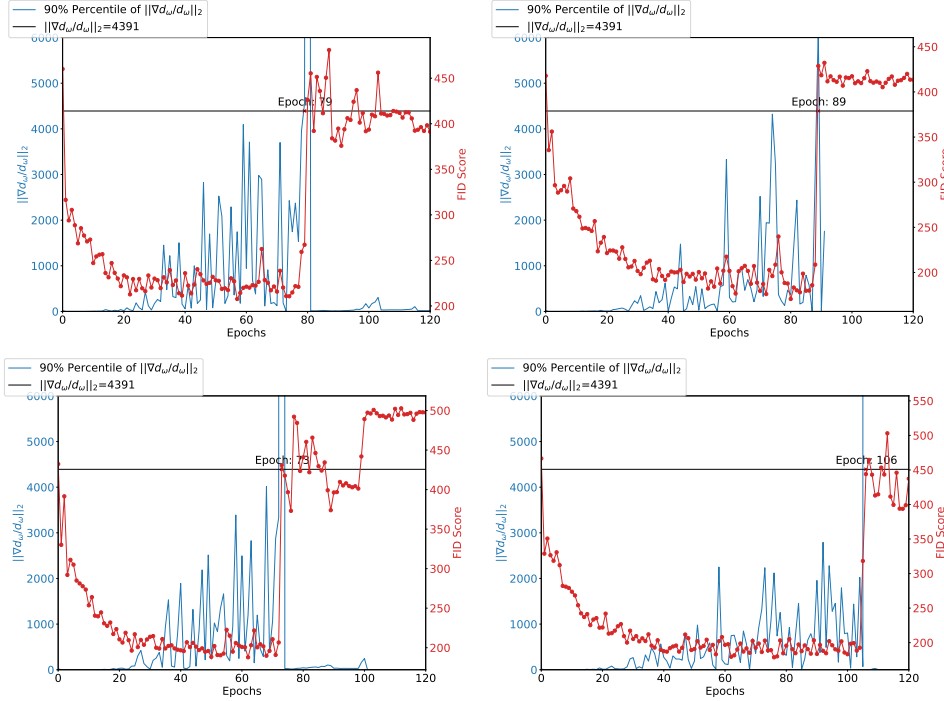

Figure 39: The tendency of $\|\nabla d_\omega / d_\omega\|_2$ and the FID scores on **CIFAR-10 in four trials**. Initially, the metric shows a smooth rise, accompanied by a steady decrease in the FID score, indicating the phase of fitting and refining. Subsequently, the metric exhibits turbulent evolution, but the FID score remains steady. This aligns with our empirical observations in appendix G.7, where generated images demonstrate good quality despite the rapid transition of particles from modes to modes. Then both the metric and FID score soar. Consequently, the FID score remains consistently high, indicating the collapsing phase.

**Comparison with the duality gaps.** We present the comparison between our metric $\|\nabla d_\omega / d_\omega\|_2$ and the duality gap (Grnarova et al., 2019), along with its improved counterpart, the perturbed duality gap (Sidheekh et al., 2021). We first briefly introduce the two metrics, and then show the results in fig. 40.

**Duality gap.** The duality gap is an optimization concept that measures the difference between the primal and dual forms of a problem. In GANs, it quantifies the suboptimality of the current generator and discriminator. For parameters $(\theta_g, \theta_d)$ at a given iteration, the duality gap is defined as:

$$\text{DG}(\theta_g, \theta_d) = \max_{\theta_d' \in \Theta_d} F(\theta_g, \theta_d') - \min_{\theta_g' \in \Theta_g} F(\theta_g', \theta_d),$$

where $\Theta_d$ and $\Theta_g$ are the parameter spaces for the discriminator and generator, respectively, and $F$ is the objective function of the Vanilla GAN: $F(\theta_g, \theta_d) = \mathbb{E}_{\boldsymbol{x} \sim p_{\text{data}}}[\log d(\boldsymbol{x})] + \mathbb{E}_{\boldsymbol{z} \sim p_z}[\log(1 - d(g(\boldsymbol{z})))]$. In practice, Grnarova et al. (2019) proposed to estimate the duality gap through the following steps:

1. Train the GAN to iteration $t$, obtaining parameters $(\theta_g^t, \theta_d^t)$.

2. Find the worst-case discriminator and generator by optimizing one while keeping the other fixed:

$$\theta_d^{\text{worst}} \approx \arg \max_{\theta_d' \in \Theta_d} F(\theta_g^t, \theta_d'), \quad \theta_g^{\text{worst}} \approx \arg \min_{\theta_g' \in \Theta_g} F(\theta_g', \theta_d^t).$$

3. Estimate the duality gap as: $\text{DG}(\theta_g^t, \theta_d^t) \approx F(\theta_g^t, \theta_d^{\text{worst}}) - F(\theta_g^{\text{worst}}, \theta_d^t)$.

**Perturbed duality gap.** The perturbed duality gap, introduced by Sidheekh et al. (2021), improves upon the standard duality gap by more effectively distinguishing between Nash and non-Nash critical points. This metric performs local perturbations to the parameters $(\theta_g^t, \theta_d^t)$ with Gaussian noise before the second optimization step, helping the model escape from saddle points. This ensures the subsequent optimization does not get stuck in suboptimal regions.

**Experimental results.** We observe that before the collapsing phase, the metric resembles the patterns in the perturbed duality gaps. However, at collapse, the metric sharply increases, while the duality gaps behave inconsistently across datasets. For MNIST, both duality gaps drop to zero, obscuring whether this signals mode collapse or convergence. In Fashion MNIST, the vanilla gap drops to zero, but the perturbed gap remains above zero, offering no clear indication. In CIFAR-10, both gaps stay steady, failing to signal collapse. These findings suggest our metric is a more reliable indicator of mode collapse than the duality gaps.

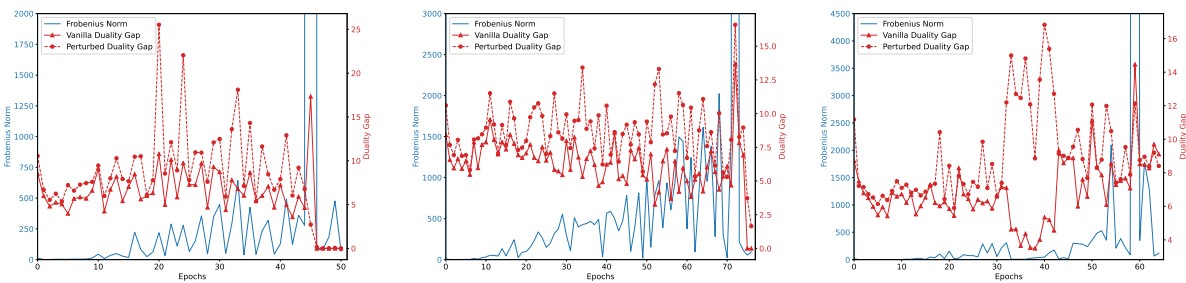

Figure 40: The tendency of $\|\nabla d_\omega / d_\omega\|_2$ and the duality gaps for MNIST, Fashion MNIST, and CIFAR-10, from left to right. The metric is depicted by solid blue lines. The duality gaps are represented by red lines, where the vanilla duality gaps are indicated with dashed lines and dotted markers, while the perturbed duality gaps are indicated with solid lines and triangular markers. We observe that prior to collapsing (occurring at the 46th, 73rd, and 60th epochs for MNIST, Fashion MNIST, and CIFAR-10, respectively), the metric exhibit similar patterns seen in the perturbed duality gaps. At the point of collapsing, the metric sharply rises above the threshold, whereas the duality gaps behave differently across the three experiments. For MNIST, both duality gaps drop to 0, making it unclear whether this indicates mode collapse or convergence. In the case of Fashion MNIST, the vanilla duality gap falls to 0 while the perturbed gap decreases but remains above 0, yet neither provides a definitive indication of the GAN's status. For CIFAR-10, both duality gaps remain steady, offering no clear signal of mode collapse. These findings suggest that the metric is more effective at signaling the collapsing phase.

### G.9 Impact on the Early Stopping Metric after Applying Techniques to Mitigate Mode Collapse

In this subsection, we validate our early stopping metric's effectiveness by demonstrating that injecting noise into the intermediate layers of the discriminator combats mode collapse and pushes back the metric.

**Experimental setup.** We devise two generator models of identical architecture and implement two discriminators, one adhering to the original design (which we will refer to as "noise-free") and the other modified to incorporate Gaussian noise with a standard deviation of 0.1 before forwarding the input to the subsequent layer (which we will refer to as "noised"). Both generators and discriminators are initialized using the same random seed. During training, the four networks are concurrently trained, with each generator paired with a discriminator. We present the generated images of the two models on Fashion MNIST in fig. 41 and histograms of $||\nabla d_\omega / d_\omega||_2$ in fig. 42.

**Results.** The noise-free GAN collapses at the 54th epoch, while the noised GAN consistently generates high-quality images. The introduction of noise results in an overall decrease in the $||\nabla d_\omega / d_\omega||_2$ compared to its noise-free counterpart before the 54th epoch. After the 54th epoch, the opposite trend is observed, attributed to the vanishing of $||\nabla d_\omega / d_\omega||_2$ in the noise-free GAN. This indicates that the strategy of injecting noise to mitigate mode collapse leads to an overall decrease in our proposed metric, thereby validating the effectiveness of the metric.

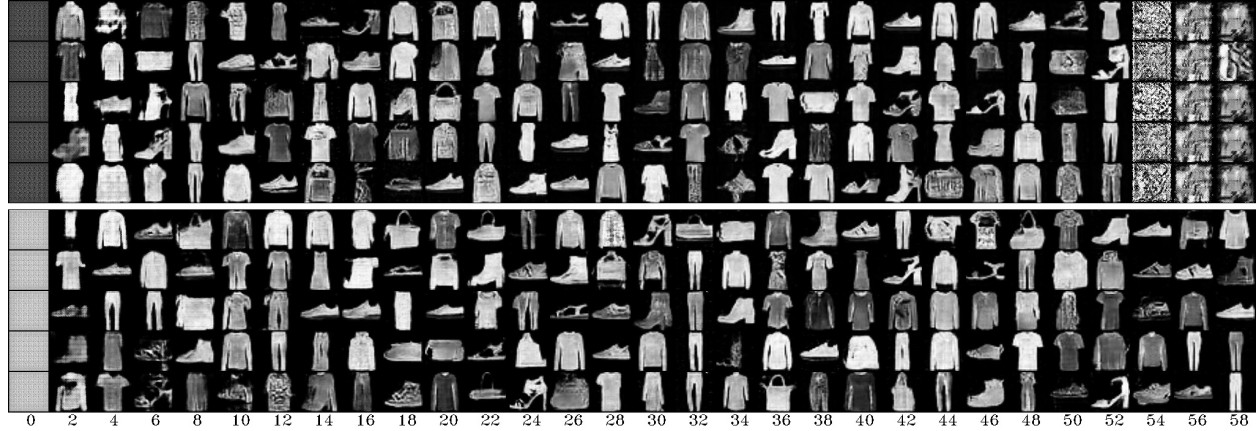

Figure 41: The generated images from the noise-free GAN and the noised GAN. **Upper**: Noise-free GAN. **Lower**: Noised GAN. The noise-free GAN collapses at the 54th epoch, whereas the noised GAN consistently produces high-quality images.

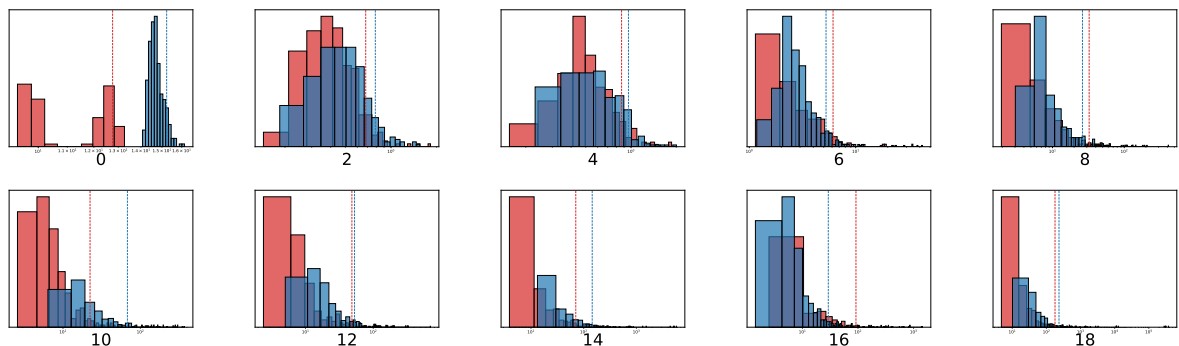

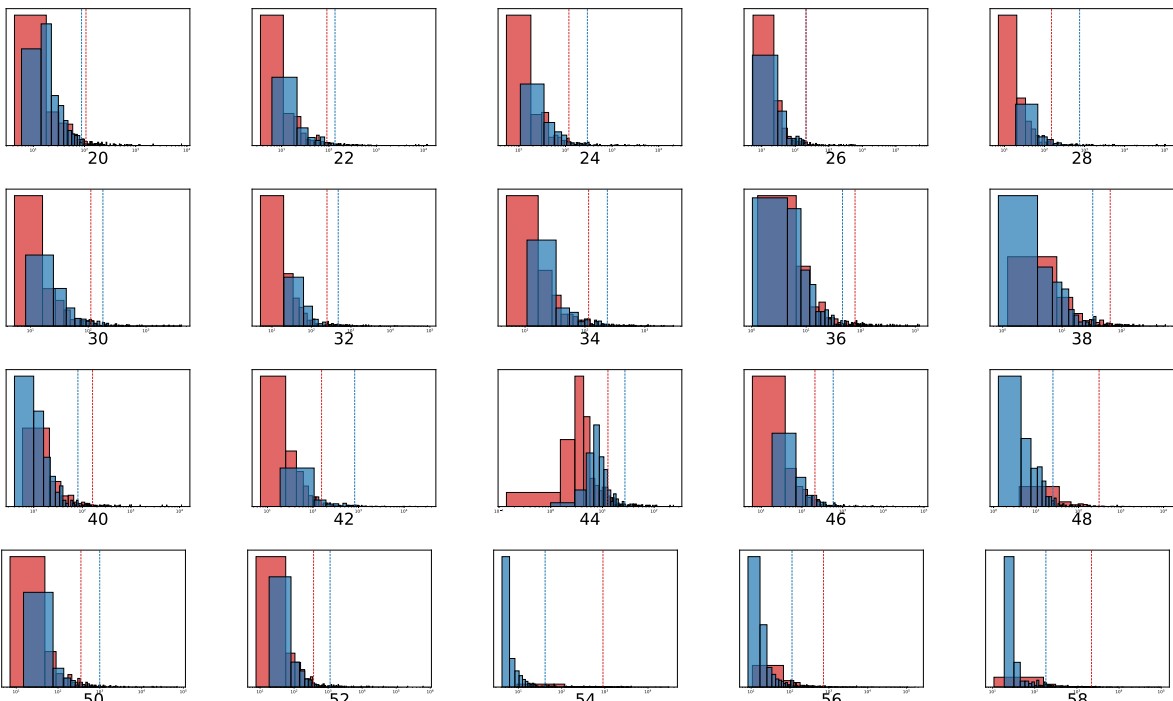

Figure 42: Histograms of the values of $\|\nabla d_\omega/d_\omega\|_2$ and their 90th percentile across epochs. The epochs are displayed at the bottom of each histogram. The $x$-axis represents $\|\nabla d_\omega/d_\omega\|_2$ values on a *logarithmic* scale, while the $y$-axis denotes density. Results are differentiated by color: red for the model with noise and blue for the model without noise. Preceding the 54th epoch where the noise-free GAN collapses, the noised model nearly always exhibits lower $\|\nabla d_\omega/d_\omega\|_2$ values compared to its noise-free counterpart. Post 54th epoch, this relationship reverses. Notably, in the noise-free model, $\|\nabla d_\omega/d_\omega\|_2$ tends towards zero, contributing to this observed divergence.

## H   Rationale Behind the Discriminator Model

The rationale behind selecting the radius as $2\sqrt{2}h$ emanates from the equivalence of an RBF kernel with bandwidth $h$ to a Gaussian kernel of variance $h^2/2$. For a Gaussian distribution, 99 percent of samples lie within a sphere centered at the mean and extending to 3 times the standard deviation. We further extend the boundary to 4 times the standard deviation in light of potential mode mixture.

## I   Extension to Other Divergence GANs

In this section, we outline how our framework can be extended to encompass other Divergence GANs. We focus on the $f$-GAN proposed in (Nowozin et al., 2016) with the $f$-divergence defined as

$$D_f(Q_\theta || p_{\text{data}}) = \int_{\boldsymbol{x}} p_{\text{data}}(\boldsymbol{x}) f\left(\frac{p_{\text{data}}(\boldsymbol{x})}{Q_\theta(\boldsymbol{x})}\right) \mathrm{d}\boldsymbol{x}.$$

The variational lower bound of $D_f(Q_\theta || p_{\text{data}})$ is used as the training objective:

$$F(\theta; \omega) = \mathbb{E}_{\boldsymbol{x} \sim p_{\text{data}}}\big[g_f\big(V_\omega(\boldsymbol{x})\big)\big] + \mathbb{E}_{\boldsymbol{x} \sim Q_\theta}\big[-f^*\big(g_f(V_\omega(x))\big)\big].$$

Here, $f^*$ is the Fenchel conjugate of $f$, $g_f$ is analogous to the generator and $V_\omega$ is similar to the discriminator. We consider its variant where the objective function of the generator is modified to

$$-\mathbb{E}_{\boldsymbol{x} \sim Q_\theta}\big[g_f\big(V_\omega(\boldsymbol{x})\big)\big],$$

while the objective function of the discriminator remains unchanged.

**General methodology.** The key to analyzing Divergence GANs is their interpretation as particle models. The update of the generator $Q_\theta$ can be recast as:

- Generate particles $Z_i = Q_\theta(z_i)$.

- Update the particles $\hat{Z}_i = Z_i + g'_f(V_\omega(Z_i))\nabla V_\omega(Z_i)$.

- Update $\theta$ by descending its stochastic gradient with respect to the Mean Square Error (MSE) loss betweeen $\hat{Z}_i$'s and $g(z_i)$'s.

**Fitting phase.** We may plot the vector field $g'_f(V_\omega(Z_i))\nabla V_\omega(Z_i)$ instead of the original $\nabla d_\omega / d_\omega$ to visualize the updating process of particles, which promotes the fitting of the modes.

**Refining phase.** Only theorem 4.3 in section 4.3 needs to be modified to accommodate the desired Divergence GAN.

**Collapsing phase.** In section 5.1, apart from modifying the update formula for particles, a more appropriate model for the discriminator needs to be established in assumption 5.1 and a new threshold may be developed on the basis of it in algorithm 2.

## J   Visualizing Generator Functions

This section visualizes generator functions $g$ that satisfy $g_\# p_z = p_{\text{data}}$, where $p_z \sim \mathcal{N}(0,1)$ and $p_{\text{data}}$ is a Gaussian mixture, as shown in fig. 43. For qualitative effects of the parameters in $p_{\text{data}}$, please refer to table 2. We then discuss about how to plot fig. 43. While $\Phi$ can be computed in MATLAB using the built-in function `normcdf`, $\Psi^{-1}$ typically necessitates solving a non-linear equation at each evaluation point. To mitigate computational expenses, we choose to calculate the inverse of $g$, which is $g^{-1} = \Phi^{-1} \circ \Psi$. In this context, $\Psi$ can be computed by employing `gmdistribution` to construct a Gaussian mixture model, followed by utilizing `cdf` to assess the cumulative distribution function (CDF) of the model at a specific point. To generate a plot of $g$, a mere interchange of the $x$ and $g^{-1}(x)$ in the `plot` function suffices.

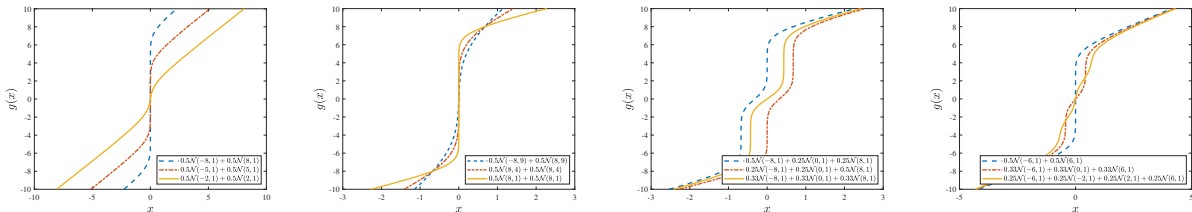

Figure 43: The functions $g$ that satisfy $g_\# p_z = p_{\text{data}}$, where $p_{\text{data}}$ is a Gaussian mixture. **First**: Varying the mean $\mu$. **Second**: Varying the variance $\sigma^2$. **Third**: Varying the mixing coefficients $\{\alpha_i\}_{i=1}^n$. **Fourth**: Varying the number of Gaussians $n$. Please refer to table 2 for a detailed description.

Table 2: Qualitative effects of the parameters in $p_{\text{data}} \sim \alpha_1 \mathcal{N}(\mu_1, \sigma^2) + \cdots + \alpha_n \mathcal{N}(\mu_n, \sigma^2)$ on $g$.

| Parameters | Qualitative Effects on $g$ |
|---|---|
| Means $\{\mu_i\}_{i=1}^n$ | Larger $\|\mu_i - \mu_{i+1}\|_2$ increases the magnitude of $g'$ between the two modes. |
| Variances $\sigma^2$ | Larger $\sigma^2$ increases the asymptotic slope of $g$ as $x \to \infty$. |
| Mixing coefficients $\{\alpha_i\}_{i=1}^n$ | Different combinations of $\alpha_i$ incline $g$ towards specific modes. |
| Number of Gaussians $n$ | Larger $n$ increases the number of segments in $g$. |

## K    Discussions

In this section, we provide additional intuitions and implications.

In terms of applicability scope, our theoretical framework is primarily derived from Divergence GANs, specifically NSGAN, where we can leverage their particle model interpretations. While Divergence GANs represent a significant category within GANs, they do not encompass some prominent GAN models, such as Wasserstein GAN with gradient penalty and MMD GAN. Exploring how our theoretical framework can be extended to incorporate these Integral Probability Metric (IPM) based GAN variants presents an intriguing avenue for future research.

Regarding the theoretical framework, it is important to note that not all Divergence GANs may fit neatly into the tripartite phases. While we often observe such empirical patterns, we acknowledge the possibility that when networks are not well-initialized or when advanced techniques are used, GAN training may deviate from the fitting phase entirely. However, these inquiries may spark independent interests and are beyond the scope of our study.

In our numerical experiments, we used relatively small-scale real-world datasets compared to modern datasets. This choice was deliberate as we aimed to assess the effectiveness of our early stopping algorithm in detecting the transition from refining to collapsing phases. Modern datasets often comprise exponentially more modes, which could potentially limit the efficacy of our algorithm, particularly considering that our algorithm takes the number of modes as an input parameter.

