# OpenReview forum: "Magnifying the Three Phases of GAN Training — Fitting, Refining and Collapsing"
_TMLR — Rejected by TMLR_

### Review · Reviewer_EpvX · 2024-06-14

**Summary Of Contributions:**

This paper presents an approach to characterize GAN training into three phases--fitting, refining, collapsing. First, the author proposed some thoughts that NSGAN can be converted into particle methods by adding the perturbation calculated via the gradient of the discriminator to the input. Second, a case study of Gaussian mixture models is carried to justify how GANs refine samples. Finally, mode collapse problem is discussed with how it happens and how it can be addressed by early stopping trick.

**Audience:**

Yes

**Broader Impact Concerns:**

n.a.

**Claims And Evidence:**

Yes

**Requested Changes:**

1. Question for the proof of theorem 2.1. I understand this proof technique is used in almost all relevant papers mentioned by the author. However, one thing I would like to point out is the once you applied the differentiation operator $\nabla_\theta$, you started considering $\hat{Z}_i$ as a constant which is non-differentiable w.r.t. $\theta$. But, in line 5 of Algorithm 1, $\hat{Z}_i$ does have involved the parameter $\theta$ through $Z_i$. In this case, you have to use a stop gradient approach in the implementation to ensure theorem 2.1 can be true. But applying the stop gradient operator makes this theorem less mathematically rigorous.
2.  Theorem 3.1 seems to be redundant. This is just the standard result of sigmoid transformation of the logit output ($\log r(x)$) of a binary classifier. What is the conclusion the author wants to reach at this point? The following implications of theorem 3.1 are just the empirical description of how the particles evolve with this gradient update. Instead, this vector field decreases the KL divergence in a L2 space [1] or the KL divergence in Wasserstein space [2]. In [2], it shows the vector field of the NSGAN corresponds to the Wasserstein gradient of a certain f-divergence but no longer the Jensen Shannon.  I also found the implication after theorem 3.1 is slightly overlapped with [2] (section 5.1).
3. For sections 3.2 and 3.3, I wish there are more theoretical results. As pointed out earlier, this NSGAN vector field is the velocity field of Wasserstein gradient flow of $f$-divergence.
4. In section 4.1, the authors claimed "under the 2-Wasserstein distance (Santambrogio, 2015)". If I understand it correctly, section 2.1 in (Santambrogio, 2015) shows this transport map applies to p-Wasserstein distance not only 2-Wasserstein distance.
5. I appreciate the authors' efforts on illustrating differences in Figure 3. While I was reading this part, I was a bit getting lost. I would like to suggest to add a table to summarize which affects which.
6. In section 5.1, I did not get why the boundary will shrink. Can the authors give more explanations on that?


references:

[1] Johnson, Rie, and Tong Zhang. "A framework of composite functional gradient methods for generative adversarial models." IEEE transactions on pattern analysis and machine intelligence 43.1 (2019): 17-32.

[2] Yi, Mingxuan, Zhanxing Zhu, and Song Liu. "MonoFlow: Rethinking divergence GANs via the perspective of Wasserstein gradient flows." International Conference on Machine Learning. PMLR, 2023.

**Strengths And Weaknesses:**

Strengths:
The writing is clear and easy to follow. The analysis complemented with excellent illustrations makes sense. The ideas on steepness and the early stopping approach are indeed novel. The empirical results are sufficient to support the claims.


Weaknesses:
First of all, rethinking divergence GANs as particle methods is not new. Some research has been done in this direction. For example, discrete time version [1] and the continuous-time version [2]. Both these two papers explains how NSGAN performs particle descent. The illustration on Gaussian mixtures make sense to me, but can this empirical result generalize to image generation problems? I believe the generalization of such an analysis needs to be discussed.

 I am still wondering if or not the Gaussian mixture case is the same as image generation problems. Furthermore, training GANs on 2D data is known to be more difficult than image generation problems. One reason could be the NTK structure from the generator [3]. This could result in some geometry making fitting a neural generator to 2D data extremely hard. One can imagine the particle GANs (removing the generator, you only consider the vector field derived from the discri) has no interacting mechanism for each particle.  However, once you add a generator, particles become correlated via the generator's parameters such that there is some kind of mean-field dynamics on particles. The paper needs to take this into account, otherwise the analysis on 2D toy data based on the discriminator guidance cannot really generalize to the original GAN scenario.

Another question regarding the early stopping, in my experience, mode collapse sometimes happens at the beginning, i.e., generator directly output single mode images instead of showing the procedure in Figure 8. Can the author give some analysis on this?

[1] Johnson, Rie, and Tong Zhang. "A framework of composite functional gradient methods for generative adversarial models." IEEE transactions on pattern analysis and machine intelligence 43.1 (2019): 17-32.

[2] Yi, Mingxuan, Zhanxing Zhu, and Song Liu. "MonoFlow: Rethinking divergence GANs via the perspective of Wasserstein gradient flows." International Conference on Machine Learning. PMLR, 2023.

[3] Franceschi, Jean-Yves, et al. "A neural tangent kernel perspective of GANs." International Conference on Machine Learning. PMLR, 2022.

---

> ### Author Response · Authors · 2024-07-18
> **Thank You and a Kind Reminder on Submitted Revised Manuscript**
>
> Dear Reviewer EpvX,
>
> Thank you for your detailed and very insightful feedback on our paper. We greatly appreciate the time and effort you have put into reviewing our manuscript and providing such valuable comments.
>
> We hope that addressing the current feedback promptly will help streamline the review process and ensure a smoother progression. As you may have noticed, we submitted the revised version of our manuscript on June 27, incorporating the constructive feedback from you. For any concerns you raised that may not seem directly addressed in the revised manuscript, we will provide detailed responses during the subsequent rebuttal phase.
>
> Once again, we sincerely thank you for your thorough review and valuable suggestions.
>
> Best regards,
>
> Paper2643 Authors

---

> > ### Comment · Reviewer_EpvX · 2024-07-18
> > **Thank you for your reminder**
> >
> > Hi, thanks for reminding me of the revised paper. It seems that I missed the notification for that. I will have a look by this week.

---

> ### Author Response · Authors · 2024-07-19
> **A Point-by-Point Response to Comments (Part 1)**
>
> Dear Reviewer EpvX,
>
> Thank you once again for the time and effort you have dedicated to reviewing our work. Below, we will address your comments point-by-point.
>
> ----
>
> ## Update on Revised Manuscript:
>
> We have submitted another revised version of our manuscript, with changes since the first submission highlighted in **red**, as suggested by the Action Editor.
>
> ----
>
> ## Responses to Weaknesses:
>
> ----
>
> ### Rethinking divergence GANs as particle methods
>
> > First of all, rethinking divergence GANs as particle methods is not new. Some research has been done in this direction. For example, discrete time version [1] and the continuous-time version [2]. Both these two papers explains how NSGAN performs particle descent.
>
> Thank you for pointing out the relevant literature on rethinking divergence GANs as particle methods. We have now included the two papers you mentioned, the discrete-time version [1] and the continuous-time version [2], in our revised manuscript.
>
> ----
>
> ### Generalization to image generation problems and the interaction between generator and discriminator
>
> > The illustration on Gaussian mixtures make sense to me, but can this empirical result generalize to image generation problems? I believe the generalization of such an analysis needs to be discussed. I am still wondering if or not the Gaussian mixture case is the same as image generation problems. Furthermore, training GANs on 2D data is known to be more difficult than image generation problems. One reason could be the NTK structure from the generator [3]. This could result in some geometry making fitting a neural generator to 2D data extremely hard. One can imagine the particle GANs (removing the generator, you only consider the vector field derived from the discri) has no interacting mechanism for each particle. However, once you add a generator, particles become correlated via the generator's parameters such that there is some kind of mean-field dynamics on particles. The paper needs to take this into account, otherwise the analysis on 2D toy data based on the discriminator guidance cannot really generalize to the original GAN scenario.
>
> Thank you for your insightful questions. We appreciate the opportunity to provide further clarification and discussion. Our response is divided into two parts:
>
> **Regarding the generalization of our results from $2$-dimensional Gaussian mixtures to image generation problems**, we used the $2$-dimensional Gaussian mixture primarily for its visualization benefits. This approach allows us to effectively illustrate the vector fields and discriminator values within a single plot, making the underlying processes more comprehensible. To demonstrate that image generation problems exhibit analogous phenomena to Gaussian mixtures (though, as you have pointed out, training may be simpler for image generation problems), we have added UMAP plots of MNIST to Figure 1. We embedded both the MNIST dataset and the generated samples into a $3$-dimensional space, observing that at the beginning, the generated samples tend to cluster within a small region of the entire space. As training progresses, the generated samples increasingly occupy the entire space spanned by the real data modes, effectively capturing the diversity present in the real dataset. We have also provided details on the effects of different initialization methods in the appendices.
>
> **Regarding the combined dynamics of the generator and the discriminator**, we want to clarify that the dynamics of the particles are not generator-free; rather, the role of the generator is implicit in the particles $Z_i = g_\theta(z_i)$. While we acknowledge the existence of mean-field dynamics on particles as you have pointed out, we recognize that accurately capturing such dynamics is inherently complex and hardly theoretically tractable. Therefore, we chose to address this by considering a class of suboptimal discriminators. To complement Sections 3 and 4, we have added analyses on how these suboptimal discriminators affect the vector field that updates particles and the evolution of steepness.
>
> ----

---

> ### Author Response · Authors · 2024-07-19
> **A Point-by-Point Response to Comments (Part 2)**
>
> ## Responses to Weaknesses (Continued):
>
> ### Early mode collapse
>
> > Another question regarding the early stopping, in my experience, mode collapse sometimes happens at the beginning, i.e., generator directly output single mode images instead of showing the procedure in Figure 8. Can the author give some analysis on this?
>
> Thank you for bringing up this point. Indeed, as we have pointed out at the end of Section 3, mode collapse can happen at the beginning:
>
> *This situation could result in a ``pre-refining'' mode collapse or nonconvergence, preventing GAN training from advancing to the refining phase. Such collapse or nonconvergence might be attributed to detrimental network initialization or imbalanced training of the generator and the discriminator. **Given its infrequency in practice, we will not delve deeper into this topic.***
>
> To help illustrate this, we draw upon the concept of the "plateau phase" in Newton's method for solving nonlinear equations, as described in (Aghili et al., [2024](https://arxiv.org/pdf/2403.03021.pdf)):
>
> *Indeed, Newton’s method is known to converge in two phases. **In the second phase, Newton’s method starts converging**, i.e. the residual decreases with the iterations. This second phase only starts when the residual is small enough, i.e., after the phase where Newton’s method explores the space of solutions. **This first phase corresponds to a plateau** when graphing the residual versus the iteration number.*
>
> Analogously, our discussion primarily centers on the "second" phase, which occurs after GANs navigate the "plateau." Although we acknowledge that mode collapse may occur in the first phase, hindering training from progressing to the second phase, we stress that this occurrence is infrequent under suitable conditions such as proper initialization and balanced training of the generator and the discriminator.
>
> In summary, while early mode collapse is a valid concern, **with appropriate initialization and training strategies, GANs can progress through the initial fitting phase and enter the refining and collapsing phase**, where mode collapse is more prominently analyzed. We hope this clarification addresses your concerns and provides a deeper understanding of our work.
>
> ----

---

> ### Author Response · Authors · 2024-07-19
> **A Point-by-Point Response to Comments (Part 3)**
>
> ## Responses to Requested Changes:
>
> ----
>
> ### Stop gradient operator
>
> > Question for the proof of Theorem 2.1. I understand this proof technique is used in almost all relevant papers mentioned by the author. However, one thing I would like to point out is the once you applied the differentiation operator $\nabla_\theta$, you started considering $\hat{Z_i}$ as a constant which is non-differentiable w.r.t. $\theta$. But, in line 5 of Algorithm 1, $\hat{Z_i}$ does have involved the parameter $\theta$ through $Z_i$. In this case, you have to use a stop gradient approach in the implementation to ensure Theorem 2.1 can be true. But applying the stop gradient operator makes this Theorem less mathematically rigorous.
>
> Thank you for pointing out the issue. We really appreciate your careful review and have made adjustments to the statements of Algorithm 1 and Theorem 2.1. In our revised manuscript, we have explicitly included the use of the stop gradient approach to handle the dependency of $\hat{Z_i}$ on the parameter $\theta$. This ensures the correctness of Theorem 2.1 while maintaining mathematical rigor.
>
> > Theorem 3.1 seems to be redundant. This is just the standard result of sigmoid transformation of the logit output ($log(r(x))$) of a binary classifier. What is the conclusion the author wants to reach at this point? The following implications of Theorem 3.1 are just the empirical description of how the particles evolve with this gradient update. Instead, this vector field decreases the KL divergence in a L2 space [1] or the KL divergence in Wasserstein space [2]. In [2], it shows the vector field of the NSGAN corresponds to the Wasserstein gradient of a certain f-divergence but no longer the Jensen Shannon. I also found the implication after Theorem 3.1 is slightly overlapped with [2] (Section 5.1).
>
> Thank you for pointing out the relevant literature and your observations regarding Theorem 3.1. We appreciate your insights and have made adjustments to our manuscript accordingly. Our response is divided into two parts:
>
> **Intent and Clarification:** We originally included Theorem 3.1 at the beginning of Section 3 to lay the groundwork for explaining the empirical observations in Sections 3.2 and 3.3. We recognize that there is some overlap with Section 5.1 of reference [2], particularly in terms of the implications related to the vector field. That being said, our Theorem primarily aims to characterize the asymptotic behavior of the vector field, which is not the focus of Section 5.1 in [2]. By establishing this foundation, we intended to provide a comprehensive understanding of the empirical results that follow.
>
> **Revisions in the Manuscript:** In the revised version of our manuscript, we have relocated Theorem 3.1, along with its proof and implications, to the appendix. In the main text, we now directly present the conclusions and reference [2] where appropriate. This restructuring ensures that our main contributions remain clear and focused while acknowledging the foundational work presented in [2].
>
> ----
>
> ### More theoretical results in Section 3
>
> > For Sections 3.2 and 3.3, I wish there are more theoretical results. As pointed out earlier, this NSGAN vector field is the velocity field of Wasserstein gradient flow of $f$-divergence.
>
> Thank you for your feedback on Sections 3.2 and 3.3. We appreciate your suggestion for more theoretical results. In our revised manuscript, we have added corresponding theoretical insights to all four modelling scenarios.
>
> We understand your point regarding the NSGAN vector field being the velocity field of the Wasserstein gradient flow of $f$-divergence, and that along this vector field, the $f$-divergence decreases. While these are general theoretical results, we chose to provide data-dependent analyses in our work. This approach allows us to offer specific theoretical insights tailored to each modelling scenario presented in the Sections.
>
> We welcome further discussions and are grateful for your valuable insights that have helped us improve our manuscript.
>
> ----
>
> ### A broader class of Wasserstein distance
>
> > In Section 4.1, the authors claimed "under the 2-Wasserstein distance (Santambrogio, 2015)". If I understand it correctly, Section 2.1 in (Santambrogio, 2015) shows this transport map applies to p-Wasserstein distance not only 2-Wasserstein distance.
>
> Thank you for your meticulous and precise observation regarding Section 4.1. Indeed the transport map discussed in Section 2.1 of (Santambrogio, 2015) applies to the $p$-Wasserstein distance, not just the $2$-Wasserstein distance. We have revised our manuscript accordingly to reflect this broader applicability, making our statement more precise.
>
> ----

---

> ### Author Response · Authors · 2024-07-19
> **A Point-by-Point Response to Comments (Part 4)**
>
> ## Responses to Requested Changes (Continued):
>
> ### More clarifications on Figure 3
>
> > I appreciate the authors' efforts on illustrating differences in Figure 3. While I was reading this part, I was a bit getting lost. I would like to suggest to add a table to summarize which affects which.
>
> Thank you for your positive feedback. We understand that the explanation might have been somewhat confusing. In response to your suggestion, we have added Table 2 to summarize the relationships and clarify which parameters affect which.
>
> ----
>
> ### Shrinking boundary
>
> > In Section 5.1, I did not get why the boundary will shrink. Can the authors give more explanations on that?
>
> Thank you for your question regarding Section 5.1. We appreciate the opportunity to provide a more detailed explanation.
>
> To illustrate why the boundary of the modes will shrink, let us imagine the mode as a high-dimensional sphere composed of many particles, akin to the seeds in a pomegranate. When a particle approaches the boundary of this sphere, it experiences a massive inward force, directed towards the center of the sphere.
>
> What happens to the particle under this immense force? According to our theoretical derivation in Section 5.2 and the experimental results in Section 6.2, the particle is expelled from the sphere. This behavior is consistent for every particle at the boundary of the sphere — they are all pushed away.
>
> As a consequence, the boundary of the sphere is depleted of particles. Given that the sphere is initially defined by the presence of these particles, the boundary naturally contracts until it reaches other particles within the sphere. This explains the observed shrinking of the mode boundaries over time.
>
> We hope this explanation clarifies the reasoning behind the shrinking boundaries.
>
> ----
>
> ## Conclusion
>
> Again, we would like to extend our sincere gratitude to you for your constructive feedback and comments. We hope our responses adequately address the concerns you raised. Should there be any further inquiries or issues, we would be delighted to engage in further discussions.
>
> Best regards,
>
> Paper2643 Authors

---

### Review · Reviewer_yy6E · 2024-06-20

**Summary Of Contributions:**

This paper introduces a novel theoretical framework that explains the training process of Generative Adversarial Networks (GANs). The training process is divided into three sequential phases: fitting, refining, and collapsing. Based on this theoretical framework, the author proposes a novel monitoring metric to serve as an early stopping criterion.

**Audience:**

Yes

**Claims And Evidence:**

No

**Requested Changes:**

Maybe we can combine the fitting and refine stage to one stage.

**Strengths And Weaknesses:**

Strengths:

1) The paper presents a comprehensive literature review and provides strong evidence to support the theory proposed in the paper.

2) The paper also demonstrates the three stages with a practical toy example experiment.

3) The newly proposed early stopping criterion appears to be innovative and could have practical implications.


Weaknesses

1) The analysis presumes that the discriminator is optimal, which does not consider scenarios where the discriminator is sub optimal.

2) In this paper, fitting refers to the process where the generated samples progressively spread to  cover the space that envelopes the mode. When the model is initialized, theoretically, it can produce any result from noise. Consequently, at the outset, it can span the entire space. Although the probability of generating a valid result is low, this paper attempts to verify the existence of a fitting stage through empirical experiments. Since the probability of generating a valid result is low, it can appear that the result space is not fully covered.

---

> ### Author Response · Authors · 2024-07-18
> **Thank You and a Kind Reminder on Submitted Revised Manuscript**
>
> Dear Reviewer yy6E,
>
> Thank you for your detailed and very insightful feedback on our paper. We greatly appreciate the time and effort you have put into reviewing our manuscript and providing such valuable comments.
>
> We hope that addressing the current feedback promptly will help streamline the review process and ensure a smoother progression. As you may have noticed, we submitted the revised version of our manuscript on June 27, incorporating the constructive feedback from you. For any concerns you raised that may not seem directly addressed in the revised manuscript, we will provide detailed responses during the subsequent rebuttal phase.
>
> Once again, we sincerely thank you for your thorough review and valuable suggestions.
>
> Best regards,
>
> Paper2643 Authors

---

> ### Author Response · Authors · 2024-07-19
> **A Point-by-Point Response to Comments (Part 1)**
>
> Dear Reviewer yy6E,
>
> Thank you once again for the time and effort you have dedicated to reviewing our work. Below, we will address your comments point-by-point.
>
> ----
>
> ## Update on Revised Manuscript:
>
> We have submitted another revised version of our manuscript, with changes since the first submission highlighted in **red**, as suggested by the Action Editor.
>
> ----
>
> ## Responses to Weaknesses:
>
> ----
>
> ### Suboptimal discriminators
>
> > The analysis presumes that the discriminator is optimal, which does not consider scenarios where the discriminator is suboptimal.
>
> Thank you for your feedback on Sections 3.2 and 3.3. We appreciate your suggestion for more theoretical results. In our revised manuscript, we have included additional analyses on how a class of suboptimal discriminators affects the vector field that updates particles (complementing Section 3) and the evolution of steepness (complementing Section 4) in the appendices.
>
> Detailedly speaking, we drew inspiration from the mathematical expression of the optimal discriminator and considered a class of suboptimal discriminators of the form:
> $$
> \hat{d}\_\omega(x) = \frac{1}{1 + f(r(x))\big(p\_{\text{data}}(x)/p\_g(x)\big)^{-1}} = \frac{1}{1 + \big(r(x)/f(r(x))\big)^{-1}},
> $$
> where $r(x) = p_{\text{data}}(x)/p_g(x)$, and $f(r(x))$ measures the deviation of the suboptimal discriminator from the optimal one. In Proposition C.1, we demonstrate that when $r(x)$ is small, the suboptimal vector $\nabla \hat{d}\_\omega(x)/(2\hat{d}\_\omega(x))$ and the optimal vector $\nabla d^*\_\omega(x)/(2d^*\_\omega(x))$ align. This implies that during the fitting phase, even with a suboptimal discriminator, we will still observe similar phenomenon as in Figure 2, i.e., the generated samples will move towards the modes. Moreover, we have analyzed the evolution of steepness for such suboptimal discriminators during the refining stage, as detailed in Proposition C.2, where we derived the recursion formula for the steepness dependent on $f$.
>
> In summary, we hope that our revisions provide a more complete theoretical understanding of the influcence of suboptimal discriminators.
>
> ----
>
> ### Clarification of the fitting phase
>
> > In this paper, fitting refers to the process where the generated samples progressively spread to cover the space that envelopes the mode. When the model is initialized, theoretically, it can produce any result from noise. Consequently, at the outset, it can span the entire space. Although the probability of generating a valid result is low, this paper attempts to verify the existence of a fitting stage through empirical experiments. Since the probability of generating a valid result is low, it can appear that the result space is not fully covered.
>
> Thank you for your insightful feedback. We appreciate the opportunity to address your points and provide further clarification. Our response is divided into two parts.
>
> **Firstly, regarding the assertion that "it can span the entire space,"** we have addressed this in **Figure 1** and **Appendix G.1**, where we used UMAP to visualize and analyze the initial distribution of points. Our analyses show that when the latent dimension is less than or equal to the image dimension, the generator maps the noise distribution onto a low-dimensional manifold. This mapping inherently restricts the generator's ability to span the entire space. Conversely, when the latent dimension is larger the image dimension, our experiments consistently show that regardless of the mainstream network initialization methods employed, the initially generated points are concentrated in a relatively small region. These findings indicate that the generator does not span the entire space at the outset, but instead starts in a confined area.
>
> **Secondly, regarding the point that "it can appear that the result space is not fully covered,"** we have adjusted our original description of fitting in the manuscript. The revised explanation now reads: "Roughly speaking, fitting refers to the process where the generated samples progressively spread to cover the space that envelopes the majority of the modes." We believe this adjustment more accurately reflects the fitting process and addresses your concern about the coverage of the result space. To further clarify why the generated samples **are** able to progressively spread to cover the space that envelopes the majority of the modes, we refer to Figure 2. As observed, there are "attraction regions" of varying sizes around each mode (primarily yellow areas). When particles fall into these regions, they are "attracted" towards the modes. This attraction mechanism results in the generated samples progressively spreading out to cover the space that envelopes the majority of the modes.
>
> ----

---

> ### Author Response · Authors · 2024-07-19
> **A Point-by-Point Response to Comments (Part 2)**
>
> ## Responses to Requested Changes:
>
> ### Combining the fitting and refining stages
>
> > Maybe we can combine the fitting and refine stage to one stage.
>
> Thank you for your suggestion to combine the fitting and refining stages into one. We value your perspective and understand the rationale behind your proposal. That said, for the following reasons, we believe it is important to keep these stages distinct.
>
> **Firstly, the fitting and refining stages exhibit very different characteristics.** As shown in Figure 1 and discussed in Appendix G.1, initially, the generated samples are concentrated in a small region (first subfigure of Figure 1). As training progresses, the samples gradually spread out to cover the majority of modes (second, third, fourth, and fifth subfigures of Figure 1). In contrast, the refining stage involves the generated samples becoming more precise and detailed, as illustrated in Figure 6.
>
> **Secondly, it is not feasible to skip the fitting stage and directly enter the refining stage.** As you have mentioned, at the beginning of the training, the probability of generating a valid result is low, so the refining stage cannot be initiated immediately. The fitting stage is essential as it allows the model to explore and cover the space effectively, laying the foundation for refinement. Without this exploration, the model would lack the necessary diversity and coverage, which are crucial for the refining process to improve the quality of the generated samples.
>
> **Thirdly, we have employed different analytical approaches for the fitting and refining stages.** For the fitting stage, in addition to empirical experiments, we have included data-dependent results in the revised manuscript, providing a detailed analysis of the particle updates. For the refining stage, we have examined the relationship between the quality of the generated samples and the steepness of the function, as discussed in our analysis of the steepness in Section 4.
>
> We hope these distinctions justify the separation of the fitting and refining stages and provide a clearer understanding of the training process. Thank you again for your valuable feedback, which has helped us enhance the clarity and rigor of our manuscript.
>
> ----
>
> ## Conclusion
>
> Again, we would like to extend our sincere gratitude to you for your constructive feedback and comments. We hope our responses adequately address the concerns you raised. Should there be any further inquiries or issues, we would be delighted to engage in further discussions.
>
> Best regards,
>
> Paper2643 Authors

---

> ### Author Response · Authors · 2024-08-05
> **Updates on the Revised Manuscript**
>
> Dear Reviewer yy6E,
>
> We have submitted a revised version of our manuscript with changes since the last submission highlighted in **blue**. These updates primarily concern the refining phase, including a clarification of the definition of "refining" at the beginning to prevent potential misunderstandings, which now reads:
> ***This section focuses on the refining phase of GAN training, where generated samples become more refined, reducing the number of samples within the modes and alleviating the mode mixture.***
> We have also provided additional explanations regarding the motivation behind steepness, and an update to Figure 3 to more directly illustrate how steepness impacts the severity of mode mixture. We believe these changes further address your concern over the difference between fitting and refining.
>
> If you have any further questions or concerns, please feel free to let us know. We highly value your feedback and are dedicated to resolving any additional issues that may arise. Thank you sincerely for your time and effort.
>
> Best regards,
>
> Paper2643 Authors

---

> ### Author Response · Authors · 2024-08-23
> **A Follow-Up Response to Fitting and Mode Coverage**
>
> Dear Reviewer yy6E,
>
> We have been consistently considering your comment below and would like to offer an additional response based on our current understanding.
>
> > In this paper, fitting refers to the process where the generated samples progressively spread to cover the space that envelopes the mode. When the model is initialized, theoretically, it can produce any result from noise. Consequently, at the outset, it can span the entire space. Although the probability of generating a valid result is low, this paper attempts to verify the existence of a fitting stage through empirical experiments. Since the probability of generating a valid result is low, it can appear that the result space is not fully covered.
>
> We agree with your probabilistic perspective. Indeed, we believe your view is more macroscopic, whereas our subsequent discussion may lean towards a more theoretically rigorous perspective.
>
> We concur with your statement that "when the model is initialized, theoretically, it can produce any result from noise." This is certainly possible, especially if we consider a continuous distribution formed by an infinite number of initial generation points. However, we also want to point out that only a finite number of points will work, and these points do not initially need to cover the full region. Even if they are clustered in a very small region, as shown in Appendix G.8, **they can still move towards the mode during the fitting stage under the influence of the attraction region** (see the final description in Section 2.2 and Theorem D.2). Furthermore, during the refining stage, we have demonstrated that the number of particles remaining between the modes gradually decreases (see Theorems 4.3 and 4.4). This is why, even if the probability of generating a valid result is low at initialization, it does not prevent particles from moving near the mode during the fitting stage and into the mode during the refining stage.
>
> Moreover, we want to emphasize that **the primary cause of mode collapse originates from the third collapsing stage, where we adopt a dynamical perspective**. In this stage, particles near the mode boundaries are significantly displaced by a large gradient magnitude ($||\nabla d_\omega||$) during the update process ($||\nabla d_\omega/d_\omega||$), and then they are reattracted to specific modes, similar to the refining stage. This may offer a more precise alternative to the initial impression that mode collapse occurs because "the result space is not fully covered" due to the low probability of generating valid results.
>
> We hope this response accurately reflects your concerns, and we are open to further discussion if there are any additional questions.
>
> Best regards,
>
> Paper2643 Authors

---

### Review · Reviewer_ntzb · 2024-08-02

**Summary Of Contributions:**

This paper aims to provide a comprehensive characterization of the training phases of GAN as fitting, refining, and collapsing. Based on toy models, they analyze each phase together with their influence on mode mixture and mode collapsing. They provide theoretical and empirical justification and develop an early stopping criterion to avoid mode collapse.

**Audience:**

Yes

**Claims And Evidence:**

No

**Requested Changes:**

I do not see an easy way to fix the current problems aside from a significant rewriting. I would recommend the authors to focus on the key theoretical tools they discover. For example, steepness, and develop a more rigorous analysis on 1) how steepness would evolve during training, 2) how it relates to mode mixture and mode collapse that is of concern. Also, adding more diverse real-world evidence would strengthen the motivation and help validate the proposed method.

**Strengths And Weaknesses:**

- **Lack of rigor.** The authors make an ambitious attempt at a holistic characterization of the GAN training process, which is commendable, but it still falls short of expectations. The proposed three stages are primarily motivated by observations from a mixture of Gaussian experiments. While this is a good start, it is insufficient to establish the universality of their characterization. Additionally, the authors claim to provide a "novel theoretical framework." However, much of the discussion is a loose combination of theoretical and toy-level experiments, lacking a unified analysis framework. This primarily consists of observed phenomena rather than a cohesive interpretation, which can hardly be considered a "theoretical framework."

- **Lack of Clarity and Consistency.** The authors divide the paper into three parts corresponding to the three phases, but it is difficult to understand the relationship between them and identify the key novelties of this work. Mode collapse, for instance, is extensively discussed in the literature, and changes during training are frequently observed. This writing style makes it challenging to understand the key novelties. The introduction of some key concepts, such as steepness, is not well motivated, as illustrated by a toy case in Fig 3. Consequently, the article jumps between various hand-wavy concepts and observations, without making concrete explanations for each of them.

- **Lack of solid support**. Most claims are justified by specific theoretical or experimental setups, which often seem arbitrary. The proposed early stopping measure is only verified by its own metric (Fig 7,8) and illustrative examples (Fig 8), without quantitative assessment or comparison with prior works. These shortcomings undermine the support for many of the paper's claims.

---

> ### Author Response · Authors · 2024-08-05
> **Thank You and a Point-by-Point Response to Comments (Part 1)**
>
> Dear Reviewer ntzb,
>
> Thank you for the time and effort you have dedicated to reviewing our work. We greatly appreciate your constructive suggestions and apologize for any confusion caused. To address your concerns more clearly, we will first provide **an overview of our responses**, followed by **a point-by-point address of your comments**.
>
> ----
>
> ## Update on Revised Manuscript:
>
> Please note that we have submitted a revised version of our manuscript with changes since the last submission highlighted in **blue**.
>
> ----
>
> ## Overview of Our Responses:
>
> We first provide an overview of our responses, summarized in the table below. We will focus on the clarification of our theoretical framework, the relationship between the three phases, the motivation of key concepts, evidence of our characterization, and a comparison with prior works. At the end, we will address the requested changes. Please note that the table includes supplementary information to aid understanding.
>
> |                                                          | **Fitting**                                                  | **Refining**                                                 | **Collapsing**                                               |
> | -------------------------------------------------------- | ------------------------------------------------------------ | ------------------------------------------------------------ | :----------------------------------------------------------- |
> | **Brief Description**                                    | Particles move towards the modes' directions and fit the modes. | Particles move towards specific modes, decreasing the number of particles within the modes and becoming refined. | Particles at the boundary of modes collapse, eventually leading to mode collapse. |
> | **Theoretical Framework: Particle Model Interpretation** | Visualizing and theoretically deriving the particle evolution field. | Deriving the evolution of steepness throughout the training process. | Using the average norm of the particle update vector as an early stopping metric. |
> | **Theoretical Framework: Continuous Data Augmentation**  | Modeling data in Sections 3.1 and 3.2.                       | Assuming conditions in Theorems 4.1, 4.2, 4.3, and 4.4.      | Formulating the discriminator.                               |
> | **Internal Relationship**                                | Occurs at the beginning, finding the space spanned by modes and laying the foundation for refining. | Occurs after fitting, when particles have found the space spanned by modes but primarily lie within the modes. | Occurs after refining, when particles are already in the modes. |
> | **Key Novelties**                          | Providing both empirical and theoretical results of particle dynamics. | Developing steepness as a measure of mode mixture severity with quantitative results; providing theoretical results regarding lower bounds of steepness; providing the evolution of steepness using particle dynamics. | Approaching mode collapse as the subsequent phase of refining; developing an early stopping metric and early stopping algorithm using particle dynamics. |
> | **Key Concepts**                                         | Particle update vector field; local and global perspectives. | Steepness; measurement of mode mixture severity.             | Overfitting of the discriminator; the scale factor; the number of warm-up training iterations. |

---

> > ### Author Response · Authors · 2024-08-05
> > **A Point-by-Point Response to Comments (Part 2)**
> >
> > ## Responses to Weaknesses:
> >
> > ### Clarification of Our Theoretical Framework
> >
> > > The authors make an ambitious attempt at a holistic characterization of the GAN training process, which is commendable, but it still falls short of expectations. ... The authors claim to provide a "novel theoretical framework." However, much of the discussion is a loose combination of theoretical and toy-level experiments, lacking a unified analysis framework. This primarily consists of observed phenomena rather than a cohesive interpretation, which can hardly be considered a "theoretical framework."
> >
> > > Most claims are justified by specific theoretical or experimental setups, which often seem arbitrary.
> >
> > We appreciate your thoughtful feedback and apologize for any confusion caused. Please allow us to clarify our theoretical framework as introduced at the beginning of Section 2:
> >
> > *... we introduce a novel approach called continuous data augmentation. We move on to discuss how to interpret Divergence GANs, particularly NSGAN, as a particle model. ... The fusion of the two methodologies forms our theoretical framework.*
> >
> > In our revised manuscript, we have also relocated the derivation of the particle evolution field from its original position in Section 3.1 to Section 2.2, emphasizing its vital role within our framework.
> >
> > **In essence, our framework integrates fundamental assumptions about data structure with the intricate dynamics of particle evolution for generated samples.** Specifically, we have formulated Non-Saturating GAN (NSGAN) as a particle model by interpreting the generated samples as particles that evolve under the influence of a field guided by the discriminator (Section 2.2). Furthermore, we have modeled the data distribution $p_{data}$ by augmenting datasets in such a way that they exhibit continuous probability density functions (Section 2.1). This augmentation ensures that our theoretical constructs are applicable to a wide range of real-world data, which often possess smooth and continuous distributions. We have adhered closely to this framework throughout the theoretical aspects of our manuscript. This includes, but is not limited to (all the numbering refers to our latest submission):
> >
> > * For **continuous data augmentation**, we have consistently applied this assumption throughout our modeling of data in Sections 3.1 and 3.2. These subsections use specific mixtures of Gaussians to enhance visualization. Additionally, this assumption plays a vital role in our analysis of in Theorems 4.1, 4.2, 4.3, and 4.4. Specifically, Theorem 4.1 establishes a lower bound for steepness using a general mixture of $1$-dimensional Gaussians. Theorem 4.2 expands upon Theorem 4.1 by addressing high-dimensional scenarios. Theorem 4.3 investigates the evolution of steepness through a particular mixture of Gaussians, while Theorem 4.4 provides quantitative results on how the steepness of generator functions influences the severity of mode mixture, again using a general mixture of $1$-dimensional Gaussians. In Section 5, this assumption is crucial for the formulation of the discriminator.
> >
> > We have reiterated or clarified this assumption in nearly all the previously mentioned sections and theorems to strengthen the original statement made in Section 2.1. This consistent reiteration is intended to emphasize the foundational role of continuous data augmentation in our theoretical framework. However, we recognize that the way these assumptions are presented may unintentionally give the impression that our theoretical or experimental setups are arbitrary. **To address this, we have modified some of the original data-dependent results to include more general settings, such as Theorem 4.1 and Theorem 4.4.**
> >
> > * For **particle evolution dynamics**, we have effectively applied this formulation in Section 3, which includes the visualization of the evolution field. This section explains how generated samples, treated as particles, evolve and result in the fitting phase. In Section 4, using insights from the velocity field of particle evolution, we derive the formula for the evolution of steepness throughout the training process, which directly impacts the severity of mode mixture. In Section 5, we have developed our early stopping metric based on this particle evolution dynamics. Specifically, in Section 5.2, we directly use the average norm of the vector as a key metric, thereby linking the theoretical framework to practical implementation.
> >
> > We believe that the consistent application of our theoretical framework, which integrates continuous data augmentation and particle evolution dynamics, runs throughout our entire manuscript and offers a comprehensive characterization of GANs. We sincerely hope that this clarification addresses your concerns regarding the noted disjointedness of our analysis.
> >
> > ----

---

> > > ### Author Response · Authors · 2024-08-05
> > > **A Point-by-Point Response to Comments (Part 3)**
> > >
> > > ### Relationship Between the Three Phases
> > >
> > > > The authors divide the paper into three parts corresponding to the three phases, but it is difficult to understand the relationship between them and identify the key novelties of this work. Mode collapse, for instance, is extensively discussed in the literature, and changes during training are frequently observed.
> > >
> > > We appreciate the opportunity to clarify the relationship between the three phases presented in our paper. **Rather than viewing these phases as isolated snapshots of the training process, we emphasize their sequential nature over time**, as illustrated in Figure 1, where training epochs progress from left to right.
> > >
> > > While we acknowledge that mode collapse is a widely discussed phenomenon in the literature, existing studies typically treat it as a stand-alone issue. **In contrast, our work approaches mode collapse as the subsequent phase of "refining," during which generated samples may initially exhibit high quality before deteriorating**. We have observed and explored this underexplored phenomenon, where the quality of generated samples from a NSGAN model improves initially and then declines. We believe this novel perspective provides valuable insights into the training dynamics of GANs and enhances the understanding of the nuances of mode collapse.
> > >
> > > We also apologize for not clearly conveying the other novelties of our manuscript. Please allow us to elaborate on them below:
> > >
> > > - Providing both empirical and theoretical results of particle dynamics in the fitting phase.
> > > - Developing steepness as a measure of mode mixture severity with quantitative results in the refining phase.
> > > - Providing theoretical results regarding lower bounds of steepness in the refining phase.
> > > - Detailing the evolution of steepness using particle dynamics in the refining phase.
> > > - Approaching mode collapse as the subsequent phase of refining.
> > > - Developing an early stopping metric and early stopping algorithm using particle dynamics in the collapsing phase.

---

> > > > ### Author Response · Authors · 2024-08-05
> > > > **A Point-by-Point Response to Comments (Part 4)**
> > > >
> > > > ### Motivation of Key Concepts
> > > >
> > > > > The introduction of some key concepts, such as steepness, is not well motivated, as illustrated by a toy case in Fig 3. Consequently, the article jumps between various hand-wavy concepts and observations, without making concrete explanations for each of them.
> > > >
> > > > We appreciate your observations regarding the concept of steepness. Indeed, steepness is a crucial tool for understanding the refining phase of GAN training. In the revised version of our manuscript, we have made several changes to Section 4 to enhance clarity and coherence. **Specifically, we have provided additional explanations to clarify the motivation and significance of steepness within the refining phase.** We have also updated Figure 3 to directly illustrate how steepness impacts the severity of mode mixture, thereby improving the visual representation of this relationship. We believe these changes will offer a clearer understanding of steepness and its role in the dynamics of GAN training. We hereby restate the motivation of steepness in Section 4.1:
> > > >
> > > > *Recall that during the fitting phase, the generated samples progressively spread to cover the space that envelopes most of the modes. By the end of this phase, many generated samples will fall within these modes, resulting in severe mode mixture. According to the update rule for particles, a particle $x$ located within the modes will be pushed in the direction of $\nabla d\_w^\*(x)/d\_w^\*(x)$, which generally points towards the nearest mode. There is a critical point between two adjacent modes where particles that start near this point are pulled apart in opposite directions. As training progresses, there exist two points $z\_1$ and $z\_2$ that are close to each other in the latent space, yet their corresponding images under the generator function $g\_\theta$, namely $g\_\theta(z\_1)$ and $g\_\theta(z\_2)$, can be far apart. When $z\_1$ and $z\_2$ are infinitesimally close, this behavior indicates that the Jacobian of $g\_\theta$ has a large entry. This motivates the concept of "steepness", which generalizes the notion of a derivative in the one-dimensional case.*
> > > >
> > > > We then provide the motivation behind other primary concepts in our manuscript:
> > > >
> > > > - Particle update vector field (in fitting): Used to visualize how particles evolve, helping us understand why and how fitting happens.
> > > > - Local and global perspectives (in fitting): Used to model real-world datasets and provide a better understanding of particle behavior on different scales.
> > > > - Steepness (in refining): As discussed above.
> > > > - Measurement of mode mixture severity (in refining): Used to quantify the severity of steepness. This is based on the property of Gaussian distributions where samples three standard deviations away from the mean are considered rare.
> > > > - Overfitting of the discriminator (in collapsing): Used to explain the cause of collapsing, which is specific to NSGAN.
> > > > - The scale factor (in collapsing): Used to control the extent to which a particle pushed away from a mode cannot be reattracted to the mode.
> > > > - The number of warm-up training iterations (in collapsing): Used to avoid stopping GAN training during the fitting phase.
> > > >
> > > > Please let us know if there are other concepts or observations that are unclear. We are fully committed to addressing your feedback to improve the manuscript.
> > > >
> > > > ----
> > > >
> > > > ### Evidence of Our Characterization
> > > >
> > > > > The proposed three stages are primarily motivated by observations from a mixture of Gaussian experiments. While this is a good start, it is insufficient to establish the universality of their characterization.
> > > >
> > > > Thank you for acknowledging that the Gaussian mixture serves as a solid starting point. We aimed to provide a clear visualization of our proposed phases in the motivation section. That being said, **we have added UMAP plots of MNIST to Figure 1 in the revised manuscript**, illustrating how image distributions resemble Gaussian mixtures, which further supports our characterization. Our framework is grounded in extensive observations and experiments, as evidenced by the results presented in Section 6, demonstrating the typical behavior of NSGAN training.
> > > >
> > > > ----

---

> > > > > ### Author Response · Authors · 2024-08-05
> > > > > **A Point-by-Point Response to Comments (Part 5)**
> > > > >
> > > > > ### Comparison with Prior Works
> > > > >
> > > > > >  The proposed early stopping measure is only verified by its own metric (Fig 7,8) and illustrative examples (Fig 8), without quantitative assessment or comparison with prior works. These shortcomings undermine the support for many of the paper's claims.
> > > > >
> > > > > Thank you for your feedback regarding the need for additional comparisons. In fact, in Figure 8 (previously Figure 9), **we have included a comparison with the FID score to provide a more comprehensive evaluation**. Additionally, the appendix contains results from multiple experiments to further substantiate our findings.
> > > > >
> > > > > Our early stopping measure is intrinsic to GAN training, whereas the FID score is extrinsic, relying on both generated and real samples. This comparison was chosen to highlight the fundamental differences between these metrics. We observed that whenever our metric surpasses the threshold, the FID scores concurrently rise to high values, indicating a significant deterioration in sample quality. This correlation underscores the validity of our early stopping measure.
> > > > >
> > > > > We also acknowledge the limited research specifically addressing early stopping criteria for NSGAN, especially concerning the phenomenon of collapsing after the refining phase. Despite this, we are dedicated to seeking out relevant prior research and will update our manuscript accordingly if suitable references are found.
> > > > >
> > > > > ----

---

> > > > > > ### Author Response · Authors · 2024-08-05
> > > > > > **A Point-by-Point Response to Comments (Part 6)**
> > > > > >
> > > > > > ## Responses to Requested Changes:
> > > > > >
> > > > > > > I do not see an easy way to fix the current problems aside from a significant rewriting. I would recommend the authors to focus on the key theoretical tools they discover. For example, steepness, and develop a more rigorous analysis on 1) how steepness would evolve during training, 2) how it relates to mode mixture and mode collapse that is of concern. Also, adding more diverse real-world evidence would strengthen the motivation and help validate the proposed method.
> > > > > >
> > > > > > Thank you for your detailed feedback and constructive suggestions. We have taken several steps to address the concerns raised and to enhance the clarity and rigor of our manuscript.
> > > > > >
> > > > > > **1. Revisions and Enhancements**
> > > > > >
> > > > > > We have made significant improvements to Section 4, which focuses on the refining phase of GAN training:
> > > > > >
> > > > > > - **Clarification and Definition**: We clarified the definition of "refining" at the beginning of the section to avoid any potential misunderstandings.
> > > > > > - **Motivation for Steepness**: We provided additional explanations regarding the motivation behind the concept of steepness and its significance within the refining phase.
> > > > > > - **Updated Visuals**: Figure 3 has been updated to more directly illustrate how steepness impacts the severity of mode mixture.
> > > > > > - **New Theorems**: We added Theorem 4.1, which establishes a lower bound for steepness in measure-preserving maps for a general mixture of Gaussians. Additionally, Theorem 4.4 presents quantitative results demonstrating how the steepness of generator functions influences the severity of mode mixture.
> > > > > >
> > > > > > These changes aim to provide a clearer understanding of steepness and its role in the refining phase.
> > > > > >
> > > > > > **2. Theoretical Framework and Structure**
> > > > > >
> > > > > > We appreciate your suggestion to focus on the key theoretical tools we discovered. **Our manuscript is built around two core theoretical tools introduced in Section 2: continuous data augmentation and the particle model interpretation of NSGAN.** These tools are fundamental to our analysis and are consistently applied throughout the paper. Please refer to our responses in ***Clarification of Our Theoretical Framework*** for a detailed explanation.
> > > > > >
> > > > > > The reason we do not choose steepness as a core concept throughout all three phases is that steepness is particularly useful for analyzing the refining phase (please refer to Figure 3 for details), especially when combined with the evolution dynamics of particles. However, it is less clear how steepness alone can effectively address the fitting and collapsing phases. Therefore, we introduce the concept of steepness specifically in the refining phase, rather than including it in Section 2 at the outset.
> > > > > >
> > > > > > The paper is organized according to the three phases of the training process to guide readers through the evolution of the model’s behavior and the significant phenomena at each stage. This structure helps present the theoretical developments and their implications in a coherent and logically sequenced manner. For a detailed explanation of this structure, please refer to our responses in ***Relationship Between the Three Phases***.
> > > > > >
> > > > > > **3. Real-World Evidence and Further Research**
> > > > > >
> > > > > > We acknowledge the need for more diverse real-world evidence to strengthen the motivation and validate our proposed methods. Currently, research specifically addressing early stopping criteria for NSGAN, particularly concerning the phenomenon of collapsing after the refining phase, is limited. Nonetheless, we are committed to identifying relevant prior research and will update our manuscript accordingly if suitable references are found.
> > > > > >
> > > > > > ----
> > > > > >
> > > > > > ## Conclusion
> > > > > >
> > > > > > Again, we would like to extend our sincere gratitude to you for your constructive feedback and comments. We hope our responses adequately address the concerns you raised. Should there be any further inquiries or issues, we would be delighted to engage in further discussions.
> > > > > >
> > > > > > Best regards,
> > > > > >
> > > > > > Paper2643 Authors
> > > > > >
> > > > > > --------------------------------------------

---

> ### Author Response · Authors · 2024-08-12
> **Update on Comparison with Duality Gaps**
>
> Dear Reviewer ntzb,
>
> Following a week of focused research and experimentation, we have refined our evaluation of the early stopping metric by incorporating additional comparisons with duality gaps [1][2], as per your suggestion.
>
> Metrics used to evaluate GAN performance generally fall into two categories: domain-specific, which assess the quality of generated images by comparing them to real images using statistical measures, and domain-agnostic, which focus on the optimization process itself without relying on real or generated images. Notably, our early stopping metric (referred to as 'the metric' hereafter) is a domain-agnostic measure. In response to your feedback, we specifically chose to compare our metric with two key representatives from each category: the FID score for domain-specific metrics, which we have already evaluated in our manuscript, and duality gaps [1][2] for domain-agnostic metrics. These selections were made to provide a clear and focused evaluation, as both the FID score and duality gaps are well-established in their respective domains.
>
> In optimization, the duality gap is used to measure the difference between the primal and dual forms of a problem. In GANs, it quantifies the suboptimality of the current generator and discriminator [1]. The perturbed duality gap improves upon the vanilla duality gap by effectively distinguishing between Nash and non-Nash critical points through local perturbations with Gaussian noise [2].
>
> We observe that before the collapsing phase, the metric resembles the patterns seen in the perturbed duality gaps. However, at collapse, the metric sharply increases, while the duality gaps behave inconsistently across datasets. For MNIST, both gaps drop to zero, obscuring whether this indicates mode collapse or convergence. In Fashion MNIST, the vanilla gap drops to zero, but the perturbed gap remains above zero, offering no clear indication. In CIFAR-10, both gaps stay steady, failing to signal collapse. These findings suggest that our metric is a more reliable indicator of mode collapse than the duality gaps. Please refer to Appendix G.8 for more details.
>
> We hope our responses adequately address the concerns you raised. Should there be any further inquiries or issues, we would be delighted to engage in further discussions.
>
> References:
>
> [1] Paulina Grnarova, Kfir Y Levy, Aurelien Lucchi, Nathanael Perraudin, Ian Goodfellow, Thomas Hofmann,
> and Andreas Krause. A domain agnostic measure for monitoring and evaluating GANs. In *Advances in
> Neural Information Processing Systems*, 2019.
>
> [2] Sahil Sidheekh, Aroof Aimen, Vineet Madan, and Narayanan C Krishnan. On duality gap as a measure for monitoring GAN training. In *International Joint Conference on Neural Networks*, 2021.
>
> Best regards,
>
> Paper2643 Authors

---

> > ### Comment · Reviewer_ntzb · 2024-08-20
> > **Thanks for the detailed response**
> >
> > Dear Reviewers,
> >
> > Sorry for the late reply. Thank you for providing a detailed response and for the significant efforts made in the revision. After reading through the response, I feel that the paper writing has been improved (especially the part on steepness). I also missed the FID scores in Figure 8 in my initial reading (300-400 seems very high for FID), and it's good to see that these metrics verify the proposed understanding.
> >
> > The table in Part I is also helpful for understanding the paper's structure, and I think it would be helpful to incorporate it into the paper (maybe with some modifications to make it more concise).
> >
> > In all, while I still hold concerns about the solidness of the analysis and the proposed understanding as a whole, I do think it provides valuable insights into the problem. So it's a bit borderline for me now. I think it would be good to hear more from the other reviewers' comments.

---

### Decision · Action_Editor_MEnC · 2024-10-04

**Recommendation:** Reject

**Comment:**

A core claim of the paper is not supported by sufficient evidence, as described above. As the claim is weaved into the current paper, I do not see an easy way to tweak the paper for it to meet the TMLR acceptance bar. However, several insights in the paper are useful, so that I encourage the authors to submit a major revision at a later time. While the decision must come as a disappointment, I hope the provided feedback has helped and helps to further improve the paper.

**Audience:**

The topic and some of the insights are of interest to the TMLR audience.

**Claims And Evidence:**

This is a borderline paper. On the positive side, the reviewers find that the paper does provide useful insights and the paper has considerably improved during revision. On the negative side, the paper could be clearer and even after multiple interactions with the reviewers, issues remain regarding core claims:

- Theorem 2.1 and Algorithm 1 is due to Yi et al, 2023. The current presentation, in particular the "We show in theorem 2.1 that
algorithm 1 is essentially equivalent to the original NSGAN. " does not give credit to them. Reviewer EpvX has flagged this in their review, but the issue has not been sufficiently addressed in the revision.

- Data augmentation presented in 2.1, which corresponds to adding noise to observed data, when training generative models is not new either.

Together, this means that the first claimed contribution on p3, "a novel theoretical framework" is not supported by sufficient evidence and needs to be removed for the paper to be acceptable. Unfortunately, the claim occurs in multiple places in the paper, so that the paper needed to be substantially rewritten, going beyond a minor revision.

**Resubmission Of Major Revision:**

The authors may consider submitting a major revision at a later time.